# Spurious Feature Diversification Improves Out-of-distribution Generalization

**Yong Lin**[*][†]    **Lu Tan**[*][§]    **Yifan Hao**[*][†]    **Ho Nam Wong**[†]    **Hanze Dong**[†]

**Weizhong Zhang**[‡]    **Yujiu Yang**[§]    **Tong Zhang**[¶]

[†] The Hong Kong University of Science and Technology, [§] Tsinghua University,
[‡] Fudan University [¶] University of Illinois Urbana-Champaign.

## Abstract

Generalization to out-of-distribution (OOD) data is a critical challenge in machine learning. Ensemble-based methods, like weight space ensembles that interpolate model parameters, have been shown to achieve superior OOD performance. However, the underlying mechanism for their effectiveness remains unclear.

In this study, we closely examine WiSE-FT, a popular weight space ensemble method that interpolates between a pre-trained and a fine-tuned model. We observe an unexpected "FalseFalseTrue" phenomenon, in which WiSE-FT successfully corrects many cases where each individual model makes incorrect predictions, which contributes significantly to its OOD effectiveness. To gain further insights, we conduct theoretical analysis in a multi-class setting with a large number of spurious features. Our analysis predicts the above phenomenon and it further shows that ensemble-based models reduce prediction errors in the OOD settings by utilizing a more diverse set of spurious features. Contrary to the conventional wisdom that focuses on learning invariant features for better OOD performance, our findings suggest that incorporating a large number of diverse spurious features weakens their individual contributions, leading to improved overall OOD generalization performance. Additionally, our findings provide the first explanation for the mysterious phenomenon of weight space ensembles outperforming output space ensembles in OOD. Empirically we demonstrate the effectiveness of utilizing diverse spurious features on a MultiColorMNIST dataset, and our experimental results are consistent with the theoretical analysis.

Building upon the new theoretical insights into the efficacy of ensemble methods, we further identify an issue of WiSE-FT caused by the overconfidence of fine-tuned models in OOD situations. This overconfidence magnifies the fine-tuned model's incorrect prediction, leading to deteriorated OOD ensemble performance. To remedy this problem, we propose a novel method called BAlaNced averaGing (BANG) to mitigate the overconfidence problem, which significantly enhances the OOD performance of WiSE-FT.

## 1 Introduction

Machine learning has seen significant advancements recently. However, the assumption that testing samples follow the same distribution as training samples, known as the Identically Independent Distributed (IID) assumption, can be violated in real-world applications. When a machine learning model encounters novel testing samples that it hasn't seen during training, it faces the out-of-distribution (OOD) generalization problem.

Ensemble-based models (ESM) have achieved significant success in addressing OOD problems in recent years. Specifically, denote the input as $x$ and the model as $f_\theta$ with parameter $\theta$. Given two models $f_{\bar\theta}$ and $f_{\tilde\theta}$, existing ESM works typically consider the output space ensemble (OSE)

---

[*]Equal contribution. Corresponding to: Yong Lin<ylindf@connect.ust.hk>

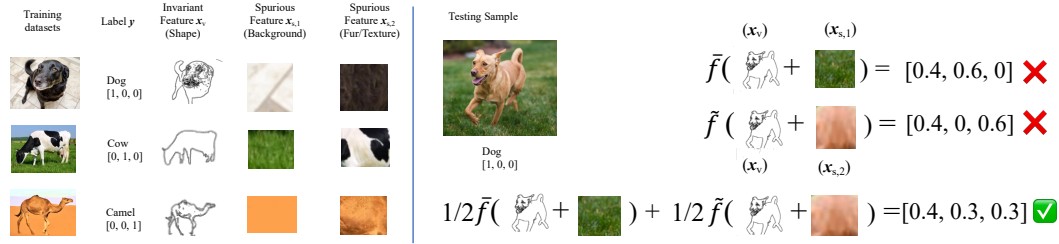

Figure 1: Illustration of FalseFalseTrue phenomenon. Consider to classify camels, cows, and dogs. The invariant feature $x_v$ is the shape of the animal. There are 2 spurious features, i.e., 1) the background $x_{s,1}$, e.g., camels are always on the sand, cows are on grass and dogs are on the floor. 2) the fur of the animals $x_{s,2}$, e.g., camels have brown fur, cows have dotted fur and dogs are all in black in the training dataset. Suppose we fit two models, $\bar{f}$ and $\tilde{f}$, on the training dataset independently. Assume that $\bar{f}$ uses the invariant feature $x_v$ and $x_{s,1}$, and $\tilde{f}$ uses $x_v$ and $x_{s,2}$. $\bar{f}$ and $\tilde{f}$ both correctly predict the label of a sample from the training distribution. Consider an OOD testing sample of a dog with brown fur on the grass. $\bar{f}$ puts a large logit for the cow class since the background(grass) is spuriously correlated with cows, i.e., $\bar{f}(x_v, x_{s,1}) = [0.4, 0.6, 0]$. $\tilde{f}$ puts a large logit for the camel class since the texture(brown fur) is spuriously correlated with camels, i.e., $\tilde{f}(x_v, x_{s,2}) = [0.4, 0, 0.6]$. **Both $\bar{f}$ and $\tilde{f}$ make mistakes on this sample. However, the average of them can make correct prediction**, i.e., $1/2\bar{f}(x_v, x_{s,1}) + 1/2\tilde{f}(x_v, x_{s,2}) = [0.4, 0.3, 0.3]$.

which outputs $f_{\bar{\theta}}(x) + f_{\tilde{\theta}}(x)$ and the weight space ensemble (WSE) which outputs $f_{(\bar{\theta}+\tilde{\theta})/2}(x)$. WSE is also called weight averaging in literature. Wortsman et al. (2022); Wortsman et al.; Rame et al. (2022) show that ESM can significantly improve the OOD performance and WSE outperforms OSE. Many works, e.g., Cha et al. (2021); Rame et al. (2022); Arpit et al. (2022); Rame et al.; Wortsman et al.; Tian et al. (2023); Kumar et al. (2022), adopt WSE to repeatedly improve the SOTA performance on many OOD benchmarks such as DomainBed (Gulrajani & Lopez-Paz, 2020) and ImageNet variants (Wortsman et al., 2022). See Appendix B.1 for more related works.

Consider two types of features for OOD: (1) invariant features that consistently predict the label across distributions, and (2) spurious features that have unstable correlations with the label. Existing OOD theories (Arjovsky et al., 2019; Rosenfeld et al., 2020; Wald et al., 2022; Ahuja et al., 2020; Zhou et al., 2022b) show that an ERM-trained model relying on spurious features can fail in worst-case. ESM, which combines multiple ERM-trained models, may still heavily depend on such features and potentially fail in worst-case scenarios as well. There have been some previous attempts to explain the effectiveness of model ensemble, but they do not offer satisfactory explanations on the overall OOD improvement of ESM. Furthermore, the difference between weight and output space ensemble remains under-explored (a thorough discussion on related works in Appendix B.2).

**An intriguing phenomenon**. To understand the benefits of ESM, we examine the WiSE-FT (Wortsman et al., 2022), which interpolates between a pre-trained and fine-tuned model. When evaluating OOD datasets, we divided them into four groups based on the correctness of predictions made by the individual models. Surprisingly, we found a "FalseFalseTrue" phenomenon: WiSE-FT can correct predictions on samples where both individual models make incorrect predictions. Further, we show that two individual models learn different feature sets, and WiSE-FT utilizes more diverse features. Based on these observations, we then motivate our theory by a toy example (shown in Figure 1). Suppose we have two models, $\bar{f}$ and $\tilde{f}$, for a 3-class classification task. For a sample from the first class, $\bar{f}$ produces logits of $(0.4, 0.6, 0)$, and $\tilde{f}$ produces logits of $(0.4, 0, 0.6)$. The ensemble model's prediction would be $(0.4, 0.3, 0.3)$. This phenomenon can happen when $\bar{f}$ and $\tilde{f}$ learn different subsets of spurious features, represented as $\bar{S}$ and $\tilde{S}$, respectively. Recall that the spurious correlations change in OOD. In the example, $\bar{f}$ generates a high logit (0.6) for the second class influenced by $\bar{S}$, while $\tilde{f}$ produces a high logit (0.6) for the third class influenced by $\tilde{S}$ (details in Section 2).

**A new perspective on OOD generalization**. In Section 3, we extend a popular theoretical setting (Rosenfeld et al., 2020; Wald et al., 2022) to a 3-class classification with multiple spurious features. Our theoretical results predicts the aforementioned phenomenon. We show that ESM incorporates more diverse spurious features, which weakens the contributions of individual spurious feature and

further leads to improved overall OOD performance. We also shed light on the difference between the weight and output space ensemble. Recall that there has been a significant effort in OOD community to learn invariant features and discard spurious features (Arjovsky et al., 2019). However, these approaches have not shown satisfactory performance when applied to real-world datasets (Gulrajani & Lopez-Paz, 2020), which may be due to the fact that invariant learning requires numerous domains (Rosenfeld et al., 2020), strong regularization (Zhou et al., 2022b), and faces additional difficulties induced by non-linearity (Rosenfeld et al., 2020), overparameterization (Lin et al., 2022a), and optimization challenges (Chen et al., 2023c). In contrast, our findings offer a new perspective that **spurious features diversification** actually improves OOD performance, which can be easily implemented as shown in ensemble-based models and has achieved remarkable empirical success. To further verify our findings, we introduce MultiColorMNIST in Section 3.4, a novel variant of CMNIST (Arjovsky et al., 2019), with multiple spurious features. Through empirical analysis, we show that individual models trained on MultiColorMNIST utilize different spurious features, and their ensemble achieves superior OOD performance by leveraging this diversity. Notably, while several methods promote feature diversity to enhance empirical performance, none of them have explored the spurious features diversification from a perspective similar to ours (details in Appendix B.2).

**An improved method**. Our theoretical results indicate that the scaling of $\bar{f}$ and $\tilde{f}$ should be similar to maintain the improvement of the model ensemble. If $\tilde{f}$ is much more confident than $\bar{f}$, resulting in a larger scaling for $\tilde{f}$, the ensemble model can become biased towards $\tilde{f}$. Unfortunately, the scaling issue arises in WiSE-FT, which combines a pre-trained model and a fine-tuned model in the weight space. Empirical evidence shows that the pre-trained model is well calibrated, whereas the fine-tuned model is highly over-confident on OOD datasets, indicating a larger scaling compared to the pre-trained model. Based on these findings, we propose BAlaNced averaGing (BANG), which combines the pre-trained model with a model fine-tuned by over-confidence preventing methods like Label Smoothing and MixUp. We demonstrate that BANG improves vanilla WiSE-FT by approximately 1.9pp in average OOD performance across five ImageNet variants.

To summarize, the following are the main contributions of the paper:

- By examining WiSE-FT, a popular method of ensemble-based models (EBM) that combines the pre-trained and fine-tuned model in the weight space, we discover an unexpected 'FalseFalseTrue' phenomenon that WiSE-FT can correct a large fraction of OOD samples on which both individual models make wrong predictions. We further show that two individual models use different sets of features and WiSE-FT utilizes more diverse features.

- Through theoretical analysis on a multi-class classification problem with multiple spurious features, we provide a natural explanation for the observed phenomenon and show EBM can improve OOD performance through spurious features diversification. Additionally, our findings provide the first-ever explanation for the mysterious phenomenon of weight space ensembles outperforming output space ensembles in OOD scenarios.

- Contrary to the traditional belief that emphasizes the exclusive learning of invariant features for OOD, our findings suggest that incorporating diverse spurious features weakens their individual contributions, leading to improved overall OOD generalization performance. Through experiments on our MultiColorMNIST dataset, which contains multiple spurious features, we provide concrete evidence for the effectiveness of diverse spurious features.

- Based on our theoretical and empirical findings, we show that WiSE-FT can suffer from the over-confidence problem of the fine-tuned model, which skews the ensemble and deteriorates the OOD performance. We further propose a novel method BANG to remedy this problem, and it significantly improves the OOD performance.

## 2 UNDERSTANDING ENSEMBLE-BASED MODELS VIA EXAMINING WISE-FT

**The FalseFalseTrue phenomenon**. In this section, we closely examine WiSE-FT (Wortsman et al., 2022) to obtain intuition on why EBM can improve OOD performance. Specifically, (Wortsman et al., 2022) ensemble pre-trained CLIP and the model fine-tuned on ImageNet in the weight space. In Appendix C.1, we divide each dataset (ImageNet as ID dataset and five ImageNet variants [1]

---

[1] They are ImageNet-V2 (Recht et al., 2019), ImageNet-R (Hendrycks et al., 2021a), ImageNet-A (Hendrycks et al., 2021b), ImageNet Sketch (Wang et al., 2019) and ObjectNet (Barbu et al., 2019). We refer to them as IN-V2, IN-R, IN-A, IN-S, and ObjNet for short. More details in Appendix E

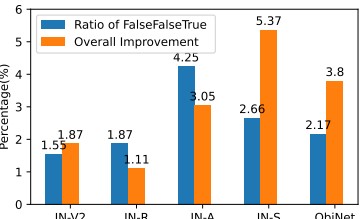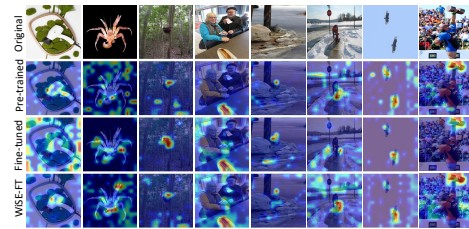

Figure 2: (Left) FalseFalseTrue ratio; (Right) GradCAM feature visualization.

as OOD datasets) into 8 groups by whether the pre-trained, fine-tuned and averaged models make correct predictions. We surprisingly find that WiSE-FT can correct a substantial part of samples on which both the pre-trained and fine-tuned models make mistakes. Specifically, we calculate the number of "FalseFalseTrue" samples, i.e., samples on which WiSE-FT is correct while both the pre-trained and fine-tuned models are incorrect. We then calculate the FalseFalseTrue ratio by dividing FalseFalseTrue number over the dataset size. Figure 2(Left) shows FalseFalseTrue ratio on each OOD dataset and compares it with "overall improvement", which is the accuracy improvement of WiSE-FT over the best of pre-trained and fine-tuned model. We can see that there are substantial parts of FalseFalseTrue samples in each dataset. Refer to Appendix C.1 for more details. It is interesting that the FalseFalseTrue ratio is even higher than the overall improvement in IN-R and IN-A, we provide in-depth analysis and explanation in Appendix C.1 and E.6.

**Illustration on when FalseFalseTrue occurs**. In this part, we try to understand the FalseFalseTrue phenomenon. We first consider the output space ensemble to be similar to the weight space ensemble in this part and will present an analysis of their difference in Section 3. Suppose we want to distinguish from camels, cows, and dogs. There is one invariant feature $\boldsymbol{x}_v$ (the shape of the animal) and two spurious features (the background $\boldsymbol{x}_{s,1}$ and the fur of the animal $\boldsymbol{x}_{s,2}$). Camels are typically found on sand, cows on grass, and dogs on the floor. Camels have brown fur, cows have dotted fur, and dogs are all black in the training dataset. See Fig. 1 for illustration. Suppose we fit two different models, $\bar{f}$ and $\tilde{f}$ on the training dataset. Further assume $\bar{f}$ uses the feature $\boldsymbol{x}_v$ and $\boldsymbol{x}_{s,1}$, and $\tilde{f}$ uses $\boldsymbol{x}_v$ and $\boldsymbol{x}_{s,2}$ [2]. Both $\bar{f}$ and $\tilde{f}$ correctly predict samples from the training distribution. Whereas, for a sample from the testing distribution, e.g., a dog with brown fur ($\boldsymbol{x}_{s,2}$) on the grass ($\boldsymbol{x}_{s,1}$): $\bar{f}$ puts a large logit for the cow class since the background, grass, is spuriously correlated with cow, i.e., $\bar{f}(\boldsymbol{x}_v, \boldsymbol{x}_{s,1}) = [0.4, 0.6, 0]$; $\tilde{f}$ puts a large logit for the camel class since the texture, brown fur, is spuriously correlated with camel, i.e., $\tilde{f}(\boldsymbol{x}_v, \boldsymbol{x}_{s,2}) = [0.4, 0, 0.6]$. Both $\bar{f}$ and $\tilde{f}$ make different mistakes under distributional shifts due to using different spurious features. However, the ensemble of them can make a correct prediction, i.e., $1/2 f_1(\boldsymbol{x}_v, \boldsymbol{x}_{s,1}) + 1/2 f_1(\boldsymbol{x}_v, \boldsymbol{x}_{s,2}) = [0.4, 0.3, 0.3]$.

**Feature visualization**. The reasoning above assumes that individual models utilize different features. GradCam (Selvaraju et al., 2016) visualization of the features used by the pre-trained (zero-shot), fine-tuned, and WiSE-FT in Figure 2(Right) confirms this assumption. The visualization shows that the pre-trained and fine-tuned models rely on different features, while WiSE-FT utilizes more diverse features. Additionally, (Allen-Zhu & Li, 2020) provides empirical evidence supporting the use of diverse features by different DNNs with the same architecture trained on the same datasets (with different initialization). They also provide formal theoretical proof for 2-layer DNNs. We include some of (Allen-Zhu & Li, 2020)'s empirical results in Appendix C.2. Additionally, there is more evidence suggesting that DNNs favor sparse feature representations and discard redundant features (Papyan et al., 2020; Andriushchenko et al., 2023).

# 3 ANALYSIS ON SPURIOUS FEATURE DIVERSIFICATION

## 3.1 THEORETICAL SETTINGS

**Notation**. For simplicity of presentation, we consider a 3-class classification problem, i.e., $\boldsymbol{y} \in \{\boldsymbol{e}_1, \boldsymbol{e}_2, \boldsymbol{e}_3\}$, where $\boldsymbol{e}_i$ denotes the 3-dimensional unit vector with $i$th element equaling 1, e.g., $\boldsymbol{e}_2 = [0, 1, 0]^\top$. In Appendix F.2, we extend the setting to $K$-class classification. $\boldsymbol{a}(k)$ means

---

[2] For simplicity of illustration, we assume that $\bar{f}$ and $\tilde{f}$ learn the same invariant feature. However, this is not necessary for EBM to outperform both individual models, as demonstrated in Section 3

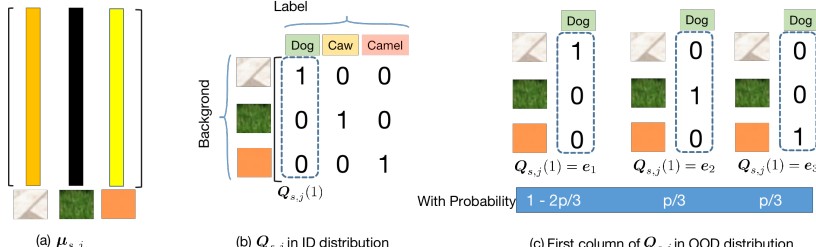

Figure 3: (a) $\boldsymbol{\mu}_{s,j} \in \mathbb{R}^{d \times 3}$ represents a spurious feature, e.g., the background. Each column of $\boldsymbol{\mu}_{s,j}$ is an attribute of the spurious feature, e.g., $\boldsymbol{\mu}_{s,j}(1)$, $\boldsymbol{\mu}_{s,j}(2)$ and $\boldsymbol{\mu}_{s,j}(3)$ are the floor, grass, and sand, respectively. (b) $\boldsymbol{Q}_{s,j} \in \{0,1\}^{3 \times 3}$ represents the relationship between labels and spurious features. In the ID distribution, $\boldsymbol{Q}_{s,j}$ equals $\boldsymbol{I}$, indicating that each spurious feature is perfectly correlated with the corresponding class. (c) In the OOD distribution, spurious correlation can fail, e.g., $\boldsymbol{Q}_{s,j}(1)$ equals $\boldsymbol{e}_2$ with probability $p/3$, indicating the background of the dog is the grass.

the $k$th element of vector $\boldsymbol{a}$, $\boldsymbol{A}(k)$ means the $k$th column of matrix $\boldsymbol{A}$. We use $\boldsymbol{I}_K$ to represent a $K \times K$ identity matrix, e.g., $\boldsymbol{I}_3 = [\boldsymbol{e}_1, \boldsymbol{e}_2, \boldsymbol{e}_3]$. We omit the subscript of $\boldsymbol{I}$ when no confusion arises.

Suppose we have $d_v$ invariant features $\{\boldsymbol{x}_{v,i}\}_{i=1}^{d_v}$ and $d_s$ spurious features $\{\boldsymbol{x}_{s,j}\}_{j=1}^{d_s}$ where $\boldsymbol{x}_{v,i}, \boldsymbol{x}_{s,j} \in \mathbb{R}^d$ and the whole feature $\boldsymbol{x} \in \mathbb{R}^{d \times (d_s + d_v)}$ is the concatenation of them, i.e., $\boldsymbol{x} = \text{Concat}\left(\{\boldsymbol{x}_{v,i}\}_{i=1}^{d_v} \cup \{\boldsymbol{x}_{s,j}\}_{j=1}^{d_s}\right) = [\boldsymbol{x}_{v,1}, \ldots, \boldsymbol{x}_{v,d_v}, \boldsymbol{x}_{s,1}, \ldots, \boldsymbol{x}_{s,d_s}]$. Consider that each model $f$ is composed of a featurizer $\Phi \in \{0,1\}^{d_v + d_s}$ and a classifier $\boldsymbol{w} \in \mathbb{R}^{d \times 3}$. $\Phi$ first selects feature by $\boldsymbol{x}\Phi$. For example, suppose $\boldsymbol{x} = [\boldsymbol{x}_1, \boldsymbol{x}_2, \boldsymbol{x}_3]$ and $\Phi = [1,1,0]^\top$, then $\boldsymbol{x}\Phi = \boldsymbol{x}_1 + \boldsymbol{x}_2$. Then the classifier $\boldsymbol{w} \in \mathbb{R}^{d \times 3}$ is fit based on the features selected by $\Phi$ as $\boldsymbol{w} = \arg\min_{\boldsymbol{v} \in \mathbb{R}^{d \times 3}} \mathcal{R}_{id}(\boldsymbol{v}, \Phi) = \arg\min_{\boldsymbol{v} \in \mathbb{R}^{d \times 3}} \mathbb{E}_{(\boldsymbol{x},\boldsymbol{y}) \sim \mathcal{D}_{id}}[\ell(\boldsymbol{v}^\top(\boldsymbol{x}\Phi), \boldsymbol{y})]$, where $\ell$ is the cross-entropy loss function and $\mathcal{D}_{id}$ is the ID distribution. (**Remark**: Refer to Appendix D.1 for detailed discussions on the setting.)

Following (Rosenfeld et al., 2020; Wald et al., 2022), we consider that each $\boldsymbol{x}_{v,i}$ and $\boldsymbol{x}_{s,j}$ are generated from the label $\boldsymbol{y}$ with the *latent* invariant features $\boldsymbol{\mu}_{v,i}$ and spurious features $\boldsymbol{\mu}_{s,i}$, where $\boldsymbol{\mu}_{v,i}, \boldsymbol{\mu}_{s,j} \in \mathbb{R}^{d \times 3}$. The full data generation process is:

**Definition 1** (Data Generation Process). *The whole data generation process is as follows:*
$$\boldsymbol{y} \sim \text{Unif}\{\boldsymbol{e}_1, \boldsymbol{e}_2, \boldsymbol{e}_3\}, \boldsymbol{x} = \text{Concat}\left(\{\boldsymbol{x}_{v,i}\}_{i=1}^{d_v} \cup \{\boldsymbol{x}_{s,j}\}_{j=1}^{d_s}\right),$$
$$\mathbb{P}_\theta(\boldsymbol{x}_{v,i} \mid \boldsymbol{y}) = \mathcal{N}\left(\boldsymbol{\mu}_{v,i}\boldsymbol{Q}_{v,i}\boldsymbol{y}, \sigma^2\boldsymbol{I}_d\right), \mathbb{P}_\theta(\boldsymbol{x}_{s,j} \mid \boldsymbol{y}) = \mathcal{N}\left(\boldsymbol{\mu}_{s,j}\boldsymbol{Q}_{s,j}\boldsymbol{y}, \sigma^2\boldsymbol{I}_d\right), \forall i, j. \quad (1)$$
*where* $\boldsymbol{Q}_{v,i}, \boldsymbol{Q}_{s,j} \in \{0,1\}^{3 \times 3}$. *Further,* $\boldsymbol{Q}_{v,i} = \boldsymbol{I}_3 = [\boldsymbol{e}_1, \boldsymbol{e}_2, \boldsymbol{e}_3]$ *always hold. In the ID distribution* $\mathcal{D}_{id}$, $\boldsymbol{Q}_{s,j} = \boldsymbol{I}_3$; *and in OOD* $\mathcal{D}_{ood}$, *the $k$th column of $\boldsymbol{Q}$, i.e.,* $\boldsymbol{Q}_{s,j}(k)$, *is as follows for* $k = 1,2,3$:
$$\boldsymbol{Q}_{s,j}(k) = \begin{cases} \boldsymbol{e}_k, & \text{with probability } 1 - p \\ \text{Unif}\{\boldsymbol{e}_1, \boldsymbol{e}_2, \boldsymbol{e}_3\}, & \text{with probability } p. \end{cases}$$

**The intuition of the data generation process**. We consider the example in Figure 1. Figure 3 shows the intuition of $\boldsymbol{\mu}_{s,j}$ and $\boldsymbol{Q}_{s,j}$. Suppose the spurious feature $\boldsymbol{\mu}_{s,j}$ is the background in Figure 1. Here $\boldsymbol{\mu}_{s,j} = [\boldsymbol{\mu}_{s,j}(1), \boldsymbol{\mu}_{s,j}(2), \boldsymbol{\mu}_{s,j}(3)] \in \mathbb{R}^{d \times 3}$ and each column $\boldsymbol{\mu}_{s,j}(k)$ for $k = 1,2,3$ represents a specific attribute that is associated with class $k$ in the training set. In other words, $\boldsymbol{\mu}_{s,j}(1), \boldsymbol{\mu}_{s,j}(2)$, and $\boldsymbol{\mu}_{s,j}(3)$ represent 3 attributes of background, namely, floor, grass, and sand, which are correlated with dog, cow, and camel, respectively. Consider a dog image (i.e., $\boldsymbol{y} = \boldsymbol{e}_1 = [1,0,0]$ ). We have $\boldsymbol{\mu}_{s,j}\boldsymbol{Q}\boldsymbol{y}|_{\boldsymbol{y}=\boldsymbol{e}_1} = \boldsymbol{\mu}_{s,j}\boldsymbol{Q}_{s,j}(1)$ and [3] further

(a) In the ID distribution $\mathcal{D}_{id}$, $\boldsymbol{Q}_{s,j}(1) = \boldsymbol{e}_1$ and $\boldsymbol{\mu}_{s,j}\boldsymbol{Q}_{s,j}\boldsymbol{y}|_{\boldsymbol{y}=\boldsymbol{e}_1} = \boldsymbol{\mu}_{s,j}\boldsymbol{e}_1 = \boldsymbol{\mu}_{s,j}(1)$. Then $\boldsymbol{x}_{s,j} = \mathcal{N}(\boldsymbol{\mu}_{s,j}(1), \sigma\boldsymbol{I})$, indicating that in $\mathcal{D}_{id}$ the background of the dog (i.e., $\boldsymbol{y} = \boldsymbol{e}_1$) is the floor (i.e., $\boldsymbol{\mu}_{s,j}(1)$).

(b) In the OOD distribution $\mathcal{D}_{ood}$, $\boldsymbol{Q}_{s,j}(1) = \boldsymbol{e}_1$ with probability $1 - p$ and $\boldsymbol{Q}_{s,j}(1) \sim \text{Unif}\{\boldsymbol{e}_1, \boldsymbol{e}_2, \boldsymbol{e}_3\}$ with probability $p$. Then we have the following:
$$\boldsymbol{\mu}_{s,j}\boldsymbol{Q}_{s,j}\boldsymbol{y}|_{\boldsymbol{y}=\boldsymbol{e}_1} = \begin{cases} \boldsymbol{\mu}_{s,j}(1), & \text{with probability } 1 - p \\ \text{Unif}\{\boldsymbol{\mu}_{s,j}(1), \boldsymbol{\mu}_{s,j}(2), \boldsymbol{\mu}_{s,j}(3)\}, & \text{with probability } p, \end{cases}$$

---

[3]Specifically, $\boldsymbol{Q}\boldsymbol{y}|_{\boldsymbol{y}=\boldsymbol{e}_1} = \boldsymbol{Q}[1,0,0]^\top = \boldsymbol{Q}_{s,j}(1)$, where $\boldsymbol{Q}_{s,j}(1)$ is the first column of $\boldsymbol{Q}_{s,j}$.

indicating that in the OOD distribution the background of the dog (i.e., $\boldsymbol{y} = \boldsymbol{e}_1$) is the floor (i.e., $\boldsymbol{\mu}_{s,j}(1)$) with probability $1 - p$ and is randomly drawn from floor, grass, and sand (i.e., $\boldsymbol{\mu}_{s,j}(1)$, $\boldsymbol{\mu}_{s,j}(2)$, and $\boldsymbol{\mu}_{s,j}(3)$) with $p$. In other words, $p$ is the probability that spurious correlation no-longer holds and a larger $p$ indicates larger distributional shift.

**Remark.** Our data generation process extends the setting of (Wald et al., 2022; Rosenfeld et al., 2020) to a 3-class classification problem with multiple features. This extension aligns with the intuition behind popular multi-class datasets used in empirical studies on OOD generalization, such as FullColorMNIST, ColoredObject, and CifarMNIST (Zhang et al., 2021; Lin et al., 2022a; Zhou et al., 2022b;a; Ahmed et al., 2021). Take ColoredObject for example, correlations between classes and background colors exist in the training dataset but fail with a certain probability in OOD.

**Definition 2** (Individual models). *Denote the whole invariant feature set as $\mathcal{V} := \{\boldsymbol{x}_{v,i}\}_{i=1}^{d_v}$ and spurious feature set $\mathcal{S} := \{\boldsymbol{x}_{s,j}\}_{j=1}^{d_s}$. Consider $\bar{f} = (\bar{\Phi}, \bar{\boldsymbol{w}})$ and $\tilde{f} = (\tilde{\Phi}, \tilde{\boldsymbol{w}})$. Suppose $\bar{\Phi}$ learns $\bar{\mathcal{V}} \subset \mathcal{V}$ and $\bar{\mathcal{S}} \subset \mathcal{S}$, and $\tilde{\Phi}$ learns $\tilde{\mathcal{V}} \subset \mathcal{V}$ and $\tilde{\mathcal{S}} \subset \mathcal{S}$. Denote $|\tilde{\mathcal{V}}| = \tilde{n}_v$, $|\tilde{\mathcal{S}}| = \tilde{n}_s$, $|\bar{\mathcal{V}}| = \bar{n}_v$, $|\bar{\mathcal{S}}| = \bar{n}_s$, $|\tilde{\mathcal{V}} \cap \bar{\mathcal{V}}| = n_{vo}$, and $|\tilde{\mathcal{S}} \cap \bar{\mathcal{S}}| = n_{so}$. Specifically, we have $\boldsymbol{x}\bar{\Phi} = \sum_{\boldsymbol{x}_v \in \bar{\mathcal{V}}} \boldsymbol{x}_v + \sum_{\boldsymbol{x}_s \in \bar{\mathcal{S}}} \boldsymbol{x}_s, \bar{\boldsymbol{w}} = \arg\min_{\boldsymbol{v} \in \mathbb{R}^{d \times 3}} \mathcal{R}_{id}(\boldsymbol{v}, \bar{\Phi})$, and $\boldsymbol{x}\tilde{\Phi} = \sum_{\boldsymbol{x}_v \in \tilde{\mathcal{V}}} \boldsymbol{x}_v + \sum_{\boldsymbol{x}_s \in \tilde{\mathcal{S}}} \boldsymbol{x}_s, \tilde{\boldsymbol{w}} = \arg\min_{\boldsymbol{v} \in \mathbb{R}^{d \times 3}} \mathcal{R}_{id}(\boldsymbol{v}, \tilde{\Phi})$.*

**Definition 3** (Output space ensemble (OSE)). *Given the two individual models defined in Definition 2, the prediction of the the output space ensemble is $f_{ose}(\boldsymbol{x}) = \frac{1}{2}(\bar{\boldsymbol{w}}^\top(\boldsymbol{x}\bar{\Phi}) + \tilde{\boldsymbol{w}}^\top(\boldsymbol{x}\tilde{\Phi}))$.*

The predicted class of the sample $(\boldsymbol{x}, \boldsymbol{y})$ is the class with the maximum logit. Specifically, denote the logit as $\hat{l} = f(\boldsymbol{x})$. The predicted class is $\hat{k} = \arg\max_{h \in \{1,2,3\}} \hat{l}(h)$ where $\hat{l}(h)$ of the $h$th dimension of the logit $\hat{l}$. The model makes correct prediction if $\mathbb{I}(e_{\hat{k}} = \boldsymbol{y})$ holds where $\mathbb{I}$ is the indicator function. The accuracy is $\mathcal{A}(f) = \mathbb{E}_{\boldsymbol{x},\boldsymbol{y}}[\mathbb{I}(e_{\hat{k}} = \boldsymbol{y})]$. We denote the OOD accuracy as $\mathcal{A}_{\text{ood}}(f) = \mathbb{E}_{\boldsymbol{Q}_s}\left[\mathbb{E}_{\boldsymbol{x},\boldsymbol{y}}[\mathbb{I}(e_{\hat{k}} = \boldsymbol{y})|\boldsymbol{Q}_s]\right]$, where we use $\boldsymbol{Q}_s$ as a short hand for $\boldsymbol{Q}_{s,1}, \ldots, \boldsymbol{Q}_{s,d_s}$. We discuss the metric in Appendix D.4. We defer the analysis of ID accuracy to Appendix D.5 since we consider infinite samples and the ID accuracy of all considered models are all close to 1.

**Assumption 1** (Small Noise). *Denote $n'_v$ and $n'_s$ as the the maximum number of invariant features and spurious features that a model can learn, respectively. We need the overall noise to be small to satisfy $\boldsymbol{F}^K(\frac{1}{\sigma(n'_v + n'_s)}) \geq 1 - \epsilon$, in which $\boldsymbol{F}$ is the cumulative distribution function of standard Gaussian random variable, and $K$ refers to the class number (here we analyze the case $K = 3$).*

**Remark.** Since we impose random noise on each feature, e.g., $\boldsymbol{x}_{v,i} = \boldsymbol{\mu}_{v,i} + \boldsymbol{z}$ where $\boldsymbol{z} \sim \mathcal{N}(0, \sigma^2 \boldsymbol{I}_d)$ where $\boldsymbol{I}_d$ is a d-dimensional identity matrix and $d \gg d_v + d_s$, it is natural to assume the overall noise is controlled, e.g., we have $\epsilon \leq 10^{-6}$ when $K = 10$, $\sigma = 1/100$, $n'_v + n'_s = 20$.

**Assumption 2** (Orthogonal features (Wald et al., 2022; Allen-Zhu & Li, 2020)). *(1) $\|\boldsymbol{\mu}_{v,i}(k)\|_2 = 1$ and $\|\boldsymbol{\mu}_{s,j}(k)\|_2 = 1$ for $i = 1, \cdots, d_v$, $j = 1, \cdots, d_s$, $k = 1, 2, 3$. (2) $\boldsymbol{v}_i(k) \perp \boldsymbol{v}_{i'}(k')$ for any $(i, k) \neq (i', k')$, $k, k' = 1, 2, 3$, $\boldsymbol{v}_i, \boldsymbol{v}_{i'} \in \{\boldsymbol{\mu}_{v,1}, \cdots, \boldsymbol{\mu}_{v,d_v}, \boldsymbol{\mu}_{s,1}, \ldots, \boldsymbol{\mu}_{s,d_s}\}$.*

## 3.2 THEORETICAL RESULTS

We first show the intuition on the simple Example 1 and then extend to the general setting in Def. 3:

**Example 1** (Illustrative examples). *Consider that there are totally 4 invariant features $\{\boldsymbol{x}_{v,i}\}_{i=1}^4$ and 6 spurious features $\{\boldsymbol{x}_{s,j}\}_{j=1}^6$, and two individual models $(\bar{\boldsymbol{w}}, \bar{\Phi})$ and $(\tilde{\boldsymbol{w}}, \tilde{\Phi})$ learn non-overlapped features as $\boldsymbol{x}\bar{\Phi} = \sum_{i=1,2} \boldsymbol{x}_{v,i} + \sum_{j=1,2,3} \boldsymbol{x}_{s,j}$, and $\boldsymbol{x}\tilde{\Phi} = \sum_{i=3,4} \boldsymbol{x}_{v,i} + \sum_{j=4,5,6} \boldsymbol{x}_{s,j}$.*

**Proposition 1** (Illustrative examples). *Consider Example 1, suppose Assumption 1 and 2 hold, and there are infinite ID and OOD samples. Omitting small terms containing $\epsilon$, we have $\mathcal{A}_{ood}(\bar{f}) = \mathcal{A}_{ood}(\tilde{f}) = 1 - \frac{1}{9}p^3$, and $\mathcal{A}_{ood}(f_{ose}) = 1 - \frac{2p^5}{81} - \frac{17p^6}{729}$.*

We can see that OSE improves OOD by $\mathcal{A}_{ood}(f_{\text{ose}}) - \max\{\mathcal{A}_{ood}(\bar{f}), \mathcal{A}_{ood}(\tilde{f})\} > 1/81p^3$.

**Intuition of the proof** (Full proof in Appendix F.1). Let's consider the samples of first class $\boldsymbol{y} = \boldsymbol{e}_1 = [1, 0, 0]$. Model $(\bar{\boldsymbol{w}}, \bar{\Phi})$ has $\boldsymbol{x}\bar{\Phi}|_{\boldsymbol{y}=\boldsymbol{e}_1} = \sum_{i=1}^2 \boldsymbol{\mu}_{v,i}\boldsymbol{Q}_{v,i}(1) + \sum_{j=1}^3 \boldsymbol{\mu}_{s,j}\boldsymbol{Q}_{s,j}(1) + z$ where $z \sim \mathcal{N}(0, 5\sigma^2 \boldsymbol{I}_d)$. By Lemma 5, we have $\bar{\boldsymbol{w}}(k) = \sum_{i=1}^2 \boldsymbol{\mu}_{v,i}(k) + \sum_{j=1}^3 \boldsymbol{\mu}_{s,j}(k)$ for each class $k = 1, 2, 3$. Omitting the small noise term, the predicted logit for class $k$ is $\bar{\boldsymbol{w}}(k)^\top(\boldsymbol{x}\bar{\Phi})|_{\boldsymbol{y}=\boldsymbol{e}_1} =$

$\sum_{i=1}^2 \boldsymbol{\mu}_{v,i}(k)^\top (\boldsymbol{\mu}_{v,i} \boldsymbol{Q}_{v,i}(1)) + \sum_{j=1}^3 \boldsymbol{\mu}_{s,j}(k)^\top (\boldsymbol{\mu}_{s,j} \boldsymbol{Q}_{s,j}(1))$ . The model will mistakenly predict $\boldsymbol{e}_2$ on the samples with true label $\boldsymbol{e}_1$ when $\bar{\boldsymbol{w}}(1)^\top \boldsymbol{x} \bar{\Phi}|_{y=\boldsymbol{e}_1} < \bar{\boldsymbol{w}}(2)^\top \boldsymbol{x} \bar{\Phi}|_{y=\boldsymbol{e}_1}$. This will happen when the three events $\{\boldsymbol{Q}_{s,j}(1) = \boldsymbol{e}_2\}_{j=1}^3$ simultaneously happen in OOD (see Appendix D.7 for detailed discussion). Each event occurs with a probability of $p/3$, resulting in a combination probability of $p^3/27$. This means that with a probability of $p^3/27$, we encounter an OOD scenario where the model $\bar{f} = (\bar{\boldsymbol{w}}, \bar{\Phi})$ incorrectly predicts almost all samples from the first class $\boldsymbol{e}_1$ as the second class $\boldsymbol{e}_2$. This failure occurs because all three spurious features happen to have values that are spuriously correlated with $\boldsymbol{e}_2$ in the training dataset. In other words, the three spurious features dominate the prediction of $\boldsymbol{e}_2$, overshadowing the two invariant features that predict the true label $\boldsymbol{e}_1$. For the OSE model, we have $\bar{\boldsymbol{w}}(k)^\top (\boldsymbol{x} \bar{\Phi}) + \tilde{\boldsymbol{w}}(k)^\top (\boldsymbol{x} \tilde{\Phi})|_{y=\boldsymbol{e}_1} = \sum_{i=1}^4 \boldsymbol{\mu}_{v,i}(k)^\top (\boldsymbol{\mu}_{v,i} \boldsymbol{Q}_{v,i}(1)) + \sum_{j=1}^6 \boldsymbol{\mu}_{s,j}(k)^\top (\boldsymbol{\mu}_{s,j} \boldsymbol{Q}_{s,j}(1))$. The model will mistakenly predict $\boldsymbol{e}_2$ on the samples with true label $\boldsymbol{e}_1$ when at least five of the six events $\{\boldsymbol{Q}_{s,j}(1) = \boldsymbol{e}_2\}_{j=1}^6$ simultaneously happen in OOD (see Appendix D.6 for details), whose probability is much less than that of $\bar{f}$. Intuitively, the failure probability of the averaged model is smaller as it utilizes more spurious features, which are less likely to make the same mistakes.

**Proposition 2** (General Results for OSE). *Consider Definition 1-3, Assumption 1-2 hold, and infinite ID and OOD samples. Omitting small constants involving $\epsilon$, we have* $\mathcal{A}_{ood}(\bar{f}) = F_p\left(\frac{(1-p)\bar{n}_s + \bar{n}_v}{\sqrt{\bar{n}_s}}\right)$, $\mathcal{A}_{ood}(\tilde{f}) = F_p\left(\frac{(1-p)\tilde{n}_s + \tilde{n}_v}{\sqrt{\tilde{n}_s}}\right)$, *and* $\mathcal{A}_{ood}(f_{ose}) = F_p\left(\frac{(1-p)(\tilde{n}_s + \bar{n}_s) + (\tilde{n}_v + \bar{n}_v)}{\sqrt{\tilde{n}_s + \bar{n}_s + 2n_{so}}}\right)$.

Here $F_p(x)$ is a cumulative density function (CDF) parameterized by $p$ as defined in Appendix F.2, which is monotonically increasing with $x$ as shown in Figure 4(a). Suppose two individuals learns the same number of features with no-overlap, i.e., $\tilde{n}_v = \bar{n}_v = n_v$, $\bar{n}_s = \tilde{n}_s = n_s$, and $n_{vo} = n_{so} = 0$, we have $\mathcal{A}_{ood}(f_{ose}) = F_p\left(\sqrt{2}t\right)$ and $\mathcal{A}_{ood}(\bar{f}) = \mathcal{A}_{ood}(\bar{f}) = F_p(t)$ where $t = (1-p)\sqrt{n_s} + \frac{n_v}{\sqrt{n_s}}$, indicating that $f_{ose}$ is better than $\bar{f}$ since $F(\cdot)$ is monotonically increasing.

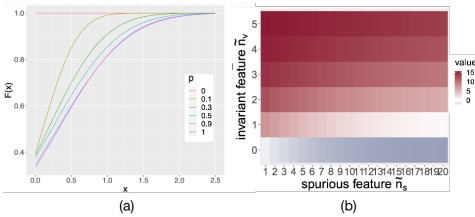

Figure 4: (a) Illustration of of $F(x)$; (b) $\mathcal{A}_{ood}(f_{ose}) - \mathcal{A}_{ood}(\bar{f})$ in Example 2;

**Example 2.** *Consider $p = 0.9$ and two individual models learn none overlapped, i.e., $n_{vo} = n_{so} = 0$, fixing $\bar{n}_v = 5, \bar{n}_s = 20$, and vary $\tilde{n}_v = 0, 1, .., 5$ and $\tilde{n}_s = 0, 1, ..., 20$.*

Figure 4(b) illustrates $\mathcal{A}_{ood}(f_{ose}) - \mathcal{A}_{ood}(\bar{f})$ on Example 2. $f_{ose}$ achieves better OOD performance than $\bar{f}$ in most cases. One exception is that if $\tilde{f}$ is much weaker than $\bar{f}$, e.g., $\bar{f}$ learns 5 invariant features but $\tilde{f}$ learns 0 invariant features, the ensemble model $f_{ose}$ is inferior than $\bar{f}$.

### 3.3 THE DIFFERENCE BETWEEN THE OUTPUT AND WEIGHT SPACE ENSEMBLE

It is an open problem on the difference between output space ensemble (OSE) and WSE (referred as OSE-WSE difference). Furthermore, the mysterious phenomenon of weight space ensembles outperforming output space ensembles in OOD scenarios has puzzled researchers (Wortsman et al., 2022; Wortsman et al.; Rame et al., 2022). We shed light on this by our bilinear theoretical model $\boldsymbol{w}^\top \boldsymbol{x} \Phi$:

**Definition 4** (Weight space ensemble (WSE)). *Given the two individual models defined in Definition 2, the prediction of the WSE is $f_{wse}(\boldsymbol{x}) = \frac{1}{4}(\bar{\boldsymbol{w}} + \tilde{\boldsymbol{w}})^\top \left(\boldsymbol{x}(\bar{\Phi} + \tilde{\Phi})\right)$.*

In Appendix D.2, we show that the OSE-WSE difference in a 2-layer DNN is closely connected with the OSE-WSE difference captured by our models in Definition 3- 4.

**Proposition 3** (General Results for WSE). *Consider Definition 1-3, Assumption 1-2, and infinite ID and OOD samples. Omitting small constants involving $\epsilon$, we have $\mathcal{A}_{ood}(f_{wse}) = F_p\left(\frac{(1-p)(\tilde{n}_s + \bar{n}_s + 2n_{so}) + (\tilde{n}_v + \bar{n}_v + 2n_{vo})}{\sqrt{\tilde{n}_s + \bar{n}_s + 14n_{so}}}\right)$.*

Comparing Proposition 2 and 3, we can see that the only difference between $\mathcal{A}_{ood}(f_{wse})$ and $\mathcal{A}_{ood}(f_{ose})$ is the number of overlapped invariant and spurious features learned by individual

models, i.e., $n_{vo}$ and $n_{so}$. Specifically, when $\bar{\Phi}$ and $\tilde{\Phi}$ selects no overlapped features, $f_{\text{WSE}}$ and $f_{\text{OSE}}$ makes the same prediction since $\boldsymbol{x}\bar{\Phi} \perp \tilde{\boldsymbol{w}}$ and $\boldsymbol{x}\tilde{\Phi} \perp \bar{\boldsymbol{w}}$ by Assumption 2 and further $(\bar{\boldsymbol{w}} + \tilde{\boldsymbol{w}})^\top \left( \boldsymbol{x}(\bar{\Phi} + \tilde{\Phi}) \right) \propto \bar{\boldsymbol{w}}^\top \boldsymbol{x}\bar{\Phi} + \tilde{\boldsymbol{w}}^\top \boldsymbol{x}\tilde{\Phi}$. When there is overlapped features: (a) for WSE, the coefficient of overlapped features is amplified by 2 in $\bar{\Phi} + \tilde{\Phi}$, and further amplified twice in $\bar{\boldsymbol{w}} + \tilde{\boldsymbol{w}}$. This results in coefficient of the overlapped feature becoming 4 in $(\bar{\boldsymbol{w}} + \tilde{\boldsymbol{w}})^\top \boldsymbol{x}(\bar{\Phi} + \tilde{\Phi})$. (b) for OSE, i.e., $\bar{\boldsymbol{w}}^\top \boldsymbol{x}\bar{\Phi} + \tilde{\boldsymbol{w}}^\top \tilde{\boldsymbol{x}}\Phi$, the coefficient of the overlapped feature is 2. See Appendix D.7.1 for a detailed discussion. In Appendix D.7.2, we provide conditions when $f_{\text{WSE}}$ outperforms $f_{\text{OSE}}$, in addition with simulation results and supportive experiments. **Our findings provide the first-ever explanation for the mysterious phenomenon of weight space ensembles outperforming output space ensembles in OOD.**

### 3.4 EXPERIMENTAL VERIFICATION ON MULTICOLORMNIST

Previous efforts in OOD community have focused on learning invariant features and discarding spurious features (Arjovsky et al., 2019). However, these approaches have not performed well on real-world datasets (Rosenfeld et al., 2020). This could be due to the requirements of invariant learning, such as the need for numerous domains (Rosenfeld et al., 2020), strong regularization (Zhou et al., 2022b), and the challenges posed by non-linearity, overparameterization, and optimization (Rosenfeld et al., 2020; Lin et al., 2022a; Chen et al., 2023c). In contrast, our findings show that learning diverse spurious features also help with OOD generalization. This approach, as shown in ensemble-based models, is easily implementable and has shown remarkable empirical success.

To further verify our findings, we contruct MultiColorMNIST, a 10-class variant of CMNIST (Arjovsky et al., 2019) with 32 spurious features, following Definition 1. As shown in Figure 5, each sample in MultiColorMNIST consists of 32 color patches, each serving as a spurious feature. We train two neural networks, denoted as $f_{\theta_1}$ and $f_{\theta_2}$, with the same architecture but different initializations on MultiColorMNIST. The results in Table 1 show that the OSE model ($f_{\theta_1}(\boldsymbol{x}) + f_{\theta_2}(\boldsymbol{x})$) improve OOD performance over individual models ($f_{\theta_1}(\boldsymbol{x})$ and $f_{\theta_2}(\boldsymbol{x})$). In Appendix C.3, (1) we show that each individual model learn a subset of spurious features in MultiColorMNIST and OSE utilizes more diverse spurious features (2) we construct SingleColorMNIST with only one spurious feature and show OSE yields little performance gain since both individual models learn the same spurious feature (similar to the results in Rame et al. (2022)).

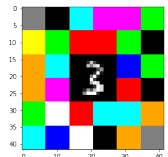

Figure 5: A sample from MultiColorMNIST

| $p$ | 0.70 | 0.75 | 0.80 | 0.85 | 0.90 |
|---|---|---|---|---|---|
| model 1 | 71.05±1.04 | 60.07±1.04 | 48.57±0.92 | 36.93±0.70 | 26.01±0.45 |
| model 2 | 71.77±0.94 | 60.75±0.91 | 49.26±0.83 | 37.74±0.66 | 26.63±0.42 |
| model ensemble | **78.64±0.73** | **67.61±0.80** | **55.25±0.75** | **42.34±0.64** | **29.28±0.40** |

Table 1: OOD performance of (output space) model ensemble on MultiColorMNIST. The spurious correlation is 1 and $1 - p$ in the training and testing set, respectively. A larger $p$ indicates larger distributional shift

## 4 BALANCED AVERAGING (BANG)

Our previous results show that EBM can boost the OOD performance. An implicit requirement is that the scaling of the two models should be roughly the same. If the two models have different scalings, e.g., one model is much more confident than the other, the EBM improvement is weakened.

**Proposition 4** (Imbalanced scaling weakens WSE). *Consider the Example 1, Definition 1-4, Assumption 1-2. Consider an WSE of two imbalanced models, $\bar{f} = (\bar{\boldsymbol{w}}, \bar{\Phi})$ and $\tilde{f}_\lambda = (\lambda\tilde{\boldsymbol{w}}, \lambda\tilde{\Phi})$, where $\lambda \geq 1$. Specifically, $f_{WSE}(\boldsymbol{x}) = 0.25(\bar{\boldsymbol{w}} + \lambda\tilde{\boldsymbol{w}})\boldsymbol{x}(\bar{\Phi} + \lambda\tilde{\Phi})$. We have $\mathcal{A}_{ood}(f_{WSE})|_{\lambda > \sqrt{5}} - \mathcal{A}_{ood}(f_{WSE})|_{\lambda = 1} < -34/729p^3$.*

See Appendix F.3 for proofs and Appendix D.8 for an illustration of the over-confidence characterized by $\lambda$. When $\lambda = 1$, indicating similar confidence levels between $\bar{f}$ and $\tilde{f}_\lambda$, the WSE is balanced. However, when $\lambda > \sqrt{5}$ and $\tilde{f}_\lambda$ is significantly more confident than $\bar{f}$, $f_{\text{WSE}}$ becomes biased towards $\tilde{f}_\lambda$, resulting in a performance drop of over $34/729p^3$. Here we set $\lambda = \sqrt{5}$ for illustration purposes and similar results can be similarly obtained for other $\lambda > 1$. Unfortunately, we find

| Methods | Model Averaging | IN | IN-V2 | IN-R | IN-A | IN-S | ObjectNet | Avg OOD |
|---|---|---|---|---|---|---|---|---|
| Zero-shot (Wortsman et al., 2022) | No | 68.3 | 61.9 | 77.6 | 49.8 | 48.2 | 53.0 | 58.1 |
| Fine-tuning (Wortsman et al., 2022) | No | 81.3 | 70.9 | 65.6 | 36.7 | 46.3 | 49.6 | 53.8 |
| Fine-tuning (LS) | No | 82.0 | 72.3 | 63.3 | 38.3 | 46.5 | 51.1 | 54.3 |
| Fine-tuning (Mixup) | No | 83.0 | 72.7 | 66.4 | 43.7 | 48.8 | 52.4 | 56.8 |
| Fine-tuning (Mixup + LS) | No | 82.9 | 72.7 | 65.8 | 43.6 | 48.5 | 52.2 | 56.6 |
| WiSE-FT (Wortsman et al., 2022) | Yes | 81.7 | 72.8 | 78.7 | 52.2 | 53.9 | 57.3 | 63.0 |
| BANG (LS) | Yes | 82.1 | 73.3 | 78.2 | 55.2 | 53.7 | 58.9 | 63.9 |
| BANG (Mixup) | Yes | 81.5 | 73.0 | 79.5 | 57.9 | 54.5 | 58.7 | 64.7 |
| BANG (Mixup + LS) | Yes | 81.6 | 73.1 | 79.7 | 58.2 | 54.8 | 58.9 | **64.9** |

Table 2: Results of fine-tuning CLIP VIT-B/16 on ImageNet. LS is short for Label Smoothing. The performance of the baseline methods are from the Table 8 of (Wortsman et al., 2022).

WiSE-FT, which is the WSE of the pre-trained model (PM) and fine-tuned model (FM), suffers from the imbalanced confidence issue. Specifically, we compare the PM and FM on their confidence and accuracy. The confidence is defined as the largest probability that a model assigns to a class (details in Appendix E.3). Figure 6 shows that the fine-tuned model is highly over-confident, especially on OOD datasets, e.g., ImageNetA have only 0.37 accuracy while the average confidence is over 0.7. Such overconfidence magnifies the FM's incorrect prediction, leading to deteriorate OOD ensemble performance (details in Appendix E.6).

A direct fix to the issue of over-confidence is to tune the temperature of the softmax of the fine-tuned model (Kumar et al., 2022). However, this method can not be directly applied to WiSE-FT since WiSE-FT ensemble model weights instead of the outputs. Moreover, the temperature scaling tuned on the ID dataset (Kumar et al., 2022) fails to calibrate the fine-tuned model on OOD datasets, where over-confidence is more severe (results of (Kumar et al., 2022) in Appendix E.5-E.6). Therefore, we propose BAlaNced averaGing (BANG), which adopt label smoothing or Mixup during fine-tuning to prevent overconfidence and then average the pre-trained model with such fine-tuned model. (1) Label smoothing replaces the label of the true class (e.g., 1) with a positive value (e.g., 0.8), while distributing the smoothing parameter (e.g., 0.2) evenly among the other classes (Müller et al., 2019). (2) Mixup (Zhang et al., 2017) generates new samples during fine-tuning by linearly mixing pairs of training data and their labels.

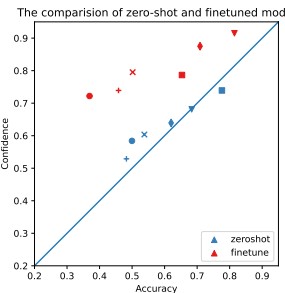

Figure 6: Comparison of confidence and accuracy between zero-shot and finetuned model. In the figure, ▽ refers to IN, ○ for IN-A, □ for IN-R, + for IN-S, ◇ for IN-V2 and × for ObjNet.

We conduct experiments with CLIP ViT-B/16(Radford et al., 2021). We impose Mixup or Label Smoothing during fine-tuning the pre-trained CLIP on ImageNet (IN), and test OOD performance on IN-V2, IN-R, IN-A, IN-S and ObjectNet. Following Wortsman et al. (2022), BANG averages the pre-trained CLIP model and the model finetuned with LS and MixUp (details in Appendix E.4). The results in Table 2 show that BANG effectively improve the performance over WiSE-FT. Specifically, BANG(LS+Mixup), where both LS and MixUp are adopted, achieves 1.9% higher average OOD accuracy than WiSE-FT. Further experimental results in the appendix show that Mixup and Label Smoothing can effectively alleviate the over-confidence of the fine-tuned model on both ID and OOD datasets.

Since Mixup and LS also improve the performance of the fine-tuned model, so a curious reader would wonder whether the improvement of BANG comes from better calibration or just due to the improvement in the fine-tuned model. We conduct further investigation in Appendix E.6 to confirm the contribution of better calibration: (1) Dividing the weight of the vanilla fine-tuned model by multiple scalars significantly enhances the performance of weight averaging, which nearly matches the performance of BANG. (2) BANG can correct substantially more samples that is mis-classified by the fine-tuned model. We also show that BANG's effectiveness can not be explained by other data augmentation methods in Appendix E.5.

ACKNOWLEDGEMENT

We are grateful for the insightful discussion with Yongqiang Chen, Damien Teney, Alexandra Rame, Pang Wei Koh, Mitchell Wortsman, Difan Zou, and Hao Wang. Thank you for your valuable inputs.

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

CONTENTS

## A   SOCIAL IMPACT

We investigate how to enable machine learning models to generalize in OOD scenarios, which makes machine learning models more reliable in real-world applications.

## B   RELATED WORKS

### B.1   A REVIEW ON THE EXISTING METHODS

**Out-of-distribution generalization**   Machine learning models are based on the I.I.D. (independently and identically distribution) assumption. Whereas, the I.I.D. assumption can be easily violated since the model can easily encounter novel testing samples that are from distributions different with the training distribution. This is also known as the out-of-distribution generalization (OOD) problem. Existing works find that the model performance deteriorates dramatically under distributional shift. This is especially the case when the model rely on spurious features that are unstable in a new domain (Geirhos et al., 2020; Arjovsky et al., 2019; Deng et al., 2023). OOD problem has attracted great attention in recent years and there are a rich line of works in this direction, such as Invariant Risk Minimization (IRM) (Arjovsky et al., 2019; Lin et al., 2022b), model averaging (Wortsman et al.; 2022; Ramé et al., 2022; Cha et al., 2021), feature alignment methods (Ganin et al., 2016; Sun & Saenko, 2016; Li et al., 2018) and so on.

Among them, Invariant Risk Minimization (IRM) has gained significant attention from the researchers (Arjovsky et al., 2019) and inspires a great line of works. Recall that there are two kinds of features for OOD generalization: invariant features that can stably predict the labels, and spurious features whose correlation with the labels is unstable. IRM tries to build robust models by extracting only invariant features. IRM has strong theoretical guarantees in linear system and clear connection with causal theory. Nevertheless, IRM methods face challenges in dealing with large scale real-world datasets, as it has been repeatedly observed that IRM cannot outperform ERM on various datasets. Some works have provided explanation for this, e.g., (Rosenfeld et al., 2020) shows that IRM needs a very great number of domains, (Rosenfeld et al., 2020) shows IRM lacks theretical guarantees on non-linear models, Lin et al. (2022b) shows it is difficult to learn invariance without domain partition and (Lin et al., 2022a; Chen et al., 2023c) show the difficulty of optimizing IRM objects on deep neural networks. In contrast, model averaging is exceptionally powerful and achieve SOTA performance in a lot of benchmark with various models and architecture (Cha et al., 2021; Wortsman et al., 2022; Wortsman et al.; Rame et al., 2022; Chu et al., 2022; Arpit et al., 2022).

**Output and weight space ensemble**   The ensemble of multiple models is a powerful idea that often leads to stronger predictive performance (Caruana et al., 2004; Dietterich, 2000; Bauer & Kohavi, 1999; Breiman, 1996). Typically, conventional ensemble methods aggregate the outputs of models, as known as the output space ensemnle. The recent application usually average the parameters of models which is generated from the same pre-training model by finetuning (Wortsman et al.; 2022; Cha et al., 2021), also known as the weight space ensemble. While averaging two models trained from scratch by different initialization often yields poor results (close to random guessing). (Neyshabur et al., 2020) finds that fine-tuning two models from the same pre-trained initialization results in two different models that were connected via a linear path in weight-space, along which the performance remains high. This is also known as linear mode connectivity (Frankle et al., 2020). A notable difference between ensemble in ID and OOD study is that the improvement of ensemble in OOD is much more significant than that in IID. (Wortsman et al., 2022) shows that an ensemble the finetuned model with the pre-trained model improve near 1pp in ImageNet (the ID domain) and over 6-8pp on the variants of ImageNet (the OOD Domain). Actually, model averaging is still among strongest methods for OOD generalization. It still remains mysterious on why averaging methods are so effective for OOD.

**Theory of Out-of-distribution generalization.**   Existing theory mostly focus on the worst case metric on analyzing the OOD performance (Wald et al., 2022; Arjovsky et al., 2019; Rosenfeld et al., 2020; Puli et al., 2021; Zhou et al., 2022b). The worst case metric requires a model to be robust at *any* OOD testing distribution. Typically, a model that only uses invariant features can minimize the worst case metric. However, as we discuss above, invariance learning is hard in

practice and performs ineffectively on real world datasets (Gulrajani & Lopez-Paz, 2020; Rosenfeld et al., 2020; Lin et al., 2022b). The worst case metric based theory can not explain the success of model averaging methods. To be specific, the averaging of two models can use spurious features that learnt by each individual model due to its pessimism as described in Appendix D.4. In contrast, our theoretical results characterize the probability of the model failure due to distributional shift, which can successfully explain the experimental results.

## B.2    ON OUR DIFFERENCE WITH EXISTING WORKS

**Difference with existing works on learning diverse features.**    There have been some works that promote feature diversity to enhance empirical performance OOD generalization by weight average (Chu et al., 2022; Rame et al., 2022; 2023), feature concatenation (Zhang & Bottou, 2023), boosted rich feature learning (Zhang et al., 2022; Chen et al., 2023b; Jain et al., 2022; Teney et al., 2022; Feng et al., 2023), and utilizing model zoo (Dong et al., 2022; Chen et al., 2023a). Teney et al. (2022) train a set of diverse models and select the best one among them for OOD. Feng et al. (2023) proposes to use an ensemble of prompts which contains diverse descriptions of a class to perform classification via CLIP. Jain et al. (2022) train two models with different feature priors and then ensemble the predictions of these models. However, existing explanations either do not distinguish the invariant or spurious features (Chu et al., 2022; Rame et al., 2022; 2023; Zhang & Bottou, 2023; Dong et al., 2022; Chen et al., 2023a), or focus only on learning the potentially missing invariant features (Chen et al., 2023b; Feng et al., 2023). In fact, according to existing invariance learning perspective Arjovsky et al. (2019) arguing that models relying on spurious features are prone to failure in OOD scenarios, these methods that learn diverse features while also incorporating spurious features may not be able to generalize effectively under distributional shift. In contrast, our spurious feature diversification viewpoint provides a explanation by characterizing why and when incorporating more diverse spurious feature diversification can improve OOD performance.

**Difference with the existing theoretical results on ensemble and boosting in IID settings.** There are existing explanations for the effectiveness of model ensemble in the IID setting, which is mainly from the perspective of variance due to over-fitting the label noise in finite samples cases (Dietterich et al., 2002). Specifically, model ensemble can have smaller variance in prediction compared with each single model. Whereas, we consider infinite sample case, where the variance of the model due to fitting label noise is zero. So model ensemble can not bring significant IID improvement in this case. However, the model trained on infinite samples can still fail due to distributional shift (Arjovsky et al., 2019). This is because the model utilizes the spurious features, which are also considered as a kind of bias (Wald et al., 2022). Our results show that model ensemble can reduce the risk of model failure and lead to better expected performance under distributional shift by spurious feature diversification. In other words, model ensemble reduces the probability of the model failure due to the bias. This is a new result in the OOD problem as shown in Proposition 1 and 2. Notably, Allen-Zhu & Li (2020) also considers ensemble in the IID setting, however, their theory can not explain the OOD performance improvement of ESM models on the data in Definition 1, explain the FalseFalseTrue phenomenon, or explain the difference of weight and output space ensemble.

Another related area to this work is boosting. Boosting can benefit by training multiple models, where each model corrects the mistakes made by the previous ones and each model would possibly utilize on different subsets of features. While previous studies on boosting mainly focused on ID scenarios (Schapire, 1990; Freund & Schapire, 1997; Schapire, 2013; 2003), we show that in the context of OOD, the improvement in performance due to using diverse features can be even more significant. This is because different irrelevant features can cause different errors when the distribution changes, and diversifying the features helps reduce the impact of each individual feature (as shown in Figure 1). By utilizing a diverse set of models, boosting allows us to take advantage of a wider range of features and effectively deal with the challenges posed by OOD situations.

**Difference with existing explanations on the OOD performance of ensemble-based methods (EBM).**    There are some previous attempts that try to explain the effectiveness of EBM for OOD. Cha et al. (2021) shows that the loss landscape changes under distributional shift and model averaging can lead to flatter minima. However, as discussed in Rame et al. (2022), the upper bound of Cha et al. (2021) is uncontrolled and their analysis based on flat minima fails to explain many experi-

mental results. Rame et al. (2022) decomposes the OOD loss into the bias, variance and covariance terms. They show that the variance term can benefit from EBM. Different from the results of Rame et al. (2022) that only tackles with the variance term, our results provide a concise characterization on the overall OOD performance. Further, Rame et al. (2022)'s results can not differentiate between the weight and output space ensemble.

# C    Supportive Empirical Results for the Theory

## C.1    FalseFalseTrue Phenomenon

In this subsection, we take a deeper look at WiSE-FT (Wortsman et al., 2022), a popular model averaging method that averages the weights of the pre-trained and fine-tuned model. (Wortsman et al., 2022) obtains the fine-tuned model by fine-tuning the pre-trained CLIP model on ImageNet. They have shown that the averaging of the pre-trained and the fine-tuned model can outweigh both of them on ImageNet (ID dataset) as well as five OOD datasets (ImageNetV2, ImageNetA, ImageNetR, ImageNetSketch and ObjectNet). We denote the pre-trained model as PM, fine-tuned model as FM, and averaged model as AM.

To understand why model averaging is effective, we divide each dataset into eight groups of samples according to whether the PM, FM and AM make correct predictions, respectively. We further use T/F to denote whether a model makes correct predictions, i.e., T for True and F for False. For example, we use PM(T)-FM(F)-AM(T) to denote the group of samples on which the predictions of PM , FM and AM are correct, wrong, and correct, respectively. A simple explanation for the improvement of the averaging of two models is that when one model makes a mistake and the other one is correct, the correct model can rectify the mistakes made by the other model. So we evaluate the performance on the group of data where one model makes a wrong prediction while the other model makes a correct prediction, i.e., the group containing PM(T)-FM(F) and PM(F)-FM(T). We refer to this group of data as TF+FT for short in the following discussion. We also look into another subset TT+FF which contains PM(T)-FM(T) and PM(F)-FM(F).

Given a subset $\mathcal{G}$ of a dataset $\mathcal{D}$, we use CorrectNum$(\mathcal{G}; f)$ to denote the number samples in $\mathcal{G}$ that are correctly predicted by a model $f$, e.g., CorrectNum(TF+FT; PM) stands for the number of samples that are correctly classified by the pre-trained model PM. We propose the metric ImproveContri$(\mathcal{G})$ which estimates how much AM performs better than PM and FM on the group $\mathcal{G}$ and how much the improvement on $\mathcal{G}$ contributes to the overall accuracy improvement on the whole dataset $\mathcal{D}$:

$$\text{ImproveContri}(\mathcal{G}) = \frac{\text{CorrectNum}(\mathcal{G}; \text{AM}) - \max\{\text{CorrectNum}(\mathcal{G}; \text{PM}), \text{CorrectNum}(\mathcal{G}; \text{FM})\}}{|\mathcal{D}|} \quad (2)$$

For Example, suppose $\mathcal{D}$ contains 1,000 samples and its subset $\mathcal{G}$ contains 200 samples. PM, FM and AM correctly predict 120, 118, 130 samples in $\mathcal{G}$, i.e., CorrectNum$(\mathcal{G}; \text{PM}) = 120$, CorrectNum$(\mathcal{G}; \text{FM}) = 118$, CorrectNum$(\mathcal{G}; \text{PM}) = 130$. AM outperform PM and FM by making 10 more correct predictions on $\mathcal{G}$, further these 10 samples contribute to $\frac{10}{1,000} \times 100\% = 1.0\%$ accuracy improvement on the whole dataset. Note that ImproveContri$(\mathcal{D})$ denotes the accuracy improvement of model averaging on the dataset $\mathcal{D}$, which is also denoted as ImproveContri(ALL) in the following discussion. The results of ImproveContri(TT+FF), ImproveContri(TF+FT) and ImproveContri(ALL) are illustrated in Figure 7(a).

We surprisingly find that ImproveContri$(\mathcal{G})$ is significant on TT+FF in all the datasets, which means the averaged model AM can exceed PM and FM on the groups where PM and FM are both right or wrong. Recall that the subset TT+FF contains four groups, PM(T)-FM(T)-AM(T), PM(T)-FM(T)-AM(F), PM(F)-FM(F)-FM(F), and PM(F)-FM(F)-FM(T). We further plot the ratio of the sample size in PM(T)-FM(T)-AM(F) and PM(F)-FM(F)-FM(T) over $|\mathcal{G}|$ in Figure 7(b), respectively. We find that PM(T)-FM(T)-AM(F) is nearly the same (about 0.5% ) in all datasets. The group PM(F)-FM(F)-AM(T) is much larger than PM(T)-FM(T)-AM(F), especially in OOD datasets. It indicates that AM can make correct predictions on many samples where the both PM and FM make wrong predictions when distributional shift occurs! Interestingly, we find ImproveContri$(TF + FT)$ is negative on some datasets, e.g, IN-R and IN-S. In Section 4 and Appendix E.6, we find that this is because the fine-tuned model is highly over-confident and the fine-tuned model dominate WiSE-FT even when it make mistakes.

**Remark**: In Figure 2(Left) of Section 2, we present the results of ImproveContri(TT+FF) to represent the samples where both individual models make incorrect predictions, but the averaged model makes correct predictions. ImproveContri(TT+FF) is calculated as the group ratio of PM(F)-FM(F)-AM(T) subtracted by PM(T)-FM(T)-AM(F). We use ImproveContri(TT+FF) instead of PM(F)-FM(F)-AM(T) because we believe that there is a certain proportion of samples in PM(F)-FM(F)-AM(T) where the averaged model corrects mistakes due to the randomness introduced by the non-

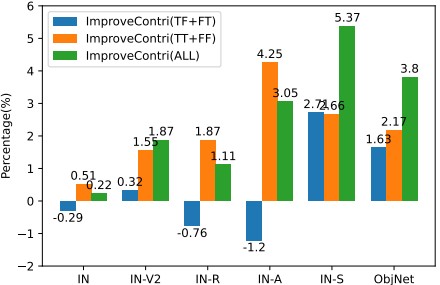 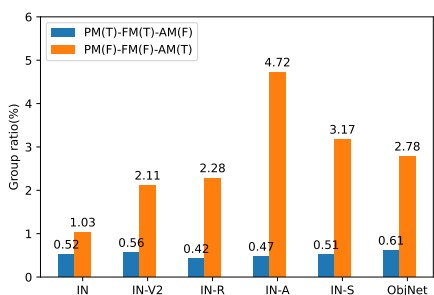

Figure 7: A closer look at WiSE-FT, which averages the pre-trained CLIP and the model obtained by fine-tuning CLIP on ImageNet. Here ImageNet is regarded as ID domain and the other 5 ImageNet variants are OOD domains, i.e., IN-V2 (ImageNetV2), IN-R(ImageNetR), IN-A(ImageNetA), IN-S (ImageNetSketch), and ObjNet (ObjectNet). (Left) ImproveContri($\mathcal{G}$) is defined in Eqn. equation 2, which estimates how the AM (averaged model) performs better than the PM (pre-trained) and FM (fine-tuned model) on the group $\mathcal{G}$ and how much the improvement on $\mathcal{G}$ contributes to the overall accuracy improvement on the whole dataset $\mathcal{D}$. (Right) The ratio of sample size in PM(T)-FM(T)-AM(F) and PM(F)-FM(F)-AM(T) over the same size of the whole dataset. Here PM(T)-FM(T)-AM(F) denotes the group where the AM make wrong predictions and the PM and FM models make correct predictions; PM(F)-FM(F)-AM(T) denotes the group where AM make correct predictions while both PM and FM make wrong predictions. Putting these two figures together, we can see the AM can correct many samples on which PM and FM make wrong predictions in OOD.

linearity of deep neural networks (DNNs) during weight averaging. To approximate such randomness, we use the size of PM(T)-FM(T)-AM(F). This adjustment helps account for a more accurate approximation of the sample ratios where the averaged model corrects the samples due to its utilization of more diverse spurious features.

## C.2 Deep neural networks learn different features

In Section 2, we have shown that the pre-trained and fine-tuned uses different features and the averaged model can utilize more diverse features. Actually, (Allen-Zhu & Li, 2020) provides empirical evidence (e.g., Figure 3 and 4 in (Allen-Zhu & Li, 2020)) supporting the use of diverse features by different deep neural networks with same architecture, even when trained on the same datasets (with different initialization). We add their empirical observations in Figure 8 for easy of reference.

## C.3 Experiments on MultiColorMNIST

**MultiColorMNIST**. We extend the CMNIST (Arjovsky et al., 2019) to MultiColorMNIST, which is constructed following Definition 1. MultiColorMNIST contains 10 classes with 32 spurious features. Each image has $42 \times 42 \times 3$ pixels. There are 32 patches in each image and each patch can take one of 10 colors. Figure **??** illustrates two samples from MultiColorMNIST. Specifically, the label of the sample is generated from the shape of the digit and each color patch is perfected correlated with the label. Let $C_i$ denote $i$th color patch for $i = 1, 2, ..., 32$. Each $C_i$ takes one of the color which is perfectly correlated with $\boldsymbol{y}$. For example, the 1st patch, i.e., $C_1$, always takes 'white' on samples with label 5; the 2nd patch, i.e., $C_2$, always takes 'yellow' on samples with label 5. Each $C_i$ is independently generated from the label $\boldsymbol{y}$ and we have $C_i \perp C_j | \boldsymbol{y}$ for $i \neq j$. See Figure 10 for detailed illustration of the data generation process which follows the theoretical Definition 1. In the OOD testing distribution, the spurious correlation can fail with probability $p$. For example, samples with label 5 can randomly pick any color with probability $p$ in OOD . The data generation process is analogous to the theoretical setting in Definition 1, where each patch is a spurious feature and each color is an attribute that the spurious feature can take.

**SingleColorMNIST**. We also introduce SingleColorMNIST for better comparision. SingleColorMNIST has 10 classes and each image has $42 \times 42 \times 3$ pixels, which is the same with MultiCol-

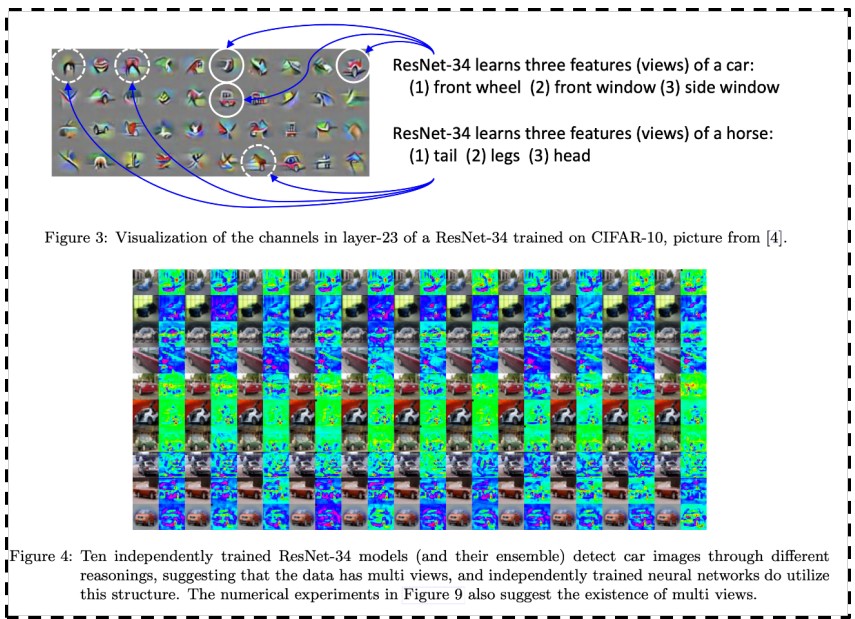

Figure 3: Visualization of the channels in layer-23 of a ResNet-34 trained on CIFAR-10, picture from [4].

Figure 4: Ten independently trained ResNet-34 models (and their ensemble) detect car images through different reasonings, suggesting that the data has multi views, and independently trained neural networks do utilize this structure. The numerical experiments in Figure 9 also suggest the existence of multi views.

Figure 8: Figures taken from (Allen-Zhu & Li, 2020) which show that different DNN (with the same architecture) learns different features even trained on the same dataset.

orMNIST. However, SingleColorMNIST only contains 1 spurious features. In other words, the 32 patches in each image are the same. The spurious correlation is defined similarly with MultiColorMNIST. Figure 9 illustrates two samples from SingleColorMNIST.

**Experimental Details**. We use the following configuration for both SingleColorMNIST and MultiColorMNIST. We use an 2 layer MLP with 64 hidden units to perform classification. We adopt Adam with learning rate $10^{-3}$ and batch size 100. We train for 5000 steps and report the performance at the last step. We train two individual models $\bar{f}$ and $\tilde{f}$ with different random initialization on MultiColorMNIST. We also evaluate the ensemble of the two models, i.e., $f_{\text{ose}}(\boldsymbol{x}) = \bar{f}(\boldsymbol{x}) + \tilde{f}(\boldsymbol{x})$. Each experiment is repeated for $n = 20$ random seeds.

**Results**. We vary $p$ in MultiColorMNIST and compare the performance of the ensemble model with each individual model. $p$ is the probability that spurious correlation no-longer holds in testing environment. A larger $p$ indicates larger distributional shift. The results of MultiColorMNIST are summarized in Table 3. We can see that model ensemble consistently improve the OOD performance. Figure 11 visualizes how much each model relies on each patch. Specifically, Figure 11 shows how much the model changes its prediction when we replace a patch with black color. We can see each individual models uses different feature sets and model ensemble uses more diverse features. Table 4 shows the results of SingleColorMNIST. We can see that model ensemble can not improve the OOD performance in SingleColorMNIST (since there is only one spurious feature in SingleColorMNIST and model ensemble can not utilize more diverse spurious features).

Comparing Table 3 and Table 4, we can see that the performance of individual model in MultiColorMNIST is higher than that in SingleColorMNIST when the $p$ is the same. This is because the individual model already learns multiple spurious features (even though it is only a small subset of the whole feature set as shown in Figure 11). This is also consistent with our theoretical results that diverse spurious features leads to better OOD performance.

**Remark**. Recall weight space ensemble (WSE) needs to be conducted between the pre-trained and fine-tuned models or different fine-tuned models starting from the same pre-trained model (Wortsman et al., 2022; Frankle et al., 2020). Since we have suitable pre-trained model for the synthetic dataset, we leave the investigation of WSE on MultiColorMNIST to future work.

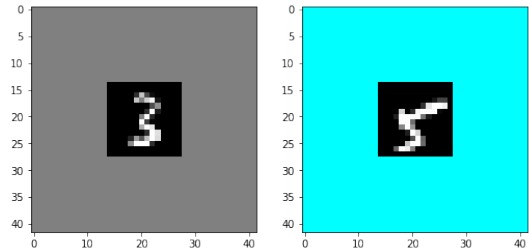

Figure 9: Two samples from SingleColorMNIST. SingleColorMNIST has 10 classes and each sample contains 1 spurious feature.

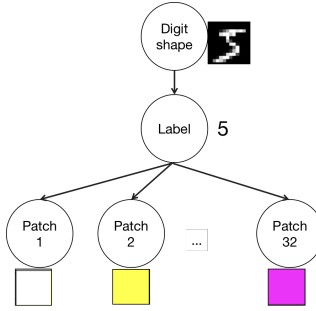

Figure 10: The data generation process of MultiColorMNIST (follows Definition 1)

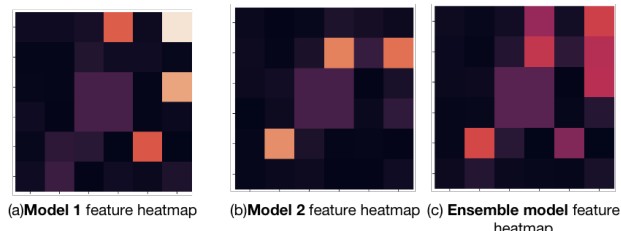

(a)**Model 1** feature heatmap  (b)**Model 2** feature heatmap  (c) **Ensemble model** feature heatmap

Figure 11: Visualization of the features uses by each model and model ensemble.

| p | model_1 | model_2 | model_ensemble |
|------|----------------|----------------|-------------------|
| 0.10 | 100.00±0.00 | 100.00±0.00 | 100.00±0.00 |
| 0.20 | 99.99±0.00 | 99.99±0.00 | 100.00±0.00 |
| 0.30 | 99.91±0.01 | 99.89±0.04 | 99.99±0.01 |
| 0.40 | 99.16±0.11 | 99.19±0.13 | **99.75±0.03** |
| 0.50 | 95.84±0.35 | 96.06±0.35 | **98.13±0.14** |
| 0.60 | 87.15±0.74 | 87.56±0.69 | **92.31±0.41** |
| 0.70 | 71.05±1.04 | 71.77±0.94 | **78.64±0.73** |
| 0.75 | 60.07±1.04 | 60.75±0.91 | **67.61±0.80** |
| 0.80 | 48.57±0.92 | 49.26±0.83 | **55.25±0.75** |
| 0.85 | 36.93±0.70 | 37.74±0.66 | **42.34±0.64** |
| 0.90 | 26.01±0.45 | 26.63±0.42 | **29.28±0.40** |

Table 3: Results on MultiColorMNIST

| p | model_1 | model_2 | model_ensemble |
|---|---|---|---|
| 0.1 | 91.04±0.03 | 91.07±0.05 | 91.04±0.04 |
| 0.2 | 82.34±0.02 | 82.39±0.07 | 82.34±0.04 |
| 0.3 | 72.93±0.08 | 72.99±0.10 | 72.93±0.09 |
| 0.4 | 64.08±0.09 | 64.21±0.19 | 64.10±0.11 |
| 0.5 | 54.89±0.12 | 54.99±0.18 | 54.88±0.14 |
| 0.6 | 45.91±0.16 | 46.09±0.31 | 45.92±0.19 |
| 0.7 | 37.39±0.15 | 37.55±0.27 | 37.39±0.17 |
| 0.8 | 27.86±0.19 | 28.06±0.32 | 27.87±0.22 |
| 0.9 | 19.28±0.18 | 19.50±0.34 | 19.29±0.20 |

Table 4: Results on SingleColorMNIST

### C.3.1 INCREASING THE NUMBER OF ENSEMBLE

In Table 1 and 3, we show that the ensemble of two models improves significantly over each individual model on MultiColorMNIST. In this part, we are going to show that if increasing the number of models in the ensemble can even increases more significantly.

Specifically, in Table 5, we show the results of different model number in the ensemble. When the ensemble number is 1, it means that we consider a single model (in other words, not performing model ensemble). If ensemble number is 16, it indicates that we independently train 16 models with different initialization and use the ensemble of these 16 models to make predictions. We can see that increasing the ensemble number can signficantly boost the OOD performance. For example, when $p = 0.8$, the OOD performance of single model (ensemble number equals 1) is 49.33%. The ensemble of two models achieves 55.92% OOD accuracy. The ensemble of 16 models can increases the OOD accuracy to 64.85%! This also gives us a hint on the effectiveness of model soup, which averages multiple checkpoints trained with different hyper-parameters.

| $p$ 
 Ensemble Number | 0.70 | 0.75 | 0.80 | 0.85 | 0.90 |
|---|---|---|---|---|---|
| 1 | 71.66±2.06 | 60.68±2.23 | 49.33±2.02 | 37.74±1.58 | 26.74±1.05 |
| 2 | 78.88±1.24 | 68.34±0.89 | 55.96±0.77 | 42.91±0.64 | 29.89±0.63 |
| 4 | 84.39±1.33 | 74.00±1.26 | 62.04±1.32 | 47.92±1.17 | 32.74±0.75 |
| 8 | 85.64±1.22 | 75.73±1.62 | 63.52±1.61 | 49.15±1.23 | 33.67±0.93 |
| 16 | 86.76±0.55 | 77.31±0.87 | 64.85±1.09 | 50.63±0.69 | 34.47±0.40 |

Table 5: Experiments on MultiColorMNIST. A larger $p$ indicates larger distributional shift.

On the other hand, if the dataset only contains a single spurious feature, e.g., the SingleColorMNIST, we find that increasing ensemble number does not help the OOD performance. These results are included in Table 6.

| $p$ 
 Ensemble Number | 0.70 | 0.75 | 0.80 | 0.85 | 0.90 |
|---|---|---|---|---|---|
| 1 | 37.40 ± 0.11 | 32.64 ± 0.11 | 27.90 ± 0.14 | 23.32 ± 0.14 | 19.32 ± 0.14 |
| 2 | 37.32 ± 0.03 | 32.54 ± 0.05 | 27.78 ± 0.04 | 23.20 ± 0.04 | 19.18 ± 0.05 |
| 4 | 37.35 ± 0.09 | 32.57 ± 0.12 | 27.81 ± 0.13 | 23.22 ± 0.11 | 19.23 ± 0.12 |
| 8 | 37.30 ± 0.01 | 32.50 ± 0.01 | 27.74 ± 0.02 | 23.16 ± 0.02 | 19.16 ± 0.00 |
| 16 | 37.35 ± 0.08 | 32.56 ± 0.09 | 27.82 ± 0.12 | 23.22 ± 0.10 | 19.24 ± 0.11 |

Table 6: Experiments on SingleColorMNIST. A larger $p$ indicates larger distributional shift.

### C.4 Simulation

In this section, we take some simulations to investigate the performance of theoretical forecasting results of OOD accuracy. Following the data generation process in Definition 1, here we consider four examples:

1. example 1-1: $\bar{n}_v = 2, \bar{n}_s = 3$ in model 1; $\tilde{n}_v = 2, \tilde{n}_s = 3$ in model 2; overlapped feature number $n_{vo} = n_{so} = 0$; noise variance $\sigma = 0.01$; distribution shift probability $p = 0.9$.

2. example 1-2: $\bar{n}_v = 2, \bar{n}_s = 3$ in model 1; $\tilde{n}_v = 2, \tilde{n}_s = 3$ in model 2; overlapped feature number $n_{vo} = n_{so} = 1$; noise variance $\sigma = 0.01$; distribution shift probability $p = 0.9$.

3. example 2-1: $\bar{n}_v = 5, \bar{n}_s = 20$ in model 1; $\tilde{n}_v = 4, \tilde{n}_s = 20$ in model 2; overlapped feature number $n_{vo} = n_{so} = 0$; noise variance $\sigma = 0.01$; distribution shift probability $p = 0.9$.

4. example 2-2: $\bar{n}_v = 5, \bar{n}_s = 20$ in model 1; $\tilde{n}_v = 5, \tilde{n}_s = 20$ in model 2; overlapped feature number $n_{vo} = 4, n_{so} = 1$; noise variance $\sigma = 0.01$; distribution shift probability $p = 0.9$.

In each example, we take 1000 simulations to report the mean OOD accuracy in Table 7. To be precise, the training data size is 20000 and the test data size is 10000 in each simulation. Then com-

|  |  | Model 1 | Model 2 | Model Average | Model Ensemble |
|---|---|---|---|---|---|
| Example 1-1 | Simulation Results | 0.866 | 0.866 | 0.974 | 0.974 |
|  | Theoretical Results | 0.865 | 0.865 | 0.973 | 0.973 |
| Example 1-2 | Simulation Results | 0.866 | 0.861 | 0.943 | 0.940 |
|  | Theoretical Results | 0.865 | 0.865 | 0.948 | 0.946 |
| Example 2-1 | Simulation Results | 0.940 | 0.894 | 0.978 | 0.978 |
|  | Theoretical Results | 0.941 | 0.910 | 0.980 | 0.980 |
| Example 2-2 | Simulation Results | 0.943 | 0.939 | 0.999 | 0.989 |
|  | Theoretical Results | 0.943 | 0.943 | 0.992 | 0.983 |

Table 7: Simulation for the OOD accuracy in different models

paring the results of theoretical results and simulation results, it is safely to say that our theoretical analysis, as well as proper approximations, could take an effective estimation for OOD accuracy.

# D DISCUSSIONS, ILLUSTRATIONS, AND SUPPORTIVE RESULTS FOR THE THEORETICAL PARTS.

## D.1 DISCUSSION ON THE THEORETICAL MODELS

Our theoretical models in Section 3 is designed to mimic the modern deep learning architectures such as Vision Transforms (ViT) (Dosovitskiy et al., 2020). Figure 12 provides a comparison between our theoretical models and Vision Transformers.

Similar to ViT, we process images as patches, where each patch corresponds to a specific feature denoted as Patch$_i$. Each Patch$_i$ is represented by high-dimensional vectors $x_i \in \mathbb{R}^d$ in the embedding space. Consequently, the whole feature is obtained by concatenating the embeddings of each patch, resulting in $\boldsymbol{x} = [x_1, x_2, ...]$. Assuming a total of $d_t$ features ($d_t = d_v + d_s$ in Section 3), we have $\boldsymbol{x} \in \mathbb{R}^{d \times d_t}$. Notably, (Allen-Zhu & Li, 2020) also uses a similar theoretical data model that concatenates the patches to analyze the convolutional neural networks, e.g., Figure 5 in (Allen-Zhu & Li, 2020).

To simplify the model, we utilize a two-layer structure consisting of a binary feature mask $\Phi$ as the feature encoder and a linear classifier $\boldsymbol{w}$, analogous to ViT which uses the transformer feature encoder with an MLP classifier. This two-layer simplification approach has been widely employed in OOD literature (Arjovsky et al., 2019; Rosenfeld et al., 2020; Zhou et al., 2022b; Peters et al., 2016; Lin et al., 2022b). The difference between our theoretical model and ViT is that ViT process the features sequentially while we select the feature at once.

The binary feature mask $\Phi$ is represented as $\{0, 1\}^{d_t}$. For instance, if we have three features, i.e., $\boldsymbol{x} = [x_1, x_2, x_3]$, and $\Phi = [1, 1, 0]$, the learned feature would be $\boldsymbol{x}^\top \Phi = x_1 + x_2$. Considering a 3-class classification task, the linear classifier $\boldsymbol{w} \in \mathbb{R}^{d \times 3}$ takes the learned feature $\boldsymbol{x}^\top \Phi$ as input and produces a 3-dimensional vector whose elements represent the logits of the three classes. The classifier $\boldsymbol{w}$ is optimized to minimize the in-distribution (ID) loss based on the learned feature. Therefore, we have:

$$\boldsymbol{w} = \arg \min_{\boldsymbol{v} \in \mathbb{R}^{d \times 3}} \mathcal{R}_{id}(\boldsymbol{v}, \Phi),$$

where $\mathcal{R}_{id}(\boldsymbol{v}, \Phi)$ represents the loss of $(\boldsymbol{v}, \Phi)$ in the ID distribution.

## D.2 COMPARISON ON OUR MODEL WITH A 2-LAYER DNN

In this paper, we consider the model $\boldsymbol{w}^\top \boldsymbol{x} \Phi$, where $\boldsymbol{w} \in \mathbb{R}^{d \times K}$ and $\Phi \in \{0, 1\}^{d_v + d_s}$ are paramters. Here the input $\boldsymbol{x} \in \mathbb{R}^{d \times (d_v + d_s)}$ and see App D.1 for detailed discussion. We then compare our model with a general 2-layer DNN to see why it can capture the difference between weight space ensemble (WSE) and output space ensemble (OSE) in DNN.

Consider a general 2-layer DNN parameterized by $(W_a \in \mathbb{R}^{d_1 \times d_2}, W_b \in \mathbb{R}^{d_2 \times K})$ with ReLU activation $\delta(\cdot)$ and output $f_{dnn}(X) = W_b^\top \delta(W_a^\top X)$ for $X \in \mathbb{R}^{d_1}$. Here we use uppercase $X$ to avoid confusion with our previous $\boldsymbol{x}$ since they have slightly different dimensions (App D.1). Since WSE is conducted on the models that is close to a pre-trained model (Wortsman et al.), e.g., $(W_{a0}, W_{b0})$, so we consider $f_{dnn}(X) = (W_{b0} + \Delta W_b)^\top \delta((W_{a0} + \Delta W_a)^\top X)$ where $\Delta W_a$ and $\Delta W_b$ is small and trainable. By Taylor expansion, we have

$$f_{dnn}(X) = \underbrace{W_{b0}^\top \delta(W_{a0}^\top X)}_{\text{(a)Fixed Term}} + \underbrace{\Delta W_{b0}^\top \delta(W_{a0}^\top X) + W_{b0} \delta'(W_{a0}^\top X)(\Delta W_a^\top X)}_{\text{(b)Linear Term}} + \underbrace{\Delta W_b \delta'(W_{a0}^\top X)(\Delta W_a^\top X)}_{\text{(c)Bilinear Term}} + \xi$$

Where $\delta'(Y)$ is $\frac{\partial \delta(Y)}{\partial Y}$. Further, we incorporate the fact that the second order derivative $\frac{\partial^2 \delta(Y)}{\partial^2 Y}$ is zero almost everywhere for ReLU activation function (except at $Y = 0$). Then $\xi$ is the error term induced by the non-linearity of ReLU activation function (while $W_{a0}^T X$ has some zero elements). To be precise, as here we just focus on fine-tuning regime and $W_{a0}^T X$ is not sparse in general situations, it is safely to say that $\xi$ is small. WSE and OSE are exactly the same for the (a) fixed term and (b) linear term. We will show that WSE and OSE differs on the (c)bilinear term, which is captured by our model in Definition 3-4.

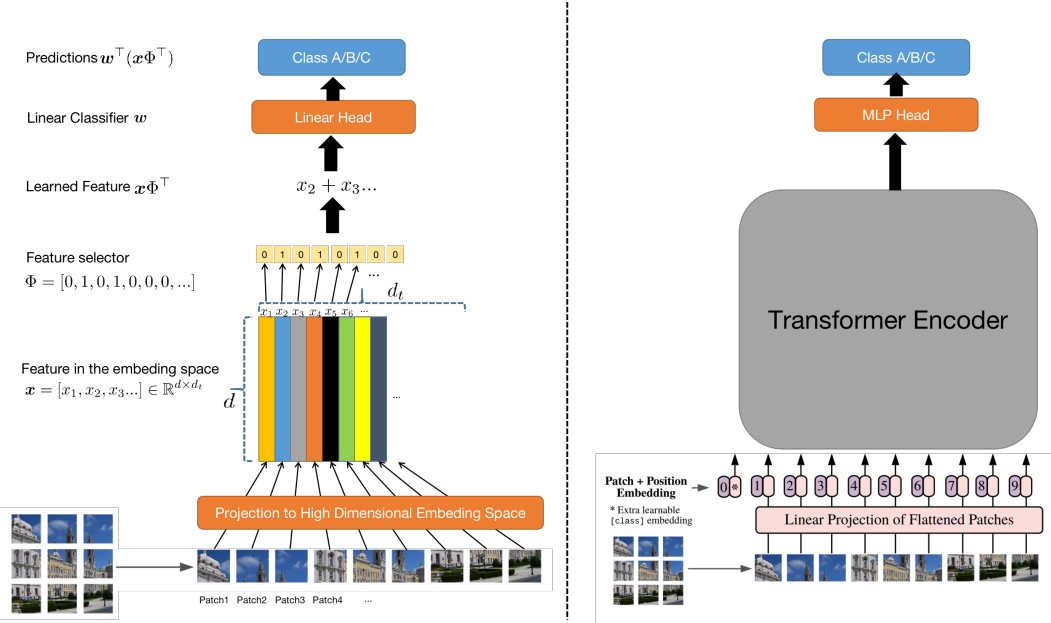

Figure 12: Comparison of our theoretical models in Section 3 with Vision Transformers (Dosovit-skiy et al., 2020). Some parts of the figures are adopted from (Dosovitskiy et al., 2020).

Consider two models, $\bar{f}_{dnn}$ and $\tilde{f}_{dnn}$, both close to the pre-trained model. Specifically,

$$\bar{f}_{dnn}(X) = (W_{b0} + \Delta\bar{W}_b)^\top \delta((W_{a0} + \Delta\bar{W}_a)^\top X);$$
$$\tilde{f}_{dnn}(X) = (W_{b0} + \Delta\tilde{W}_b)^\top \delta((W_{a0} + \Delta\tilde{W}_a)^\top X);$$

Then the output space ensemble of $\bar{f}_{dnn}(X)$ and $\tilde{f}_{dnn}(X)$ is

$$f_{dnn,ose} = 0.5 \left( (W_{b0} + \Delta\bar{W}_b)^\top \delta((W_{a0} + \Delta\bar{W}_a)^\top X) + (W_{b0} + \Delta\tilde{W}_b)^\top \delta((W_{a0} + \Delta\tilde{W}_a)^\top X) \right)$$

$$= \underbrace{W_{b0}^\top \delta(W_{a0}^\top X)}_{\text{(a)Fixed Term}} + \underbrace{0.5(\Delta\bar{W}_{b0} + \Delta\tilde{W}_{b0})^\top \delta(W_{a0}^\top X) + W_{b0}\delta'(W_{a0}^\top X)(0.5(\Delta\bar{W}_a + \Delta\tilde{W}_a)^\top X)}_{\text{(b)Linear Term}}$$

$$+ \underbrace{0.5 \left( \Delta\bar{W}_b\delta'(W_{a0}^\top X)(\Delta\bar{W}_a^\top X) + \Delta\tilde{W}_b\delta'(W_{a0}^\top X)(\Delta\tilde{W}_a^\top X) \right)}_{\text{(c)Bilinear Term}}$$

$$f_{dnn,wse} = (W_{b0} + 0.5(\Delta\bar{W}_b + \Delta\tilde{W}_b))^\top \delta((W_{a0} + 0.5(\Delta\bar{W}_a + \Delta\tilde{W}_a))^\top X)$$

$$= \underbrace{W_{b0}^\top \delta(W_{a0}^\top X)}_{\text{(a)Fixed Term}} + \underbrace{0.5(\Delta\bar{W}_{b0} + \Delta\tilde{W}_{b0})^\top \delta(W_{a0}^\top X) + W_{b0}\delta'(W_{a0}^\top X)(0.5(\Delta\bar{W}_a + \Delta\tilde{W}_a)^\top X)}_{\text{(b)Linear Term}} +$$

$$+ \underbrace{0.25(\Delta\bar{W}_b + \Delta\tilde{W}_b)\delta'(W_{a0}^\top X)((\Delta\bar{W}_a + \Delta\tilde{W}_a)^\top X)}_{\text{(c)Bilinear Term}}$$

Comparing $f_{dnn,ose}$ with $f_{dnn,wse}$, we can see that the difference of them lies in the bilinear term:

$$f_{dnn,wse} - f_{dnn,ose} = \left( 0.25(\Delta\bar{W}_b + \Delta\tilde{W}_b)\delta'(W_{a0}^\top X)((\Delta\bar{W}_a + \Delta\tilde{W}_a)^\top X) \right)$$
$$- 0.5 \left( \Delta\bar{W}_b\delta'(W_{a0}^\top X)(\Delta\bar{W}_a^\top X) + \Delta\tilde{W}_b\delta'(W_{a0}^\top X)(\Delta\tilde{W}_a^\top X) \right) \quad (3)$$

We can see the bilinear term difference has a clear analogy with our models in Definition 3-4. Specifically, according to our Definition of OSE and WSE in Definition 3-4, we have

$$f_{\text{wse}} - f_{\text{ose}} = 0.25(\bar{w} + \tilde{w})^\top x(\bar{\Phi} + \tilde{\Phi}) - 0.5(\bar{w}^\top x\bar{\Phi} + \tilde{w}^\top x\tilde{\Phi}). \quad (4)$$

Comparing equation 3 and equation 4, we can see that $w$ is analogous to $\Delta W_b$ and $\Phi$ is analogous to $\Delta W_a$. equation 3 and equation 4 differ by a scaling $\delta'(W_{a0}^\top X)$, which is a fixed matrix independent of the trainable parameter $(\Delta W_a, \Delta W_b)$.

### D.3 Illustration of the transformation matrix Q

Consider the 3-class classification problem. In the ID distribution, we have,

$$\boldsymbol{Q}_{s,i} = [\boldsymbol{e}_1, \boldsymbol{e}_2, \boldsymbol{e}_3] = \boldsymbol{I}_3 = \begin{bmatrix} 1,0,0 \\ 0,1,0 \\ 0,0,0 \end{bmatrix},$$

This indicates that each spurious feature is perfectly correlated with the invariant feature, as illustrated in Figure 13 (left). For instance, $\boldsymbol{Q}_{s,j} = \boldsymbol{I}_3$ implies that the background of the dog, crow, and camel are floor, grass, and sand, respectively.

In the OOD distribution, $\boldsymbol{Q}_{s,j}$ is no longer equal to $\boldsymbol{I}$, indicating that the correlation between animals and the background may fail with a certain probability. Figure 13 (right) illustrates $\boldsymbol{Q}_{s,j}(1)$, which represents the first column of $\boldsymbol{Q}_{s,j}$. $\boldsymbol{Q}_{s,j}(1)$ can take the value $\boldsymbol{e}_2$ with a probability of $p/3$, indicating that the background of the dog is grass in this case. Similarly, $\boldsymbol{Q}_{s,j}(1)$ can take the value $\boldsymbol{e}_3$ with a probability of $p/3$, indicating that the background of the dog is sand with a probability of $p/3$.

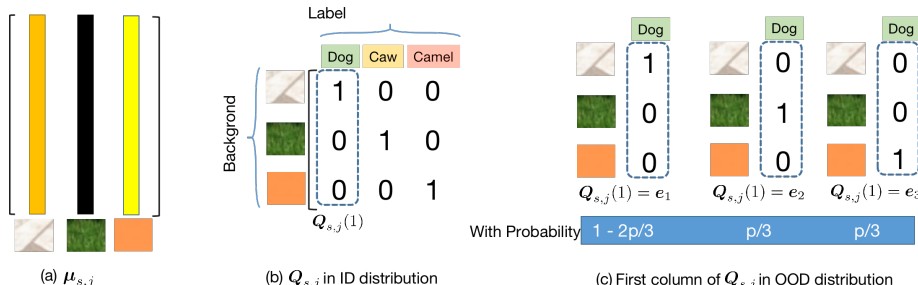

Figure 13: Illustrations of the matrix $\boldsymbol{Q}_{s,j}$.

### D.4 On the pessimism of worst-case theoretical analysis for OOD

### D.5 ID performance

Recall that in Section 3 the OOD accuracy is defined by

$$\mathcal{A}_{\text{ood}}(f) = \mathbb{E}_{\boldsymbol{Q}_s} \left[ \mathbb{E}_{\boldsymbol{x}, \boldsymbol{y}} [\mathbb{I}(\boldsymbol{e}_{\hat{k}} = \boldsymbol{y}) | \boldsymbol{Q}_s] \right].$$

The ID accuracy $\mathcal{A}_{\text{id}}(f)$ is defined similarly by fixing $[\boldsymbol{Q}_{s,1}, \dots, \boldsymbol{Q}_{s,d_s}] = [\boldsymbol{I}, \dots, \boldsymbol{I}]$. According to Lemma 3, we know that the ID accuracy of all models involved in Definition 2, Example 1-2 are larger than $1 - \epsilon$.

### D.6 Intuition of OOD Performance Improvement of OSE

We use Example 1 to show the main intuition of the output space ensemble (OSE). In Example 1, two individual models learn non-overlapped feature, so model ensemble and averaging are the same. According to the proof in Appendix F.1, consider the samples from the first class, i.e., $\boldsymbol{y} = \boldsymbol{e}_1$, the

predicted logit of the each class is

$$\boldsymbol{w}(1)^\top \boldsymbol{x}\bar{\Phi}|_{\boldsymbol{y}=\boldsymbol{e}_1} = \sum_{i=1}^{2} \boldsymbol{\mu}_{v,i}(1)^\top \left(\boldsymbol{\mu}_{v,i}\boldsymbol{Q}_{v,i}(1)\right) + \sum_{j=1}^{3} \boldsymbol{\mu}_{s,j}(1)^\top \left(\boldsymbol{\mu}_{s,j}\boldsymbol{Q}_{s,j}(1)\right),$$

$$\boldsymbol{w}(2)^\top \boldsymbol{x}\bar{\Phi}|_{\boldsymbol{y}=\boldsymbol{e}_1} = \sum_{i=1}^{2} \boldsymbol{\mu}_{v,i}(2)^\top \left(\boldsymbol{\mu}_{v,i}\boldsymbol{Q}_{v,i}(1)\right) + \sum_{j=1}^{3} \boldsymbol{\mu}_{s,j}(2)^\top \left(\boldsymbol{\mu}_{s,j}\boldsymbol{Q}_{s,j}(1)\right),$$

$$\boldsymbol{w}(3)^\top \boldsymbol{x}\bar{\Phi}|_{\boldsymbol{y}=\boldsymbol{e}_1} = \sum_{i=1}^{2} \boldsymbol{\mu}_{v,i}(3)^\top \left(\boldsymbol{\mu}_{v,i}\boldsymbol{Q}_{v,i}(1)\right) + \sum_{j=1}^{3} \boldsymbol{\mu}_{s,j}(3)^\top \left(\boldsymbol{\mu}_{s,j}\boldsymbol{Q}_{s,j}(1)\right),$$

where we omit the noise term whose impact on the accuracy is less than $\epsilon$ according to Lemma 3. Further, let $\boldsymbol{\mu}$ denote any $\boldsymbol{\mu}_{v,i}$ and $\boldsymbol{\mu}_{s,j}$ and $\boldsymbol{Q}$ denote its corresponding transformation matrix. For example, $\boldsymbol{\mu} = \boldsymbol{\mu}_{s,j}$ and $\boldsymbol{Q} = \boldsymbol{Q}_{s,j}$. Suppose $\boldsymbol{Q}(1) = \boldsymbol{e}_{k_2}$, we have

$$\boldsymbol{\mu}(k_1)^\top \left(\boldsymbol{\mu}\boldsymbol{Q}(1)\right) = \boldsymbol{\mu}(k_1)^\top \left(\boldsymbol{\mu}\boldsymbol{e}_{k_2}\right) = \boldsymbol{\mu}(k_1)^\top \boldsymbol{\mu}(k_2) = \begin{cases} 1, & \text{if } k_1 = k_2, \\ 0, & \text{otherwise.} \end{cases}$$

For the invariant features, $\boldsymbol{Q}_{v,i}(1) = \boldsymbol{e}_1$ always hold and for spurious features, $\boldsymbol{Q}_{s,j}(1)$ takes $\boldsymbol{e}_1$ with probability $1 - \frac{2p}{3}$, and takes $\boldsymbol{e}_2$ or $\boldsymbol{e}_3$ with $\frac{p}{3}$, respectively. So the predicted logit of the each class is simply

$$\bar{\boldsymbol{w}}(1)^\top \boldsymbol{x}\bar{\Phi}|_{\boldsymbol{y}=\boldsymbol{e}_1} = 2 + \sum_{j=1}^{3} \mathbb{I}(\boldsymbol{Q}_{s,j}(1) = \boldsymbol{e}_1),$$

$$\bar{\boldsymbol{w}}(2)^\top \boldsymbol{x}\bar{\Phi}|_{\boldsymbol{y}=\boldsymbol{e}_1} = 0 + \sum_{j=1}^{3} \mathbb{I}(\boldsymbol{Q}_{s,j}(1) = \boldsymbol{e}_2),$$

$$\bar{\boldsymbol{w}}(3)^\top \boldsymbol{x}\bar{\Phi}|_{\boldsymbol{y}=\boldsymbol{e}_1} = 0 + \sum_{j=1}^{3} \mathbb{I}(\boldsymbol{Q}_{s,j}(1) = \boldsymbol{e}_3),$$

Let us consider the probability of $\bar{\boldsymbol{w}}(2)^\top \boldsymbol{x}\bar{\Phi}|_{\boldsymbol{y}=\boldsymbol{e}_1} > \bar{\boldsymbol{w}}(1)^\top \boldsymbol{x}\bar{\Phi}|_{\boldsymbol{y}=\boldsymbol{e}_1}$, i.e., the model $(\bar{\Phi}, \bar{w})$ mistakenly predicts the second the class $\boldsymbol{e}_2$ even if the true class is $\boldsymbol{e}_1$. This will happen when $\{\mathbb{I}(\boldsymbol{Q}_{s,j}(1) = \boldsymbol{e}_2)\}_{j=1,2,3}$ holds simultaneously, whose probability would be $\frac{p^3}{27}$. Intuitively, 3 spurious features takes the value in OOD that is correlated with the second class $\boldsymbol{e}_2$, overwhelming the two invariant features correlated with $\boldsymbol{e}_1$.

As for the averaged model, we have

$$\bar{\boldsymbol{w}}(1)^\top \boldsymbol{x}\bar{\Phi}|_{\boldsymbol{y}=\boldsymbol{e}_1} = 4 + \sum_{j=1}^{6} \mathbb{I}(\boldsymbol{Q}_{s,j}(1) = \boldsymbol{e}_1),$$

$$\bar{\boldsymbol{w}}(2)^\top \boldsymbol{x}\bar{\Phi}|_{\boldsymbol{y}=\boldsymbol{e}_1} = 0 + \sum_{j=1}^{6} \mathbb{I}(\boldsymbol{Q}_{s,j}(1) = \boldsymbol{e}_2),$$

$$\bar{\boldsymbol{w}}(3)^\top \boldsymbol{x}\bar{\Phi}|_{\boldsymbol{y}=\boldsymbol{e}_1} = 0 + \sum_{j=1}^{6} \mathbb{I}(\boldsymbol{Q}_{s,j}(1) = \boldsymbol{e}_3).$$

We will have $\bar{\boldsymbol{w}}(2)^\top \boldsymbol{x}\bar{\Phi}|_{\boldsymbol{y}=\boldsymbol{e}_1} > \bar{\boldsymbol{w}}(1)^\top \boldsymbol{x}\bar{\Phi}|_{\boldsymbol{y}=\boldsymbol{e}_1}$ if either of the following occurs

- $\{\mathbb{I}(\boldsymbol{Q}_{s,j}(1) = \boldsymbol{e}_2)\}_{j=1}^{6}$ holds simultaneously, whose probability would be $\frac{p^6}{729}$

- Five of $\{\mathbb{I}(\boldsymbol{Q}_{s,j}(1)\}_{j=1}^{6}$ takes $\boldsymbol{e}_2$ and the remaining one takes $\boldsymbol{e}_3$, i.e., $\sum_{j=1}^{6} \mathbb{I}(\boldsymbol{Q}_{s,j}(1) = \boldsymbol{e}_2) = 5$ and $\sum_{j=1}^{6} \mathbb{I}(\boldsymbol{Q}_{s,j}(1) = \boldsymbol{e}_3) = 1$. Such probability is $\frac{6p^6}{729}$.

The total probability is then $\frac{7p^6}{729} \approx \frac{p^6}{104} \leq \frac{p^3}{104} < \frac{p^3}{27}$. See Figure 14 for a visualization of the main intuition.

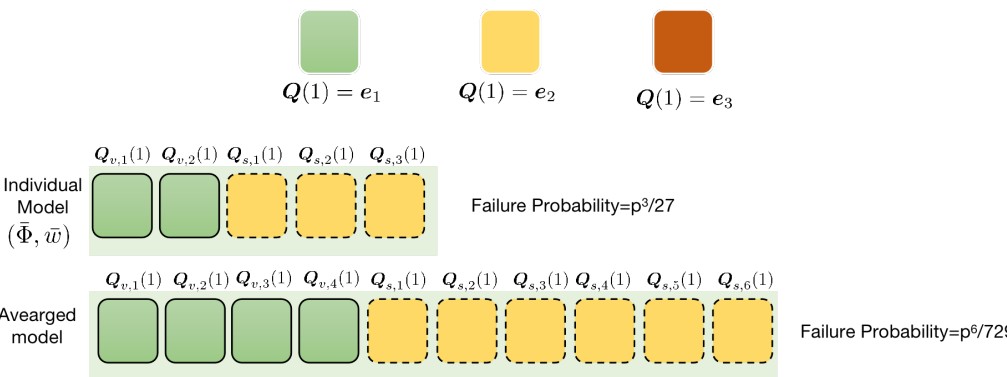

Figure 14: Comparison of the failure probability of individual model and averaged model. With probability $p^3/27$, the individual model $(\bar{\Phi}, \bar{w})$ will encounter an OOD distribution where it mistakenly predicting the second class $e_2$ on the samples from the first class $e_1$. For the averaged model, such probability would be roughly about $p^6/729$. Refer to Appendix D.6 for detailed explanation.

## D.7 THE DIFFERENCE BETWEEN WSE AND OSE IN OOD

### D.7.1 EXPLAINING THE DIFFERENCE BETWEEN WSE AND OSE

We use the following Example 3 to show the main intuition of the difference between model averaging and ensemble.

**Example 3.** *Two individual models learn overlapped features $\boldsymbol{x}_{v,2}$ and $\boldsymbol{x}_{s,3}$ as*

$$\boldsymbol{x}\bar{\Phi}^\top = \boldsymbol{x}_{v,1} + \boldsymbol{x}_{v,2} + \boldsymbol{x}_{s,1} + \boldsymbol{x}_{s,2} + \boldsymbol{x}_{s,3}, \quad \boldsymbol{x}\tilde{\Phi}^\top = \boldsymbol{x}_{v,2} + \boldsymbol{x}_{v,3} + \boldsymbol{x}_{s,3} + \boldsymbol{x}_{s,4} + \boldsymbol{x}_{s,5},$$

**Proposition 5.** *Consider the Example 3, suppose Assumption 1 and 2 hold, and there are infinite ID and OOD samples, the averaged and ensemble models are defined as Definition 3. Omitting small terms containing $\epsilon$, we have $\mathcal{A}_{ood}(\bar{f}) = \mathcal{A}_{ood}(\tilde{f}) = 1 - \frac{1}{9}p^3$ and $\mathcal{A}_{ood}(f_{ose}) = 1 - \frac{4p^4}{81} - \frac{8p^5}{243}$ and $\mathcal{A}_{ood}(f_{WSE}) = 1 - \frac{4p^4}{81} - \frac{p^5}{27}.$*

Full Proof in Appendix F.4. In Example 3, two individual models learn overlapped feature, $\boldsymbol{x}_{v,i}(k)$ and $\boldsymbol{x}_{s,3}(k)$. By Lemma 5 , for $k = 1, 2, 3$, we have

$$\bar{w}(k) = \sum_{i=1}^{2} \boldsymbol{\mu}_{v,i}(k) + \sum_{j=1}^{3} \boldsymbol{\mu}_{s,j}(k),$$

$$\tilde{w}(k) = \sum_{i=2}^{3} \boldsymbol{\mu}_{v,i}(k) + \sum_{j=3}^{5} \boldsymbol{\mu}_{s,j}(k),$$

So we have

$$\bar{w}(k) + \tilde{w}(k) = \sum_{i=1,3} \boldsymbol{\mu}_{v,i}(k) + 2\boldsymbol{\mu}_{v,2}(k) + \sum_{j=1,2,4,5} \boldsymbol{\mu}_{s,j}(k) + 2\boldsymbol{\mu}_{s,3}(k)$$

For samples from the first class, we also have

$$\boldsymbol{x}(\bar{\Phi}+\tilde{\Phi})|_{\boldsymbol{y}=\boldsymbol{e}_1} = \sum_{i=1,3} \boldsymbol{\mu}_{v,i}\boldsymbol{Q}_{v,i}(1) + 2\boldsymbol{\mu}_{v,2}\boldsymbol{Q}_{v,2}(1) + \sum_{j=1,2,4,5} \boldsymbol{\mu}_{s,j}\boldsymbol{Q}_{s,j}(1) + 2\boldsymbol{\mu}_{s,3}\boldsymbol{Q}_{s,3}(1) + \sum_{i=1}^{10} \boldsymbol{z}_i$$

where $\boldsymbol{z}_i \sim \mathcal{N}(0, \sigma^2 \boldsymbol{I}_d), \forall i$. We then have

$$(\bar{w}(k) + \tilde{w}(k))^\top \boldsymbol{x}(\bar{\Phi} + \tilde{\Phi})|_{\boldsymbol{y}=\boldsymbol{e}_1}$$

$$= \sum_{i=1,3} \boldsymbol{\mu}_{v,i}(k)^\top \left(\boldsymbol{\mu}_{v,i}\boldsymbol{Q}_{v,i}(1)\right) + 4\boldsymbol{\mu}_{v,2}(k)^\top \left(\boldsymbol{\mu}_{v,2}\boldsymbol{Q}_{v,2}(1)\right)$$

$$+ \sum_{j=1,2,4,5} \boldsymbol{\mu}_{s,j}(k)^\top \left(\boldsymbol{\mu}_{s,j}\boldsymbol{Q}_{s,j}(1)\right) + 4\boldsymbol{\mu}_{s,3}(k)^\top \left(\boldsymbol{\mu}_{s,3}\boldsymbol{Q}_{s,3}(1)\right) + \xi \tag{5}$$

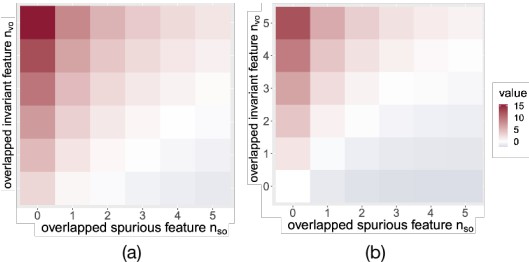

Figure 15: (a) $\mathcal{A}_{\text{ood}}(f_{\text{wse}}) - \mathcal{A}_{\text{ood}}(\bar{f})$ on Example 4, (b) $\mathcal{A}_{\text{ood}}(f_{\text{wse}}) - \mathcal{A}_{\text{ood}}(f_{\text{ose}})$ on Example 4,

As for model ensemble, we have

$$
\begin{aligned}
&\bar{\boldsymbol{w}}(k)^\top \boldsymbol{x}\bar{\Phi} + \tilde{\boldsymbol{w}}(k)^\top \boldsymbol{x}\tilde{\Phi} \\
&= \sum_{i=1,2} \boldsymbol{\mu}_{v,i}(k)^\top \left(\boldsymbol{\mu}_{v,i}\boldsymbol{Q}_{v,i}(1)\right) + \sum_{j=1,2,3} \boldsymbol{\mu}_{s,j}(k)^\top \left(\boldsymbol{\mu}_{s,j}\boldsymbol{Q}_{s,j}(1)\right) \\
&\quad + \sum_{i=2,3} \boldsymbol{\mu}_{v,i}(k)^\top \left(\boldsymbol{\mu}_{v,i}\boldsymbol{Q}_{v,i}(1)\right) + \sum_{j=3,4,5} \boldsymbol{\mu}_{s,j}(k)^\top \left(\boldsymbol{\mu}_{s,j}\boldsymbol{Q}_{s,j}(1)\right) \qquad (6) \\
&= \sum_{i=1,3} \boldsymbol{\mu}_{v,i}(k)^\top \left(\boldsymbol{\mu}_{v,i}\boldsymbol{Q}_{v,i}(1)\right) + 2\boldsymbol{\mu}_{v,2}(k)^\top \left(\boldsymbol{\mu}_{v,2}\boldsymbol{Q}_{v,2}(1)\right) \\
&\quad + \sum_{j=1,2,4,5} \boldsymbol{\mu}_{s,j}(k)^\top \left(\boldsymbol{\mu}_{s,j}\boldsymbol{Q}_{s,j}(1)\right) + 2\boldsymbol{\mu}_{s,3}(k)^\top \left(\boldsymbol{\mu}_{s,3}\boldsymbol{Q}_{s,3}(1)\right) + \xi' \qquad (7)
\end{aligned}
$$

Comparing equation 5 and equation 6, we can see that

- For model averaging, the overlapped features $\boldsymbol{\mu}_{v,2}$ and $\boldsymbol{\mu}_{s,3}$ (corresponding to $\boldsymbol{x}_{v,2}$ and $\boldsymbol{x}_{s,3}$) have coefficients amplified by 2 in $\bar{\Phi} + \tilde{\Phi}$, and further amplified twice in $\bar{\boldsymbol{w}} + \tilde{\boldsymbol{w}}$. This results in coefficients of the overlapped feature becoming 4 in $(\bar{\boldsymbol{w}} + \tilde{\boldsymbol{w}})^\top \boldsymbol{x}(\bar{\Phi} + \tilde{\Phi})$.
- For model ensemble, i.e., $\bar{\boldsymbol{w}}^\top \boldsymbol{x}\bar{\Phi} + \tilde{\boldsymbol{w}}^\top \tilde{\boldsymbol{x}}\Phi$, the coefficients of the overlapped feature are 2.

### D.7.2 THE THEORETICAL CONDITION OF WSE OUTPERFORMING OSE

Recall Proposition 2 that we have

$$
\begin{aligned}
\mathcal{A}_{\text{ood}}(f_{\text{wse}}) &= F_p \left( \frac{(1-p)(\tilde{n}_s + \bar{n}_s + 2n_{so}) + (\tilde{n}_v + \bar{n}_v + 2n_{vo})}{\sqrt{\tilde{n}_s + \bar{n}_s + 14n_{so}}} \right), \\
\mathcal{A}_{\text{ood}}(f_{\text{ose}}) &= F_p \left( \frac{(1-p)(\tilde{n}_s + \bar{n}_s) + (\tilde{n}_v + \bar{n}_v)}{\sqrt{\tilde{n}_s + \bar{n}_s + 2n_{so}}} \right).
\end{aligned}
$$

A direct consequence of Proposition 2 is as follows, which illustrates when model averaging can be more effective than model ensemble:

**Proposition 6.** *Consider the models in Definition 2, suppose Assumption 1 and 2 hold, there are infinite ID and OOD samples. Suppose the number of features that $\bar{\Phi}$ and $\tilde{\Phi}$ learn are the same, i.e., $\bar{n}_v = \tilde{n}_v \doteq n_v$, $\bar{n}_s = \tilde{n}_s \doteq n_s$ and denote $\rho_s \doteq n_{so}/n_s, \rho_v \doteq n_{vo}/n_v$. Omitting small constants involving $\epsilon$, we have $\mathcal{A}_{ood}(f_{wse}) > \mathcal{A}_{ood}(f_{ose})$ when $\frac{\rho_v}{\rho_s} > \frac{3(1-p)n_s}{n_v}$, and $\mathcal{A}_{ood}(f_{wse}) \leq \mathcal{A}_{ood}(f_{ose})$ when $\frac{\rho_v}{\rho_s} \leq \frac{3(1-p)n_s}{n_v}$.*

As shown in Appendix D.7.1, the coefficient of an overlapped feature in model averaging is 4, and The coefficient of an overlapped feature in model ensemble is 2. If more $\bar{\Phi}$ and $\tilde{\Phi}$ learns more overlapped invariant features, the model averaging would put more weight on the invariant features, leading to better OOD performance.

In Figure 4 (c) and (d), we illustrate $\mathcal{A}_{\text{ood}}(f_{\text{wse}}) - \mathcal{A}_{\text{ood}}(\bar{f})$ and $\mathcal{A}_{\text{ood}}(f_{\text{wse}}) - \mathcal{A}_{\text{ood}}(f_{\text{ose}})$ on the following example:

**Example 4.** *Consider both models learn the same number of features, i.e., fixing $\bar{n}_v = \tilde{n}_v = 10$ and $\bar{n}_s = \tilde{n}_s = 20$, vary $n_{vo} = 0, 1, ..., 5$ and $n_{so} = 0, 1, ..., 5$.*

We can see that $f_{\text{wse}}$ achieves larger OOD improvement over $f_{\text{ose}}$ when two individual models learns more overlapped invariant features (e.g., larger $n_{vo}$) and less overlapped spurious features (e.g., smaller $n_{so}$). In Appendix D.7.2, we provide conditions when $f_{\text{wse}}$ outperforms $f_{\text{ose}}$, discuss why this can happen easily in real-world datasets, provide some primary experimental results.

**Why does it easily happen in OOD on many real-world applications?** Recall that there are totally $d_v$ invariant features and $d_s$ spurious features. It is a common believe that spurious features are high-dimensional and invariant features are low-dimensional, i.e., $d_s \gg d_v$ (Arjovsky et al., 2019; Rosenfeld et al., 2020). Since the spurious features are high dimensional and (Allen-Zhu & Li, 2020; Zhang & Bottou, 2023) indicate that different models can learn different (limited size) subsets of features, the overlap ratio of spurious feature $\rho_s$ is relatively low. On the other hand, there are a small number of invariant features and recent studies (Rosenfeld et al., 2022; Qiu et al., 2023; Kirichenko et al., 2022) show that models always learn some invariant features for the fine-tuned task during ERM fine-tuning regardless of the presence of spurious features, so we conjecture that the overlapped ratio of invariant feature $\rho_v$ is relatively higher.

However, we recognize that our discussion regarding the overlap ratio of invariant spurious features being larger than spurious features is not supported by rigorous proof, but rather it remains a conjecture. Further research in this area is necessary to provide more conclusive evidence and establish a solid foundation for this claim. In the next part, we will conduct experiments to provide some initial support for this conjecture.

### D.7.3 EMPIRICAL VERIFICATION

It is very difficult to directly empirically verified Proposition 6 because

- For real-world datasets, it is hard to identify whether and how much a model relies on invariant or spurious features. Verifying Proposition 6 needs to estimate how much two models relies on the same feature.

- For synthetic datasets, such as CMNIST (Arjovsky et al., 2019), there is no feasible pre-trained models available. On the other hand, weight space ensemble needs to be conducted on models close to pre-trained models.

In this part, we design a primary experiment to get around the above obstacles. Consider the ensemble of two models: pre-trained CLIP ($\bar{f}$) and the CLIP fine-tuned on ImageNet ($\tilde{f}$).

First, we use ImageNet variants (ImageNet-V2, ImageNet-Sketch, ImageNet-A, ImageNet-R, ObjectNet) for OOD performance evaluation. Recall that ImageNet variants share the same invariant features with ImageNet. Also recent studies (Rosenfeld et al., 2022; Qiu et al., 2023; Kirichenko et al., 2022) show that ERM fine-tuned models always learn some invariant features for the fine-tuned task regardless of the presence of spurious features. So $\tilde{f}$ learns the invariant features for ImageNet variants. At the same time, the pre-trained CLIP $\bar{f}$ can stably perform zero-shot classification on ImageNet and its variants, indicating that $\bar{f}$ also learns good invariant features for ImageNet variants. According to the previous discussion, $\bar{f}$ and $\tilde{f}$ have some overlapped invariant features for ImageNet variants, leading to better weight space ensemble than output space ensemble on ImageNet variants (shown in Figure 16(Left)).

We then evaluate the OSE and WSE on three other distinct datasets, i.e., Places365, StanfordCars, DTD and Food101 (refer as PSDF datasets). These tasks have different label space with ImageNets, and contains different invariant features with ImageNet. Then in this case, the model $\tilde{f}$ fine-tuned on the ImageNet learns little invariant for PSDF datasets. So overlap invariant features used by the pre-trained model $\bar{f}$ and fine-tuned $\tilde{f}$ are rather limited, indicating $\rho_v$ is close to zero. Then according to Proposition 6, WSE would be no better than OSE. This is consistent with the results in Figure 16(right).

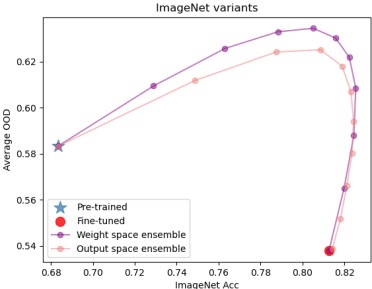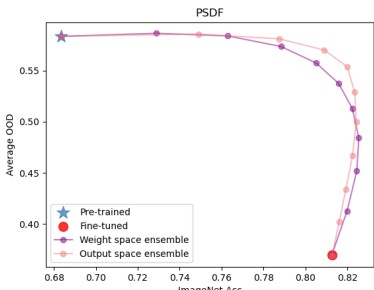

Figure 16: Comparison of model ensemble and averaging. Left) OOD performance on ImageNet variants, Right) OOD performance on PSDF (Places365, StanfordCars, DTD and Food101).

### D.8 ILLUSTRATING THE OVER-CONFIDENCE.

In Section 4, we use $\lambda$ to characterize the over-confidence of $f_\lambda = (\lambda w, \lambda \Phi)$. Specifically, we have

$$f_\lambda(x) = \lambda^2 w^\top x \Phi.$$

Denote $q := w^\top x \Phi$, which is a 3-dimensional vector for a 3-class classification problem. Consider an example, i.e., $q = [2, 1, 1]$. Recall that $q(k)$ is the $k$-th element of $q$ for $k = 1, 2, 3.$. The predicted probability for the first class when $\lambda = 1$ is

Probability of class 1 $= \dfrac{\exp(q(1))}{\exp(q(1)) + \exp(q(2))\exp(q(3))} = \dfrac{\exp(2)}{\exp(2) + \exp(1) + \exp(1)} = 0.576.$

When the predicted class of $f_\lambda$ would be the same for $\lambda > 1$ and $\lambda = 1$. Whereas, when $\lambda > 1$, the predicted probability for the largest class would be amplified, e.g., when $\lambda = \sqrt{5}$

$$
\begin{aligned}
\text{Probability of class 1} &= \frac{\exp(\lambda^2 q(1))}{\exp(\lambda^2 q(1)) + \exp(\lambda^2 q(2))\exp(\lambda^2 q(3))} \\
&= \frac{\exp(2\lambda^2)}{\exp(2\lambda^2) + \exp(\lambda^2) + \exp(\lambda^2)} \\
&= 0.99
\end{aligned}
$$

So we can see that a larger $\lambda$ won't change the predicted class, but would make $f_\lambda$ more confident.

## E MORE EXPERIMENTAL DETAILS AND RESULTS ON BANG

### E.1 DETAILS ON IMAGENET VARIANTS

Details for ImageNet variants:

- ImageNet-V2(IN-V2): A recreated version of the ImageNet test set, but with a different set of data distribution.

- ImageNet-R(IN-R): Renditions of 200 ImageNet classes resulting in 30,000 images.

- ImageNet Sketch(IN-S): Sketch style images of the same categories as ImageNet, with a total of 50000 images.

- ObjectNet(ON): Objects in this dataset are captured in cluttered and natural environments at unusual poses.

- ImageNet-A(IN-A): This dataset consists of naturally occurring images that are misclassified by a ResNet-50 model for 200 ImageNet classes.

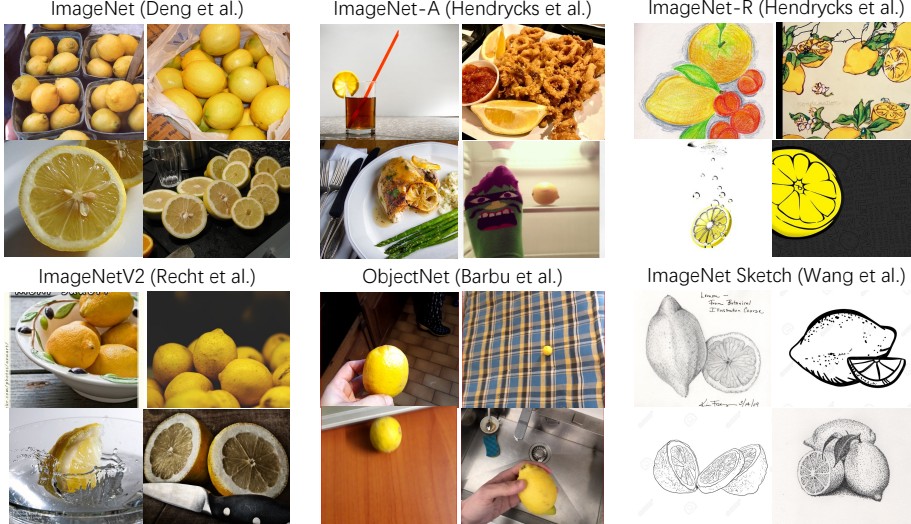

Figure 17: Datasets on ImageNet and its variants. For each dataset, we pick 4 samples of the class *lemon* and show illustrative images from each dataset. The dataset descriptions are similar to that of Wortsman et al. (2022).

### E.2 DETAILS OF PLACES365, STANFORDCARS, DTD AND FOOD101 (PSDF)

- Places365 (Zhou et al. (2017)): A scene recognition dataset. In this paper, we use the validation set from the Places365-standard, which is composed of 36,000 validation images from 365 scene classes.

- StanfordCars (Krause et al. (2013)): This dataset contains 196 classes of cars. Classes are typically at the level of Make, Model, Year, ex. 2012 Tesla Model S or 2012 BMW M3 coupe. In this paper, we evaluate models on the test set, comprising 8,041 images.

- Describable Textures Dataset (DTD) (Cimpoi et al. (2014)): DTD is a texture database, organized according to a list of 47 categories inspired from human perception such as banded, dotted and gauzy. In the paper, we use the test set with 40 images per class.

- Food101 (Bossard et al. (2014)): This dataset consists of 101 food categories. In the paper, we use the test set with 250 test images for each class.

### E.3 DETAILS ON CALCULATING THE CONFIDENCE

Consider a $K$-class classification problem. Denote the $l$th element of the output as $\text{Prob}_l$, indicating the probability the model assigns to the $l$th class. We have $\sum_{l=1}^{K} \text{Prob}_l = 1$. The confidence is defined as as:

$$\text{Confidence} = \max\left(\{\text{Prob}_l\}_{l=1}^{K}\right).$$

### E.4 EXPERIMENTAL DETAILS

We use the CLIP model ViT-B/16Radford et al. (2021). We fine-tune the pre-trained model on ImageNet. We use the AdamW optimizer with the default PyTorch AdamW hyperparameters and choose 512 as batch size. We use a learning rate of $3 \times 10^{-5}$, gradient clipping at global norm 1 and fine-tune for a total of 10 epochs. The settings mentioned above are the same with Wortsman et al. (2022). For our method BANG, we try four smoothing for LS (label smoothing): 0.05, 0.10, 0.15 and 0.20. We adopt 0.10 in our reported results in Table 2. Further results in Table 9 show that BANG is relatively insensitive to the hyper-parameter. We do not tune the hyper-parameters of Mixup Zhang et al. (2017). We use the default hyperparamter as MMPreTrainContributors (2023).

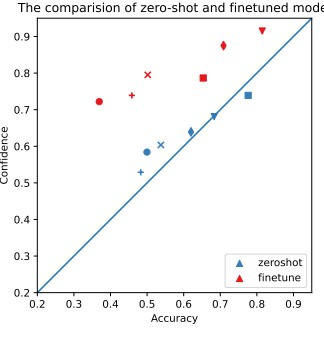

(a) The vanilla fine-tuned model

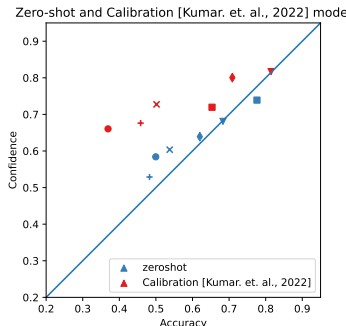

(b) The vanilla fine-tuned model calibrated on in the ID dataset Kumar et al. (2022).

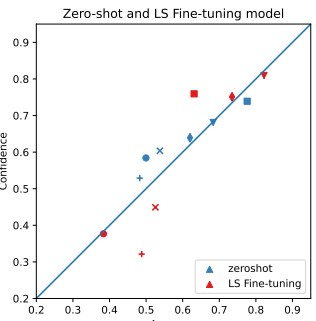

(c) The model fine-tuned with label smoothing

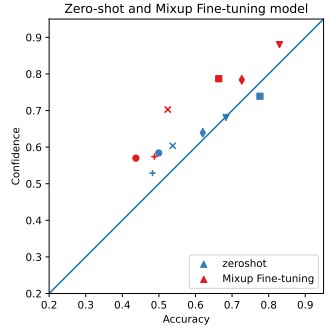

(d) The model fine-tuned with Mixup

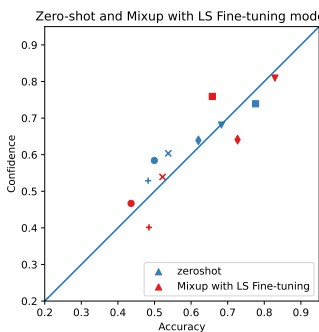

(e) The model fine-tuned with both Mixup and LS

Figure 18: Comparison of confidence and accuracy between zero-shot and the model finetuned with different methods. In the figure, the ID dataset is the ImageNet dataset, which is represented by ▽. the five OOD datasets are: ○ for ImageNetA, □ for ImageNetR, + for ImageNetSketch, ◇ for ImageNetV2 and × for ObjectNet.

| Methods | Model Averaging | IN | IN-V2 | IN-R | IN-A | IN-S | ObjectNet | Avg OOD |
|---|---|---|---|---|---|---|---|---|
| Zero-shot Wortsman et al. (2022) | No | 68.3 | 61.9 | 77.6 | 49.8 | 48.2 | 53.0 | 58.1 |
| Fine-tuning Wortsman et al. (2022) | No | 81.3 | 70.9 | 65.6 | 36.7 | 46.3 | 49.6 | 53.8 |
| Flip | No | 81.3 | 70.5 | 63.1 | 36.8 | 44.6 | 51.4 | 53.3 |
| Rotate | No | 81.4 | 70.7 | 65.2 | 35.6 | 45.3 | 49.5 | 53.3 |
| Color | No | 81.4 | 71.5 | 65.3 | 37.3 | 46.7 | 50.4 | 54.2 |
| Mixup | No | 83.0 | 72.7 | 66.4 | 43.7 | 48.8 | 52.4 | 56.8 |
| Flip | Yes | 81.8 | 72.7 | 78.2 | 52.9 | 53.6 | 58.4 | 63.1 |
| Rotate | Yes | 81.7 | 72.8 | 78.8 | 52.7 | 53.7 | 57.3 | 63.1 |
| Color | Yes | 81.7 | 72.9 | 78.5 | 53.2 | 54.2 | 58.2 | 63.4 |
| Mixup | Yes | 81.5 | 73.0 | 79.5 | 57.9 | 54.5 | 58.7 | 64.7 |

Table 8: Results of fine-tuning CLIP VIT-B/16 with flip, color, and rotation data augmentation on ImageNet.

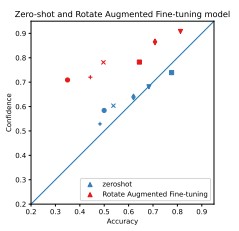 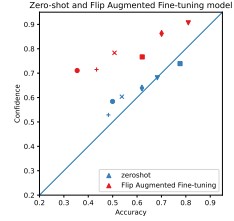 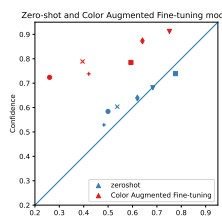

(a) With rotate augmentation     (b) With flip augmentation     (c) With color augmentation

Figure 19: Comparison of confidence and accuracy between zero-shot and the model finetuned with different data augmentation. In the figure, $\triangledown$ refers to ImageNet dataset, $\circ$ for ImageNetA, $\square$ for ImageNetR, $+$ for ImageNetSketch, $\diamondsuit$ for ImageNetV2 and $\times$ for ObjectNet.

### E.5 MORE RESULTS ON BANG AND DISCUSSIONS

**Mixup and label smoothing can alleviate the over-confidence of the fine-tuned model** Compare Figure 18c, 18d, 18e with Figure 18a, we can see that imposing Mixup and LS during fine-tuning can alleviate the over-confidence of the fine-tuned model on both ID (ImageNet, denoted by $\triangledown$ in the figure) and OOD datasets, which is consistent with existing results Park & Caragea (2022).

**Comparison with Calibrated Ensemble Kumar et al. (2022).** Kumar et al. (2022) calibrates the fine-tuned model on the ID dataset by searching for a temperature $T$ of the softmax. Figure 18b shows that the confidence of the calibrated fine-tuned model approximately equals its accuracy on the ID dataset (ImageNet). However, such model is still highly over-confidence in OOD datasets, e.g., the confidence is over 0.6 while the accuracy is lower than 0.4 on ImageNetA (denoted by $\circ$), which is consistent with the findings in Ovadia et al. (2019) and also the discussions in the Section 4.2 of Ovadia et al. (2019). So the scaling issue shown in Proposition 4 still exists in OOD datasets. Notably, Calibrated Ensemble itself Kumar et al. (2022) can not be directly applied on model averaging: Model averaging merges the parameters of each layer. However, calibrated Ensemble only tunes the temperature of the softmax, which does not affect the lower layers, indicating that the layers other than the output layer can still suffer from scaling issues. We try a direct adaptation of (Kumar et al., 2022) to WiSE-FT: divide the weights in the last layer $w$ by a scalar (temperature) and then perform weight averaging. This also does not yields satisfactory results (Appendix E.6) and the reason is discussed above.

**Comparison between Mixup with other data augmentations** We also compare Mixup with other data augmentations. We fine-tune the CLIP on ImageNet with flip, rotate, and color augmentation, respectively. We then performance weight averaging on these fine-tuned model with the pre-trained model as Wortsman et al. (2022) does. Table 8 shows that flip, rotate, and color augmentation can not enhance the performance of model averaging. Figure 19 also shows that these augmentation methods can not alleviate the over-confidence of the fine-tuned model.

**BANG is relatively insensitive to hyper-parameters**. Table 9 shows the performance of BANG with different hyper-parameters of label smoothing. BANG is relatively insensitive to such hyper-parameters, e.g., the average OOD performance of BANG(Mixup+LS) all remains at about 64.9% for the four hyper-parameters.

| Methods | Model Averaging | IN(ImageNet) | IN-V2 | IN-R | IN-A | IN-Sketch | ObjectNet | Avg OOD |
|---|---|---|---|---|---|---|---|---|
| Zero-shot Wortsman et al. (2022) | No | 68.3 | 61.9 | 77.6 | 49.8 | 48.2 | 53.0 | 58.1 |
| Fine-tuning Wortsman et al. (2022) | No | 81.3 | 70.9 | 65.6 | 36.7 | 46.3 | 49.6 | 53.8 |
| Fine-tuning(LS(0.05)) | No | 82.0 | 71.5 | 62.8 | 37.7 | 45.5 | 50.6 | 53.6 |
| Fine-tuning(LS(0.10)) | No | 82.0 | 72.3 | 63.3 | 38.3 | 46.5 | 51.1 | 54.3 |
| Fine-tuning(LS(0.15)) | No | 82.1 | 72.1 | 63.3 | 38.0 | 46.6 | 50.7 | 54.1 |
| Fine-tuning(LS(0.20)) | No | 82.1 | 72.1 | 62.8 | 36.9 | 46.2 | 50.5 | 53.7 |
| Fine-tuning(Mixup) | No | 83.0 | 72.7 | 66.4 | 43.7 | 48.8 | 52.4 | 56.8 |
| Fine-tuning(Mixup + LS(0.05)) | No | 83.0 | 73.2 | 65.9 | 43.9 | 48.5 | 52.3 | 56.7 |
| Fine-tuning(Mixup + LS(0.10)) | No | 82.7 | 73.0 | 66.4 | 43.3 | 48.6 | 52.4 | 56.8 |
| Fine-tuning(Mixup + LS(0.15)) | No | 82.9 | 72.7 | 65.8 | 43.6 | 48.5 | 52.2 | 56.6 |
| Fine-tuning(Mixup + LS(0.20)) | No | 82.9 | 73.2 | 66.4 | 44.6 | 48.5 | 52.4 | 57.0 |
| WiSE-FT Wortsman et al. (2022) | Yes | 81.7 | 72.8 | 78.7 | 52.2 | 53.9 | 57.3 | 63.0 |
| BANG(LS(0.05)) | Yes | 82.2 | 73.0 | 78.1 | 54.7 | 53.8 | 58.3 | 63.6 |
| BANG(LS(0.10)) | Yes | 82.1 | 73.3 | 78.2 | 55.2 | 53.7 | 58.9 | 63.9 |
| BANG(LS(0.15)) | Yes | 82.0 | 73.2 | 78.1 | 55.0 | 53.4 | 58.9 | 63.7 |
| BANG(LS(0.20)) | Yes | 81.7 | 73.1 | 77.9 | 54.2 | 53.6 | 58.6 | 63.4 |
| BANG(Mixup) | Yes | 81.5 | 73.0 | 79.5 | 57.9 | 54.5 | 58.7 | 64.7 |
| BANG(Mixup + LS(0.05)) | Yes | 81.6 | 73.1 | 79.7 | 58.2 | 54.8 | 58.9 | 64.9 |
| BANG(Mixup + LS(0.10)) | Yes | 81.5 | 73.0 | 79.8 | 57.9 | 54.8 | 59.0 | 64.9 |
| BANG(Mixup + LS(0.15)) | Yes | 81.7 | 72.9 | 79.6 | 57.7 | 54.6 | 59.1 | 64.8 |
| BANG(Mixup + LS(0.20)) | Yes | 81.6 | 73.1 | 79.9 | 57.8 | 54.8 | 59.0 | 64.9 |

Table 9: Results of BANG with CLIP-B/16. We show different hyper-parameters of label smoothing. Mixup use the default hyper-parameter of MMPreTrainContributors (2023).

| WiSE-FT | Exp 1 | Exp 2 | BANG |
|---|---|---|---|
| 63.0% | 63.0% | 64.1% | 64.9% |

Table 10: WiSE-FT can benefit significantly from better calibration by scaling the fine-tuned model. (Exp 1)-(Exp 2) are described in Appendix E.6.

### E.6   WISE-FT BENEFITS SIGNIFICANTLY FROM BETTER CALIBRATION

In Section 4, we theoretically show that model WSE can suffer from the imbalance issue where two individual models have different scaling. This can happen if one model is much more confident than the other. Unfortunately, we observe that the popular method, WiSE-FT suffers from this issue. Specifically, WiSE-FT averages the pre-trained model with the fine-tuned model. In Section 4, we show that the fine-tuned model is high-overconfident compared with the pre-trained model. We propose BANG, which averages the pre-trained model with the model fine-tuned with Label Smoothing (LS) or MixUp. Since LS and MixUp can also improve the fine-tuned performance, we conduct the following experiment to isolate the effect of better calibration from better fine-tuned performance.

**Scale the fine-tuned model during weight space ensemble**. A straightforward method to alleviate over-confidence is to tune the temperature of the softmax of the fine-tuned model (Kumar et al., 2022). However, this method can not be directly applied to WiSE-FT since WiSE-FT averages model weights instead of ensemble the outputs. We first apply a direct adaptation of (Kumar et al., 2022) to WiSE-FT: divide the weights in the last layer $w$ by a scalar (temperature), which is equivalent to softmax tempering. However, recall the model averaging $(\bar{w} + \tilde{w})^\top x (\bar{\Phi} + \tilde{\Phi})$ also suffer from the imbalance issue of $\Phi$. Specifically, Proposition 4 shows that the averaged feature can be biased towards $\tilde{\Phi}$ if the scaling of $\tilde{\Phi}$ is larger than $\bar{\Phi}$. So merely adjusting the weight of the classifier $w$ cannot alleviate this bias. The experiment result (Exp 1) in Table 10 also shows that merely re-scaling the classifier can hardly improve WiSE-FT. In practice, we use a transformer (VIT-B/16) with 12 block layers and 1 linear layer. We obtain the averaged model $(\hat{\theta}, \hat{w})$ as follows

- (Exp 1) Re-scale the classifier of FM (fine-tuned, $\tilde{\theta}$, $\tilde{w}$) model during averaging, i.e., $\hat{\theta} = 0.5(\bar{\theta} + \tilde{\theta})$ and $\hat{w} = (1 - \alpha)\bar{w} + \alpha\tilde{w}$.

- (Exp 2) Re-scale whole network of FM as, $\hat{\theta} = (1 - \alpha)\bar{\theta} + \alpha\tilde{\theta}$ and $\hat{w} = (1 - \alpha)\bar{w} + \alpha\tilde{w}$.

We search for the best $\alpha$ among 0.2-0.5 (with interval 0.1) for each ood dataset. The results in Table 10 shows can merely scaling the fine-tuned model to alleviate its over-confidence can significantly improvement the performance of WiSE-FT.

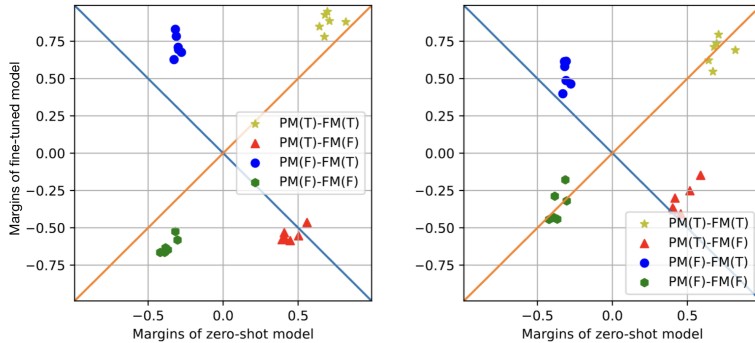

Figure 20: (Left) Margins of WiSE-FT, where the fine-tuned model is obtained through vanilla fine-tuning. (Right) Margins of BANG, where the fine-tuned model is obtained through fine-tuning with MixUP+LS.

|  | $N_1/N_2$ |
| --- | --- |
| WiSE-FT | 10.6% |
| BANG | 13.4% |

Table 11: The ratio (versus the entire dataset) of samples where model averaging can correct the prediction in the PM(T)-FM(F) group. $N_1$ denote the number of samples where model averaging can correct the prediction in the PM(T)-FM(F) group and $N_2$ denote the total sample size in the dataset.

**BANG can correct more samples on which the fine-tuned model make mistakes.** In this part, We denote the pre-trained model as PM, fine-tuned model as FM, and averaged model as AM. We divide each dataset into four groups of samples according to whether the PM and FM make correct predictions, respectively. We further use T/F to denote whether a model makes correct predictions, i.e., T for True and F for False. For example, we use PM(T)-FM(F) to denote the group of samples on which the predictions of PM and FM AM are correct and wrong, respectively. We visualize the average margin of fine-tuned and pre-trained models on four groups, i.e., PM(T)-FM(T), PM(T)-FM(F), PM(F)-FM(T), and PM(F)-FM(F). The margin is the difference between the probability assigned to the correct class and the maximum probability among the wrong classes, i.e.,

$$\text{Margin} = \text{Prob}_l - \max_{k \neq l} \text{Prob}(k)$$

where $l$ is the it the true class, $k = 1, ..K$, and $\text{Prob}(k)$ is the probability that a assign to the class $k$. Averaging models with negative and positive margins can potentially correct mistakes. Figure 20 (Left) and (Right) visualize the the margins of pre-trained and fine-tuned models on each group of each datasets for Wise-ft and BANG. In Wise-ft, the fine-tuned model exhibits significantly negative margins in the PM(T)-FM(F) group. Specifically, Margin(PM) + Margin(FM) on the group PM(T)-FM(F) is negative for WiSE-FT on Figure 20(Left), indicating dominance of fine-tuned models in WiSE-FT even fine-tuned make mistakes. This also explains that why in some datasets, e.g., IN-R and IN-A, ImproveContri(TF+FT) is negative as shown in Figure 7. However, in BANG, Margin(PM) + Margin(FM) on the group PM(T)-FM(F) is positive on average as shown in Figure 20(Right), suggesting that BANG is capable of correcting more mistakes within the PM(T)-FM(F) group. Table 11 shows that ratio (versus the entire dataset) of samples where model averaging can correct the prediction in the PM(T)-FM(F) group. Specifically, let $N_1$ denote the number of samples where WiSE-FT can correct the prediction in the PM(T)-FM(F) group and $N_2$ denote the total sample size in the dataset, Table 11 compares the $N_1/N_2$ (averaged over 5 OOD datasets) of WiSE-FT and BANG. Table 11 shows BANG can correct substantially more mistakes made by the fine-tuned model.

# F PROOFS

## F.1 PROOF OF PROPOSITION 1

*Proof.* (a) **Two individual models**. Recall that in the 3-class classification problem, $\boldsymbol{w} = [\boldsymbol{w}(1), \boldsymbol{w}(2), \boldsymbol{w}(3)] \in \mathbb{R}^{d \times 3}$. We first solve the $\bar{\boldsymbol{w}}$ on the infinite ID samples. Lemma 5 , for $k = 1, 2, 3$, we have

$$\bar{\boldsymbol{w}}(k) = \sum_{i=1}^{2} \boldsymbol{\mu}_{v,i} \boldsymbol{Q}_{v,i}(k) + \sum_{j=1}^{3} \boldsymbol{\mu}_{s,j} \boldsymbol{Q}_{s,j}(k) = \sum_{i=1}^{2} \boldsymbol{\mu}_{v,i}(k) + \sum_{j=1}^{3} \boldsymbol{\mu}_{s,j}(k),$$

where the last inequality is because: $\boldsymbol{Q}_{v,i} = \boldsymbol{I}$ always hold and $\boldsymbol{Q}_{s,j} = \boldsymbol{I}$ in the ID distribution. Then $\boldsymbol{Q}_{v,i}(k) = \boldsymbol{e}_k$ and $\boldsymbol{Q}_{s,j}(k) = \boldsymbol{e}_k$ (recall that $\boldsymbol{Q}(k)$ is the $k$th column of the $3 \times 3$ matrix $\boldsymbol{Q}$). Then for each $\boldsymbol{\mu}\boldsymbol{Q}(k) = \boldsymbol{\mu}\boldsymbol{e}_k = \boldsymbol{\mu}(k)$ and $\boldsymbol{\mu}(k)$ is the $k$th column of the $d \times 3$ matrix $\boldsymbol{\mu}$. Similarly, we have

$$\tilde{\boldsymbol{w}}(k) = \sum_{i=3}^{4} \boldsymbol{\mu}_{v,i}(k) + \sum_{j=4}^{6} \boldsymbol{\mu}_{s,j}(k).$$

We first look at the model $(\bar{\boldsymbol{w}}, \bar{\Phi})$ and consider the OOD accuracy of the samples from first class $k = 1$. For each sample from the first class in OOD, we have

$$\boldsymbol{x}\bar{\Phi}|_{\boldsymbol{y}=\boldsymbol{e}_1} = \sum_{i=1}^{2} \boldsymbol{\mu}_{v,i} \boldsymbol{Q}_{v,i}(1) + \sum_{j=1}^{3} \boldsymbol{\mu}_{s,j} \boldsymbol{Q}_{s,j}(1) + \sum_{i=1}^{5} \boldsymbol{z}_i$$

where $\boldsymbol{z}_i \sim \mathcal{N}(0, \sigma^2 \boldsymbol{I}_d), \forall i$. The model $(\bar{\boldsymbol{w}}, \bar{\Phi})$ makes correct prediction on the samples from $\boldsymbol{y} = \boldsymbol{e}_1$ if the following holds

$$\boldsymbol{w}(1)^\top \boldsymbol{x}\bar{\Phi}|_{\boldsymbol{y}=\boldsymbol{e}_1} > \boldsymbol{w}(2)^\top \boldsymbol{x}\bar{\Phi})|_{\boldsymbol{y}=\boldsymbol{e}_1}, \text{ and } \boldsymbol{w}(1)^\top \boldsymbol{x}\bar{\Phi}|_{\boldsymbol{y}=\boldsymbol{e}_1} > \boldsymbol{w}(3)^\top \boldsymbol{x}\bar{\Phi}|_{\boldsymbol{y}=\boldsymbol{e}_1}$$

So for each OOD sample, we have

$$\boldsymbol{w}(1)^\top \boldsymbol{x}\bar{\Phi})|_{\boldsymbol{y}=\boldsymbol{e}_1}$$

$$= \left( \sum_{i=1}^{2} \boldsymbol{\mu}_{v,i}(1) + \sum_{j=1}^{3} \boldsymbol{\mu}_{s,j}(1) \right)^\top \left( \sum_{i=1}^{2} \boldsymbol{\mu}_{v,i} \boldsymbol{Q}_{v,i}(1) + \sum_{j=1}^{3} \boldsymbol{\mu}_{s,j} \boldsymbol{Q}_{s,j}(1) + \sum_{i=1}^{5} \boldsymbol{z}_i \right),$$

$$= \sum_{i=1}^{2} \boldsymbol{\mu}_{v,i}(1)^\top (\boldsymbol{\mu}_{v,i} \boldsymbol{Q}_{v,i}(1)) + \sum_{j=1}^{3} \boldsymbol{\mu}_{s,j}(1)^\top (\boldsymbol{\mu}_{s,j} \boldsymbol{Q}_{s,j}(1)) + \xi$$

$$= 2 + \sum_{j=1}^{3} \underbrace{\boldsymbol{\mu}_{s,j}(1)^\top (\boldsymbol{\mu}_{s,j} \boldsymbol{Q}_{s,j}(1))}_{A_j} + \xi$$

where the second equality is by the Assumption 2 that different $\boldsymbol{\mu}$ are all orthogonal to each other; the last equality is because $\boldsymbol{Q}_{v,i}(1) = \boldsymbol{e}_1$ always hold and further by Assumption 2 we have

$$\boldsymbol{\mu}_{v,i}(1)^\top (\boldsymbol{\mu}_{v,i} \boldsymbol{Q}_{v,i}(1)) = \boldsymbol{\mu}_{v,i}(1)^\top (\boldsymbol{\mu}_{v,i} \boldsymbol{e}_1) = \boldsymbol{\mu}_{v,i}(1)^\top \boldsymbol{\mu}_{v,i}(1) = 1.$$

Similarly we have

$$\boldsymbol{w}(2)^\top \boldsymbol{x}\bar{\Phi}|_{\boldsymbol{y}=\boldsymbol{e}_1}$$

$$= \left( \sum_{i=1}^{2} \boldsymbol{\mu}_{v,i}(2) + \sum_{j=1}^{3} \boldsymbol{\mu}_{s,j}(2) \right)^\top \left( \sum_{i=1}^{2} \boldsymbol{\mu}_{v,i} \boldsymbol{Q}_{v,i}(1) + \sum_{j=1}^{3} \boldsymbol{\mu}_{s,j} \boldsymbol{Q}_{s,j}(1) + \sum_{i=1}^{5} \boldsymbol{z}_i \right),$$

$$= \sum_{i=1}^{2} \boldsymbol{\mu}_{v,i}(2)^\top (\boldsymbol{\mu}_{v,i} \boldsymbol{Q}_{v,i}(1)) + \sum_{j=1}^{3} \boldsymbol{\mu}_{s,j}(2)^\top (\boldsymbol{\mu}_{s,j} \boldsymbol{Q}_{s,j}(1)) + \xi$$

$$= 0 + \sum_{j=1}^{3} \underbrace{\boldsymbol{\mu}_{s,j}(2)^\top (\boldsymbol{\mu}_{s,j} \boldsymbol{Q}_{s,j}(1))}_{B_j} + \xi,$$

where the last equality is because $\boldsymbol{\mu}_{v,i}(2)^\top \left(\boldsymbol{\mu}_{v,i}\boldsymbol{Q}_{v,i}(1)\right) = \boldsymbol{\mu}_{v,i}(2)^\top \boldsymbol{\mu}_{v,i}(1) = 0$. Similarly, we also have

$$\boldsymbol{w}(3)^\top \boldsymbol{x}\bar{\Phi}|_{\boldsymbol{y}=\boldsymbol{e}_1} = \sum_{j=1}^3 \underbrace{\boldsymbol{\mu}_{s,j}(3)^\top \left(\boldsymbol{\mu}_{s,j}\boldsymbol{Q}_{s,j}(1)\right)}_{C_j} + \xi.$$

It is easy to see that, for $k_1, k_2 = 1, 2, 3$,

$$\boldsymbol{\mu}_{s,j}(k_1)^\top \left(\boldsymbol{\mu}_{s,j}\boldsymbol{e}_{k2}\right) = \begin{cases} 0, & \text{if } k_1 = k_2, \\ 1, & \text{otherwise.} \end{cases}$$

Since in the OOD distribution, $\boldsymbol{Q}_{s,j}(1)$ can be any of $\boldsymbol{e}_1$, $\boldsymbol{e}_2$ and $\boldsymbol{e}_3$, we have $A_j, B_j, C_j \in \{0, 1\}$ and $A_j + B_j + C_j = 1$ for $j = 1, 2, 3$. Specifically, $A_j = 1, B_j = 0, C_j = 0$ if $\boldsymbol{Q}_{s,j} = \boldsymbol{e}_1$, $A_j = 0, B_j = 1, C_j = 0$ if $\boldsymbol{Q}_{s,j} = \boldsymbol{e}_2$, and $A_j = 0, B_j = 0, C_j = 1$ if $\boldsymbol{Q}_{s,j} = \boldsymbol{e}_3$. We then have

$$\left(\boldsymbol{w}(1)^\top \boldsymbol{x}\bar{\Phi} - \boldsymbol{w}(2)^\top \boldsymbol{x}\bar{\Phi}\right)|_{\boldsymbol{y}=\boldsymbol{e}_1} = \begin{cases} -1, & \text{if } \sum_{j=1}^3 \mathbb{I}(\boldsymbol{Q}_{s,j}(1) = \boldsymbol{e}_2) = 3, \\ 0, & \text{if } \sum_{j=1}^3 \mathbb{I}(\boldsymbol{Q}_{s,j}(1) = \boldsymbol{e}_2) = 2 \text{ and } \sum_{j=1}^3 \mathbb{I}(\boldsymbol{Q}_{s,j}(1) = \boldsymbol{e}_3) = 1, \\ \geq 1 \text{ otherwise.} \end{cases}$$

Recall Definition 1, in the OOD distribution, we have

$$\boldsymbol{Q}_{s,j}(1) = \begin{cases} \boldsymbol{e}_1, & \text{with probability } 1 - \frac{2}{3}p, \\ \boldsymbol{e}_2, & \text{with probability } \frac{p}{3}, \\ \boldsymbol{e}_3, & \text{with probability } \frac{p}{3}. \end{cases}$$

Combing Lemma 3 with the results above we have

- $\mathcal{A}_{\text{ood}}(\bar{f}) \in [0, \epsilon]$ when $\sum_{j=1}^3 \mathbb{I}(\boldsymbol{Q}_{s,j}(1) = \boldsymbol{e}_2) = 3$ (equivalent to $\{\mathbb{I}(\boldsymbol{Q}_{s,j}(1) = \boldsymbol{e}_2)\}_{j=1}^3$ holds simultaneously) or $\sum_{j=1}^3 \mathbb{I}(\boldsymbol{Q}_{s,j}(1) = \boldsymbol{e}_3) = 3$, the probability is $2p^3/27$.

- $\mathcal{A}_{\text{ood}}(\bar{f}) \in [1/2 - \epsilon, 1/2 + \epsilon]$ when $\sum_{j=1}^3 \mathbb{I}(\boldsymbol{Q}_{s,j}(1) = \boldsymbol{e}_2) = 2$ and $\sum_{j=1}^3 \mathbb{I}(\boldsymbol{Q}_{s,j}(1) = \boldsymbol{e}_3) = 1$ or $(\sum_{j=1}^3 \mathbb{I}(\boldsymbol{Q}_{s,j}(1) = \boldsymbol{e}_3) = 2$ and $\sum_{j=1}^3 \mathbb{I}(\boldsymbol{Q}_{s,j}(1) = \boldsymbol{e}_2) = 1)$, the probability of which is $2 \cdot C_3^1 p^3/27 = 2p^3/9$.

- $\mathcal{A}_{\text{ood}}(\bar{f}) \in [1 - \epsilon, 1]$ otherwise, the probability of which is $1 - 8p^3/27$.

So the overall expected OOD acuracy is $\mathcal{A}_{\text{ood}}(\bar{f}) = (2p^3/9 \cdot 1/2 + (1 - 8p^3/27) \cdot 1) \pm \varepsilon \in [1 - 5p^3/27 - \varepsilon, 1 - 5p^3/27 + \varepsilon]$. We have $\mathcal{A}_{\text{ood}}(\tilde{f}) \in [1 - 5p^3/27 - \epsilon, 1 - 5p^3/27 + \epsilon]$ following the same proof.

(b) **Output space ensemble and weight space ensemble**.
Similar to the proof above, for weight space ensemble we have

$$\left(\bar{\boldsymbol{w}}(1) + \tilde{\boldsymbol{w}}(1)\right)^\top \boldsymbol{x}(\bar{\Phi} + \tilde{\Phi})|_{\boldsymbol{y}=\boldsymbol{e}_1}$$

$$= \left(\sum_{i=1}^4 \boldsymbol{\mu}_{v,i}(1) + \sum_{j=1}^6 \boldsymbol{\mu}_{s,j}(1)\right)^\top \left(\sum_{i=1}^4 \boldsymbol{\mu}_{v,i}\boldsymbol{Q}_{v,i}(1) + \sum_{j=1}^6 \boldsymbol{\mu}_{s,j}\boldsymbol{Q}_{s,j}(1) + \sum_{i=1}^5 \boldsymbol{z}_i\right),$$

$$= \sum_{i=1}^4 \boldsymbol{\mu}_{v,i}(1)^\top \left(\boldsymbol{\mu}_{v,i}\boldsymbol{Q}_{v,i}(1)\right) + \sum_{j=1}^6 \boldsymbol{\mu}_{s,j}(1)^\top \left(\boldsymbol{\mu}_{s,j}\boldsymbol{Q}_{s,j}(1)\right) + \xi$$

$$= 4 + \sum_{j=1}^6 \underbrace{\boldsymbol{\mu}_{s,j}(1)^\top \left(\boldsymbol{\mu}_{s,j}\boldsymbol{Q}_{s,j}(1)\right)}_{A_j} + \xi$$

We also have

$$(\bar{\boldsymbol{w}}(2) + \tilde{\boldsymbol{w}}(2))^{\top} \boldsymbol{x}(\bar{\Phi} + \tilde{\Phi})|_{\boldsymbol{y}=\boldsymbol{e}_1} = \sum_{j=1}^{6} \underbrace{\boldsymbol{\mu}_{s,j}(2)^{\top} (\boldsymbol{\mu}_{s,j} \boldsymbol{Q}_{s,j}(1))}_{B_j} + \xi,$$

$$(\bar{\boldsymbol{w}}(3) + \tilde{\boldsymbol{w}}(3))^{\top} \boldsymbol{x}(\bar{\Phi} + \tilde{\Phi})|_{\boldsymbol{y}=\boldsymbol{e}_1} = \sum_{j=1}^{6} \underbrace{\boldsymbol{\mu}_{s,j}(3)^{\top} (\boldsymbol{\mu}_{s,j} \boldsymbol{Q}_{s,j}(1))}_{C_j} + \xi$$

Then

$$\left(\boldsymbol{w}(1)^{\top} \boldsymbol{x}\Phi - \boldsymbol{w}(2)^{\top} \boldsymbol{x}\Phi\right)|_{\boldsymbol{y}=\boldsymbol{e}_1} = \begin{cases} -2, & \text{if } \sum_{j=1}^{6} \mathbb{I}(\boldsymbol{Q}_{s,j}(1) = \boldsymbol{e}_2) = 6, \\ -1, & \text{if } \sum_{j=1}^{6} \mathbb{I}(\boldsymbol{Q}_{s,j}(1) = \boldsymbol{e}_2) = 5 \text{ and if } \sum_{j=1}^{6} \mathbb{I}(\boldsymbol{Q}_{s,j}(1) = \boldsymbol{e}_3) = 1, \\ 0, & \text{if } \left(\sum_{j=1}^{6} \mathbb{I}(\boldsymbol{Q}_{s,j}(1) = \boldsymbol{e}_2) = 5 \text{ and } \sum_{j=1}^{6} \mathbb{I}(\boldsymbol{Q}_{s,j}(1) = \boldsymbol{e}_1) = 1\right) \text{ or }, \\ & \left(\sum_{j=1}^{6} \mathbb{I}(\boldsymbol{Q}_{s,j}(1) = \boldsymbol{e}_2) = 4 \text{ and } \sum_{j=1}^{6} \mathbb{I}(\boldsymbol{Q}_{s,j}(1) = \boldsymbol{e}_3) = 2\right). \\ \geq 1 & \text{otherwise.} \end{cases}$$

Then by Lemma 3, we have

$$\mathcal{A}_{\text{ood}}(f_{\text{wse}}) \in \begin{cases} [0, \epsilon], & \text{with probability } 2((p/3)^6 + 6 \cdot (p/3)^6) = 14p^6/729, \\ [\frac{1}{2} - \epsilon, \frac{1}{2} + \epsilon], & \text{with probability} \\ & 2(6 \cdot (p/3)^5 \cdot (1 - 2p/3) + 6C2 \cdot (p/3)^6) = 2p^6/243 + 4p^5/81 \\ [1 - \epsilon, 1], & \text{with probability } 1 - 20p^6/729 - 4p^5/81. \end{cases}$$

Then the overall expected OOD accuracy $\mathcal{A}_{\text{ood}}(f_{\text{wse}})$ is in
$[1 - 2p^5/81 - 17p^6/729 - \varepsilon, 1 - 2p^5/81 - 17p^6/729 + \varepsilon]$.

The accuracy of the model ensemble and model averaging are the same in Example 1 since

$$(\bar{\boldsymbol{w}}(k) + \tilde{\boldsymbol{w}}(k))^{\top} \boldsymbol{x}(\bar{\Phi} + \tilde{\Phi})|_{\boldsymbol{y}=\boldsymbol{e}_1} = \sum_{i=1}^{4} \boldsymbol{\mu}_{v,i}(k)^{\top} (\boldsymbol{\mu}_{v,i} \boldsymbol{Q}_{v,i}(1)) + \sum_{j=1}^{6} \boldsymbol{\mu}_{s,j}(k)^{\top} (\boldsymbol{\mu}_{s,j} \boldsymbol{Q}_{s,j}(1)) + \xi$$

and

$$\bar{\boldsymbol{w}}(k)^{\top} (\boldsymbol{x}\bar{\Phi}) + \tilde{\boldsymbol{w}}(k)^{\top} (\boldsymbol{x}\tilde{\Phi})|_{\boldsymbol{y}=\boldsymbol{e}_1} = \sum_{i=1}^{4} \boldsymbol{\mu}_{v,i}(k)^{\top} (\boldsymbol{\mu}_{v,i} \boldsymbol{Q}_{v,i}(1)) + \sum_{j=1}^{6} \boldsymbol{\mu}_{s,j}(k)^{\top} (\boldsymbol{\mu}_{s,j} \boldsymbol{Q}_{s,j}(1)) + \xi.$$

$\square$

## F.2 PROOF OF PROPOSITION 2

Before starting the proof process, we restate Proposition 2 and Definition 1 for $K$ ($K \geq 3$) class situation as follows:

**Definition 5** (Data Generation Process). *The whole data generation process are as follows:*

$$\boldsymbol{y} \sim Unif\{\boldsymbol{e}_1, \boldsymbol{e}_2, \dots, \boldsymbol{e}_K\}, \boldsymbol{x} = Concat\left(\{\boldsymbol{x}_{v,i}\}_{i=1}^{d_v} \cup \{\boldsymbol{x}_{s,j}\}_{j=1}^{d_s}\right),$$

$$\mathbb{P}_{\theta}(\boldsymbol{x}_{v,i} \mid \boldsymbol{y}) = \mathcal{N}\left(\boldsymbol{\mu}_{v,i} \boldsymbol{Q}_{v,i} \boldsymbol{y}, \sigma^2 \boldsymbol{I}_d\right), \mathbb{P}_{\theta}(\boldsymbol{x}_{s,j} \mid \boldsymbol{y}) = \mathcal{N}\left(\boldsymbol{\mu}_{s,j} \boldsymbol{Q}_{s,j} \boldsymbol{y}, \sigma^2 \boldsymbol{I}_d\right), \forall i, j. \quad (8)$$

*where* $\boldsymbol{Q}_{v,i}, \boldsymbol{Q}_{s,j} \in \{0, 1\}^{K \times K}$. *Further,* $\boldsymbol{Q}_{v,i} = \boldsymbol{I}_3 = [\boldsymbol{e}_1, \boldsymbol{e}_2, \dots, \boldsymbol{e}_K]$ *always hold. In the ID distribution* $\mathcal{D}_{id}$, $\boldsymbol{Q}_{s,j} = \boldsymbol{I}_K$; *and in OOD* $\mathcal{D}_{ood}$, *the kth column of* $\boldsymbol{Q}$, *i.e.,* $\boldsymbol{Q}_{s,j}(k)$, *is as follows for* $k = 1, 2, \dots, K$:

$$\boldsymbol{Q}_{s,j}(k) = \begin{cases} \boldsymbol{e}_k, & \text{with probability } 1 - p \\ Unif\{\boldsymbol{e}_1, \boldsymbol{e}_2, \dots, \boldsymbol{e}_K\}, & \text{with probability } p. \end{cases}$$

**Proposition 7** (General Results for OSE). *Consider Definition 1-3, Assumption 1-2 hold, and infinite ID and OOD samples. Omitting small constants involving $\epsilon$, we have*

$$\mathcal{A}_{ood}(\bar{f}) = F_p\left(\frac{(1-p)\bar{n}_s + \bar{n}_v}{\sqrt{\bar{n}_s}}\right),$$

$$\mathcal{A}_{ood}(\tilde{f}) = F_p\left(\frac{(1-p)\tilde{n}_s + \tilde{n}_v}{\sqrt{\tilde{n}_s}}\right),$$

$$\mathcal{A}_{ood}(fose) = F_p\left(\frac{(1-p)(\tilde{n}_s + \bar{n}_s) + (\tilde{n}_v + \bar{n}_v)}{\sqrt{\tilde{n}_s + \bar{n}_s + 2n_{so}}}\right).$$

In the proof process, here we take a notation first:

$$\boldsymbol{L}(t_1, \ldots, t_{K-1}) = \mathbb{P}_{\boldsymbol{z} \sim \mathcal{N}(\boldsymbol{0}, \sigma^2 \boldsymbol{I}_{K-1})}\left(\boldsymbol{a}_i^T \boldsymbol{z} + t_i > 0, \forall i = 1, \ldots, K-1\right)$$

in which

$$\| \boldsymbol{a}_i \|_2^2 = 2, \quad \boldsymbol{a}_i^T \boldsymbol{a}_j = 1,$$

for any $i \neq j$.

Consider the extracted features in both models as

$$\{\boldsymbol{x}_{v,\bar{i}}\}_{\bar{i}=1}^{\bar{n}_v - n_{vo}} \cup \{\boldsymbol{x}_{s,\bar{j}}\}_{\bar{j}=1}^{\bar{n}_s - n_{so}} \cup \{\boldsymbol{x}_{v,\tilde{i}}\}_{\tilde{i}=1}^{\tilde{n}_v - n_{vo}} \cup \{\boldsymbol{x}_{s,\tilde{i}}\}_{\tilde{i}=1}^{\tilde{n}_s - n_{so}} \cup \{\boldsymbol{x}_{v,i}\}_{i=1}^{n_{vo}} \cup \{\boldsymbol{x}_{s,i}\}_{i=1}^{n_{so}},$$

in which $n_{vo}, n_{so}$ are the numbers of overlapped invariant features and spurious features respectively.

Then for each single model, we have

$$\boldsymbol{x}\bar{\Phi} = \sum_{\bar{i}=1}^{\bar{n}_v - n_{vo}} \boldsymbol{x}_{v,\bar{i}} + \sum_{\bar{j}=1}^{\bar{n}_s - n_{so}} \boldsymbol{x}_{s,\bar{j}} + \sum_{i=1}^{n_{vo}} \boldsymbol{x}_{v,i} + \sum_{i=1}^{n_{so}} \boldsymbol{x}_{s,i},$$

$$\bar{\boldsymbol{w}}(k) = \frac{\sum_{\bar{i}=1}^{\bar{n}_v - n_{vo}} \boldsymbol{\mu}_{v,\bar{i}}(k) + \sum_{\bar{j}=1}^{\bar{n}_s - n_{so}} \boldsymbol{\mu}_{s,\bar{j}}(k) + \sum_{i=1}^{n_{vo}} \boldsymbol{\mu}_{v,i}(k) + \sum_{i=1}^{n_{so}} \boldsymbol{\mu}_{s,i}(k)}{\sqrt{\bar{n}_v + \bar{n}_s}}, \quad k = 1, \ldots, K$$

$$\boldsymbol{x}\tilde{\Phi} = \sum_{\tilde{i}=1}^{\tilde{n}_v - n_{vo}} \boldsymbol{x}_{v,\tilde{i}} + \sum_{\tilde{i}=1}^{\tilde{n}_s - n_{so}} \boldsymbol{x}_{s,\tilde{i}} + \sum_{i=1}^{n_{vo}} \boldsymbol{x}_{v,i} + \sum_{i=1}^{n_{so}} \boldsymbol{x}_{s,i},$$

$$\tilde{\boldsymbol{w}}(k) = \frac{\sum_{\tilde{i}=1}^{\tilde{n}_v - n_{vo}} \boldsymbol{\mu}_{v,\tilde{i}}(k) + \sum_{\tilde{i}=1}^{\tilde{n}_s - n_{so}} \boldsymbol{\mu}_{s,\tilde{i}}(k) + \sum_{i=1}^{n_{vo}} \boldsymbol{\mu}_{v,i}(k) + \sum_{i=1}^{n_{so}} \boldsymbol{\mu}_{s,i}(k)}{\sqrt{\tilde{n}_v + \tilde{n}_s}}, \quad k = 1, \ldots, K.$$

Then we can analysis the forecasting accuracy for both averaging model and ensemble model respectively.

### F.2.1 PROOF FOR SINGLE MODEL

Considering the extracted features in Algorithm 1 as

$$\{\boldsymbol{x}_{v,\bar{i}}\}_{\bar{i}=1}^{\bar{n}_v} \cup \{\boldsymbol{x}_{s,\bar{j}}\}_{\bar{j}=1}^{\bar{n}_s},$$

for convenience, we denote

$$\bar{\boldsymbol{x}} := \boldsymbol{x}\bar{\Phi} = \sum_{\bar{i}=1}^{\bar{n}_v} \boldsymbol{x}_{v,\bar{i}} + \sum_{\bar{j}=1}^{\bar{n}_s} \boldsymbol{x}_{s,\bar{j}},$$

then according to Lemma 5, we can obtain the estimated classifier on label $\boldsymbol{e}_k$:

$$\bar{\boldsymbol{w}}(k) = \sum_{\bar{i}=1}^{\bar{n}_v} \frac{1}{\sqrt{\bar{n}_v + \bar{n}_s}} \boldsymbol{\mu}_{v,\bar{i}} + \sum_{\bar{j}=1}^{\bar{n}_s} \frac{1}{\sqrt{\bar{n}_v + \bar{n}_s}} \boldsymbol{\mu}_{s,\bar{j}}.$$

Based on this classifier, the forecasting accuracy on ID case is

$$\mathbb{P}(\hat{y} = y) = \frac{1}{K} \sum_{k=1}^{K} \mathbb{E}_{\boldsymbol{x}|y=e_k} \left\{ \mathbf{1}(\bar{\boldsymbol{x}}^\top \bar{\boldsymbol{w}}(k) > \bar{\boldsymbol{x}}^\top \bar{\boldsymbol{w}}(k'), \forall k' \neq k) \right\}$$

$$= \frac{1}{K} \sum_{k=1}^{K} \mathbb{P}_{\boldsymbol{z} \sim \mathcal{N}(\mathbf{0}, (\bar{n}_v + \bar{n}_s)\sigma^2 \boldsymbol{I}_d)} \left( (\bar{\boldsymbol{w}}(k) - \bar{\boldsymbol{w}}(k'))^T \boldsymbol{z} + (\bar{\boldsymbol{w}}(k) - \bar{\boldsymbol{w}}(k'))^T \mathbb{E}(\bar{\boldsymbol{x}} \mid \boldsymbol{y} = \boldsymbol{e}_k) > 0, \forall k' \neq k \right)$$

$$= \frac{1}{K} \sum_{k=1}^{K} \mathbb{P}_{\boldsymbol{z} \sim \mathcal{N}(\mathbf{0}, \sigma^2 \boldsymbol{I}_d)} \left( (\bar{\boldsymbol{w}}(k) - \bar{\boldsymbol{w}}(k'))^T \boldsymbol{z} + \bar{\delta}_{k,k'} > 0, \forall k' \neq k \right),$$

in which we denote that

$$\bar{\delta}_{k,k'} = \frac{1}{\bar{n}_v + \bar{n}_s} \left( \sum_{\bar{i}=1}^{\bar{n}_v} 1 + \sum_{\bar{j}=1}^{\bar{n}_s} 1 \right) = 1,$$

for any $k' \neq k$. And considering Assumption 2, we have

$$(\bar{\boldsymbol{w}}(k) - \bar{\boldsymbol{w}}(k'))^T (\bar{\boldsymbol{w}}(k) - \bar{\boldsymbol{w}}(k'')) = 1, \quad \| \bar{\boldsymbol{w}}(k) - \bar{\boldsymbol{w}}(k') \|_2^2 = 2,$$

for any $k \neq k' \neq k''$. Then with Lemma 2, the IID forecasting accuracy can be expressed as

$$\mathbb{P}(\boldsymbol{y} = \hat{\boldsymbol{y}}) = \boldsymbol{L}(1, \dots, 1),$$

which can not be influenced by $\bar{n}_v, \bar{n}_s$.

Then we turn to the OOD forecasting accuracy. For class $k$, we suppose there are $r_k$ spurious features maintaining their parameters, and $r_{k \to k'}$ refer to the number of spurious features flipping to the class $k'$, the corresponding probability is

$$\mathbb{P}([r_k, [r_{k \to k'}, \forall k' \neq k]]) = \frac{\bar{n}_s!}{r_k! \Pi_{k' \neq k} r_{k \to k'}!} (1 - p + \frac{p}{K})^{r_k} (\frac{K-1}{K} p)^{\bar{n}_s - r_k},$$

and the conditional OOD forecasting accuracy on label $\boldsymbol{e}_k$ is

$$\mathbb{P}(\hat{\boldsymbol{y}} = \boldsymbol{e}_k \mid [r_k, [r_{k \to k'}, \forall k' \neq k]], \boldsymbol{y} = \boldsymbol{e}_k)$$

$$= \mathbb{E}_{\bar{\boldsymbol{x}}|[r_k, [r_{k \to k'}, \forall k' \neq k]], \boldsymbol{y} = \boldsymbol{e}_k} \left\{ \mathbf{1}(\bar{\boldsymbol{x}}^T \bar{\boldsymbol{w}}(k) > \boldsymbol{x}^T \bar{\boldsymbol{w}}(k'), \forall k' \neq k) \right\}$$

$$= \mathbb{P}_{\boldsymbol{z} \sim \mathcal{N}(\mathbf{0}, (\bar{n}_v + \bar{n}_s)\sigma^2 \boldsymbol{I}_d)} \left( (\bar{\boldsymbol{w}}(k) - \bar{\boldsymbol{w}}(k'))^T \boldsymbol{z} + \frac{\bar{n}_v + r_k - r_{k \to k'}}{\sqrt{\bar{n}_v + \bar{n}_s}}, \forall k' \neq k \right)$$

$$= \boldsymbol{L}(\frac{\bar{n}_v + r_k - r_{k \to k'}}{\bar{n}_v + \bar{n}_s}, \forall k' \neq k),$$

according to this, the OOD forecasting accuracy can be expressed as

$$\mathbb{P}(\hat{\boldsymbol{y}} = \boldsymbol{y}) = \mathbb{E}_{\boldsymbol{y}}[\mathbb{P}(\hat{\boldsymbol{y}} = \boldsymbol{y} \mid \boldsymbol{y})] = \mathbb{P}(\hat{\boldsymbol{y}} = \boldsymbol{e}_1 \mid \boldsymbol{y} = \boldsymbol{e}_1)$$

$$= \sum_{r_1, r_{1 \to k'}, \forall k' \geq 2} \mathbb{P}([r_1, [r_{1 \to k'}, \forall k' \neq 1]]) \boldsymbol{L}(\frac{\bar{n}_v + r_1 - r_{1 \to k'}}{\bar{n}_v + \bar{n}_s}, \forall k' \geq 1).$$

with Lemma 3, we can get related properties about $\boldsymbol{L}(\cdot)$, then take upper and lower bounds respectively.

Considering the close form of $G(\cdot)$ in equation 13, we denote $n_v = \bar{n}_v, n_s = \bar{n}_s, n_{vo} = n_{so} = 0, C = 0$, then the OOD forecasting accuracy can be lower bounded as

$$\mathbb{P}(\hat{\boldsymbol{y}} = \boldsymbol{y}) \geq \mathbb{P}(\mathcal{A})(1 - \epsilon) + \sum_{N=1}^{K-1} \mathbb{P}(\mathcal{C}(N))(h(N) - \epsilon)$$

$$\geq \mathbb{P}(\mathcal{A}) + \sum_{N=1}^{K-1} \mathbb{P}(\mathcal{C}(N))h(N) - \epsilon = \boldsymbol{G}(\bar{n}_v, \bar{n}_s, 0, 0, 0) - \epsilon,$$

and on the other hand, it can also be upper bounded by

$$\mathbb{P}(\hat{\boldsymbol{y}} = \boldsymbol{y}) \leq \mathbb{P}(\mathcal{A}) + \sum_{N=1}^{K-1} \mathbb{P}(\mathcal{C}(N))h(N) + \epsilon \mathbb{P}(\mathcal{B}) \leq \boldsymbol{G}(\bar{n}_v, \bar{n}_s, 0, 0, 0) + \epsilon,$$

Similar with Algorithm 1, we can also get the ID and OOD forecasting accuracy in Algorithm 2, which is related to another $\tilde{n}_v$ invariant features and $\tilde{n}_s$ spurious features.

For the OOD forecasting accuracy, we'd like to take some intuitive approximation for $G(n_v, n_s, 0, 0, 0)$. As the number $n_v, n_s$ are large enough, we can take approximation by multivariate Gaussian distribution. To be specific, we denote $\boldsymbol{r} = [r_1, r_{1\to 2}, \ldots, r_{1\to K}]$, then can regard them as $\boldsymbol{r} \sim \mathcal{N}(\boldsymbol{\gamma}, \boldsymbol{\Sigma})$, in which

$$\boldsymbol{\gamma} = [n_s(1 - p + p/K), n_s p/K, \ldots, n_s p/K]^T,$$

$$\boldsymbol{\Sigma}_{i,i} = \frac{\gamma_i(n_s - \gamma_i)}{n_s}, \quad \boldsymbol{\Sigma}_{i,j} = \frac{-\gamma_i \gamma_j}{n_s}.$$

If we denote a new $(K - 1) \times K$ matrix as

$$\boldsymbol{T} = \begin{pmatrix} 1 & -1 & 0 & \ldots & 0 \\ 1 & 0 & -1 & \ldots & 0 \\ \vdots & \ddots & \ddots & \ldots & 0 \\ 1 & 0 & 0 & \ldots & -1 \end{pmatrix}$$

And the new $(K - 1)$-dim random variable, i.e,

$$\boldsymbol{\eta} \doteq \boldsymbol{T}^T \boldsymbol{r} + n_v \mathbf{1}$$

is still Gaussian, to be specific, if we denote its distribution as $\boldsymbol{\eta} \sim \mathcal{N}(\boldsymbol{\alpha}, \boldsymbol{M})$, then we have

$$\boldsymbol{\alpha} = (n_s(1 - p) + n_v)\mathbf{1},$$

$$\boldsymbol{M}_{i,i} = n_s \frac{p(K + 2 - pK)}{K},$$

$$\boldsymbol{M}_{i,j} = n_s \frac{p(K + 1 - pK)}{K},$$

$\boldsymbol{G}(n_v, n_s, 0, 0, 0)$ can be approximated as

$$\mathbb{P}(\boldsymbol{\eta}_1 > 0, \ldots, \boldsymbol{\eta}_{K-1} > 0),$$

which is equal to $F_p\left((n_s(1 - p) + n_v)/\sqrt{n_s}\right)$, and $F_p(\cdot)$ is defined in Appendix F.2.5.

### F.2.2 PROOF FOR WEIGHT SPACE ENSEMBLE

For averaging model, we denote

$$\hat{\boldsymbol{x}} := \frac{1}{2}\boldsymbol{x}(\bar{\Phi} + \tilde{\Phi}) = \frac{1}{2}\sum_{\bar{i}=1}^{\bar{n}_v - n_{vo}} \boldsymbol{x}_{v,\bar{i}} + \frac{1}{2}\sum_{\bar{j}=1}^{\bar{n}_s - n_{so}} \boldsymbol{x}_{s,\bar{j}} + \frac{1}{2}\sum_{\tilde{i}=1}^{\tilde{n}_v - n_{vo}} \boldsymbol{x}_{v,\tilde{i}} + \frac{1}{2}\sum_{\tilde{i}=1}^{\tilde{n}_s - n_{so}} \boldsymbol{x}_{s,\tilde{i}} + \sum_{i=1}^{n_{vo}} \boldsymbol{x}_{v,i} + \sum_{i=1}^{n_{so}} \boldsymbol{x}_{s,i},$$

then after scaling, the averaging classifier on label $\boldsymbol{e}_k$:

$$\hat{\boldsymbol{w}} := \frac{1}{2}(\bar{\boldsymbol{w}} + \tilde{\boldsymbol{w}})$$

$$= \frac{\sum_{\bar{i}=1}^{\bar{n}_v - n_{vo}} \boldsymbol{\mu}_{v,\bar{i}}(k) + \sum_{\bar{j}=1}^{\bar{n}_s - n_{so}} \boldsymbol{\mu}_{s,\bar{j}}(k) + \sum_{\tilde{i}=1}^{\tilde{n}_v - n_{vo}} \boldsymbol{\mu}_{v,\tilde{i}}(k) + \sum_{\tilde{i}=1}^{\tilde{n}_s - n_{so}} \boldsymbol{\mu}_{s,\tilde{i}}(k) + 2\sum_{i=1}^{n_{vo}} \boldsymbol{\mu}_{v,i}(k) + 2\sum_{i=1}^{n_{so}} \boldsymbol{\mu}_{s,i}(k)}{\sqrt{\bar{n}_v + \bar{n}_s + \tilde{n}_v + \tilde{n}_s + 2n_{vo} + 2n_{so}}}.$$

Based on this classifier, if we denote $\hat{n} = (\bar{n}_v + \bar{n}_s + \tilde{n}_v + \tilde{n}_s + 2n_{vo} + 2n_{so})/4$, the forecasting accuracy on ID case is

$$\mathbb{P}(\hat{y} = y) = \frac{1}{K}\sum_{k=1}^{K} \mathbb{E}_{\boldsymbol{x}|y=e_k}\{\mathbf{1}(\hat{\boldsymbol{x}}^\top \hat{\boldsymbol{w}}(k) > \hat{\boldsymbol{x}}^\top \hat{\boldsymbol{w}}(k')), \forall k' \neq k)\}$$

$$= \frac{1}{K}\sum_{k=1}^{K} \mathbb{P}_{\boldsymbol{z}\sim\mathcal{N}(\mathbf{0},\hat{n}\sigma^2 \boldsymbol{I}_d)}\left((\hat{\boldsymbol{w}}(k) - \hat{\boldsymbol{w}}(k'))^T \boldsymbol{z} + (\hat{\boldsymbol{w}}(k) - \hat{\boldsymbol{w}}(k'))^T \mathbb{E}(\hat{\boldsymbol{x}} \mid \boldsymbol{y} = \boldsymbol{e}_k) > 0, \forall k' \neq k\right)$$

$$= \frac{1}{K}\sum_{k=1}^{K} \mathbb{P}_{\boldsymbol{z}\sim\mathcal{N}(\mathbf{0},\sigma^2 \boldsymbol{I}_d)}\left((\hat{\boldsymbol{w}}(k) - \hat{\boldsymbol{w}}(k'))^T \boldsymbol{z} + \hat{\delta}_{k,k'} > 0, \forall k' \neq k\right),$$

in which we denote that

$$\hat{\delta}_{k,k'} = \frac{\sum_{\bar{i}=1}^{\bar{n}_v - n_{vo}} 1 + \sum_{\bar{j}=1}^{\bar{n}_s - n_{so}} 1 + \sum_{\tilde{i}=1}^{\tilde{n}_v - n_{vo}} 1 + \sum_{\tilde{j}=1}^{\tilde{n}_s - n_{so}} 1 + \sum_{i=1}^{n_{vo}} 4 + \sum_{i=1}^{n_{so}} 4}{\bar{n}_v + \bar{n}_s + \tilde{n}_v + \tilde{n}_s + 2n_{vo} + 2n_{so}} = 1,$$

for any $k' \neq k$. And considering Assumption 2, we have

$$(\hat{\boldsymbol{w}}(k) - \hat{\boldsymbol{w}}(k'))^T (\hat{\boldsymbol{w}}(k) - \hat{\boldsymbol{w}}(k')) = 1, \quad \| \hat{\boldsymbol{w}}(k) - \hat{\boldsymbol{w}}(k') \|_2^2 = 2,$$

for any $k \neq k' \neq k''$. Then with Lemma 2, the IID forecasting accuracy can be expressed as

$$\mathbb{P}(\boldsymbol{y} = \boldsymbol{y}_{\text{wse}}) = \boldsymbol{L}(1, \ldots, 1) \in [1 - \epsilon, 1],$$

which can not be influenced by $\bar{n}_v, \bar{n}_s, \tilde{n}_v, \tilde{n}_s, n_{vo}, n_{so}$.

Then we turn to the OOD forecasting accuracy and for each $k = 1, \ldots, K$, we take some notations as follows:

$$\begin{aligned}
\bar{r}_k &= |\{\mathbb{I}(\boldsymbol{\mu}_{s,i}(k) = \boldsymbol{\mu}_{s,i}(k))\}_{i=1}^{\bar{n}_s} - n_{so}| \\
\tilde{r}_k &= |\{\mathbb{I}(\boldsymbol{\mu}_{s,i}(k) = \boldsymbol{\mu}_{s,i}(k))\}_{i=1}^{\tilde{n}_s - n_{so}}| \\
r_k^o &= |\{\mathbb{I}(\boldsymbol{\mu}_{s,i}(k) = \boldsymbol{\mu}_{s,i}(k))\}_{i=1}^{n_{so}}| \\
\bar{r}_{k \to k'} &= |\{\mathbb{I}(\boldsymbol{\mu}_{s,i}(k) = \boldsymbol{\mu}_{s,i}(k'))\}_{i=1}^{\bar{n}_s} - n_{so}| \\
\tilde{r}_{k \to k'} &= |\{\mathbb{I}(\boldsymbol{\mu}_{s,i}(k) = \boldsymbol{\mu}_{s,i}(k'))\}_{i=1}^{\tilde{n}_s - n_{so}}| \\
r_{k \to k'}^o &= |\{\mathbb{I}(\boldsymbol{\mu}_{s,i}(k) = \boldsymbol{\mu}_{s,i}(k'))\}_{i=1}^{n_{so}}|.
\end{aligned} \tag{9}$$

To be specific, for class $k$, we suppose there are $\bar{r}_k, \tilde{r}_k$ spurious features (no overlapped) maintaining their parameters, related to Algorithm 1, 2, and correspondingly, $\bar{r}_{k \to k'}, \tilde{r}_{k \to k'}$ refer to the number of spurious features flipping to the class $k'$, and $r_k^o, r_{k \to k'}^o$ are defined similar in overlapped spurious features. Then denoting $R_k(r) = [\bar{r}_k, \tilde{r}_k, r_k^o, [\bar{r}_{k \to k'}, \tilde{r}_{k \to k'}, r_{k \to k'}^o, \forall k' \neq k]]$, we obtain the corresponding probability as

$$\mathbb{P}(R_k(r)) = \frac{(\bar{n}_s + \tilde{n}_s - 2n_{so})! n_{so}!}{(\bar{r}_k + \tilde{r}_k)! r_k^o! \Pi_{k' \neq k}(\bar{r}_{k \to k'} + \tilde{r}_{k \to k'})! r_{k \to k'}^o!} (1 - p + \frac{p}{K})^{\bar{r}_k + \tilde{r}_k + r_k^o} \left(\frac{K-1}{K} p\right)^{\bar{n}_s + \tilde{n}_s - n_{so} - \bar{r}_k - \tilde{r}_k - r_k^o},$$

and the conditional OOD forecasting accuracy on label $\boldsymbol{e}_k$ is

$$\begin{aligned}
&\mathbb{P}(\hat{\boldsymbol{y}} = \boldsymbol{e}_k \mid R_k(r), \boldsymbol{y} = \boldsymbol{e}_k) \\
&= \mathbb{E}_{\hat{\boldsymbol{x}} \mid R_k(r), \boldsymbol{y} = \boldsymbol{e}_k} \left\{ \mathbf{1}(\hat{\boldsymbol{x}}^T \hat{\boldsymbol{w}}(k) > \hat{\boldsymbol{x}}^T \hat{\boldsymbol{w}}(k'), \forall k' \neq k) \right\} \\
&= \mathbb{P}_{\boldsymbol{z} \sim \mathcal{N}(\boldsymbol{0}, \hat{n}\sigma^2 \boldsymbol{I}_d)} \left( (\hat{\boldsymbol{w}}(k) - \hat{\boldsymbol{w}}(k'))^T \boldsymbol{z} + \frac{\bar{n}_v + \tilde{n}_v + 2n_{vo} + \bar{r}_k + \tilde{r}_k + 4r_k^o - \bar{r}_{k \to k'} - \tilde{r}_{k \to k'} - 4r_{k \to k'}^o}{\sqrt{\bar{n}_v + \bar{n}_s + \tilde{n}_v + \tilde{n}_s + 2n_{vo} + 2n_{so}}}, \forall k' \neq k \right) \\
&= \boldsymbol{L}\left( \frac{\bar{n}_v + \tilde{n}_v + 2n_{vo} + \bar{r}_k + \tilde{r}_k + 4r_k^o - \bar{r}_{k \to k'} - \tilde{r}_{k \to k'} - 4r_{k \to k'}^o}{\bar{n}_v + \bar{n}_s + \tilde{n}_v + \tilde{n}_s + 2n_{vo} + 2n_{so}}, \forall k' \neq k \right),
\end{aligned}$$

according to this, the OOD forecasting accuracy can be expressed as

$$\begin{aligned}
\mathbb{P}(\hat{\boldsymbol{y}} = \boldsymbol{y}) &= \mathbb{E}_{\boldsymbol{y}}[\mathbb{P}(\hat{\boldsymbol{y}} = \boldsymbol{y} \mid \boldsymbol{y})] = \mathbb{P}(\hat{\boldsymbol{y}} = \boldsymbol{e}_1 \mid \boldsymbol{y} = \boldsymbol{e}_1) \\
&= \sum_{\hat{r}_1} \mathbb{P}(R_1(r)) \boldsymbol{L}\left( \frac{\bar{n}_v + \tilde{n}_v + 2n_{vo} + \bar{r}_1 + \tilde{r}_1 + 4r_1^o - \bar{r}_{1 \to k'} - \tilde{r}_{1 \to k'} - 4r_{1 \to k'}^o}{\bar{n}_v + \bar{n}_s + \tilde{n}_v + \tilde{n}_s + 2n_{vo} + 2n_{so}}, \forall k' \neq k \right).
\end{aligned}$$

with Lemma 3, we can get related properties about $\boldsymbol{L}(\cdot)$, then take upper and lower bounds respectively. Still recalling the expression in equation 13 with $n_v = \bar{n}_v + \tilde{n}_v, n_s = \bar{n}_s + \tilde{n}_s, n_{vo} = n_{vo}, n_{so} = n_{so}, C = 4$, we can obtain the lower bound for OOD forecasting accuracy as

$$\mathbb{P}(\hat{\boldsymbol{y}} = \boldsymbol{y}) \geq \mathbb{P}(\mathcal{A})(1 - \epsilon) + \sum_{N=1}^{K-1} \mathbb{P}(\mathcal{C}(N))(h(N) - \epsilon)$$

$$\geq \mathbb{P}(\mathcal{A}) + \sum_{N=1}^{K-1} \mathbb{P}(\mathcal{C}(N)) h(N) - \epsilon = \boldsymbol{G}(\bar{n}_v + \tilde{n}_v, \bar{n}_s + \tilde{n}_s, n_{vo}, n_{so}, 4) - \epsilon,$$

and on the other hand, it can be upper bounded by

$$\mathbb{P}(\hat{\boldsymbol{y}} = \boldsymbol{y}) \leq \mathbb{P}(\mathcal{A}) + \sum_{N=1}^{K-1} \mathbb{P}(\mathcal{C}(N)) h(N) + \epsilon \mathbb{P}(\mathcal{B}) \leq \boldsymbol{G}(\bar{n}_v + \tilde{n}_v, \bar{n}_s + \tilde{n}_s, n_{vo}, n_{so}, 4) + \epsilon,$$

Similar to the analysis before, for ID forecasting accuracy, we have

$$\mathcal{J}_{id} = 0 \leq 3\epsilon,$$

and for OOD forecasting accuracy, we can draw a conclusion that

$$\mathcal{J}_{ood} \geq \boldsymbol{G}(\bar{n}_v + \tilde{n}_v, \bar{n}_s + \tilde{n}_s, n_{vo}, n_{so}, 4) - \max\{\boldsymbol{G}(\bar{n}_v, \bar{n}_s, 0, 0, 0), \boldsymbol{G}(\tilde{n}_v, \tilde{n}_s, 0, 0, 0)\} - 3\epsilon.$$

Similar to the analysis above, we'd like to take some intuitive approximation for OOD forecasting accuracy in the ensemble model. As the number $\bar{n}_v, \bar{n}_s, \tilde{n}_v, \tilde{n}_s, n_{vo}, n_{so}$ are large enough, we can take approximation by multivariate Gaussian distribution. To be specific, we denote $\bar{\boldsymbol{r}} = [\bar{r}_1, \bar{r}_{1\rightarrow 2}, \ldots, \bar{r}_{1\rightarrow K}]$, $\tilde{\boldsymbol{r}} = [\tilde{r}_1, \tilde{r}_{1\rightarrow 2}, \ldots, \tilde{r}_{1\rightarrow K}]$ and $\boldsymbol{r}^o = [r_1^o, r_{1\rightarrow 2}^o, \ldots, r_{1\rightarrow K}^o]$, then can regard them as $\bar{\boldsymbol{r}} \sim \mathcal{N}(\bar{\boldsymbol{\gamma}}, \bar{\boldsymbol{\Sigma}})$, $\tilde{\boldsymbol{r}} \sim \mathcal{N}(\tilde{\boldsymbol{\gamma}}, \tilde{\boldsymbol{\Sigma}})$ and $\boldsymbol{r}^o \sim \mathcal{N}(\boldsymbol{\gamma}^o, \boldsymbol{\Sigma}^o)$ (they are independent), in which

$$\bar{\boldsymbol{\gamma}} = [(\bar{n}_s - n_{so})(1 - p + p/K), (\bar{n}_s - n_{so})p/K, \ldots, (\bar{n}_s - n_{so})p/K]^T,$$

$$\bar{\boldsymbol{\Sigma}}_{i,i} = \frac{\bar{\gamma}_i(\bar{n}_s - n_{so} - \bar{\gamma}_i)}{\bar{n}_s - n_{so}}, \quad \bar{\boldsymbol{\Sigma}}_{i,j} = \frac{-\bar{\gamma}_i \bar{\gamma}_j}{\bar{n}_s - n_{so}},$$

$$\tilde{\boldsymbol{\gamma}} = [(\tilde{n}_s - n_{so})(1 - p + p/K), (\tilde{n}_s - n_{so})p/K, \ldots, (\tilde{n}_s - n_{so})p/K]^T,$$

$$\tilde{\boldsymbol{\Sigma}}_{i,i} = \frac{\tilde{\gamma}_i(\tilde{n}_s - n_{so} - \tilde{\gamma}_i)}{\tilde{n}_s - n_{so}}, \quad \tilde{\boldsymbol{\Sigma}}_{i,j} = \frac{-\tilde{\gamma}_i \tilde{\gamma}_j}{\tilde{n}_s - n_{so}},$$

$$\boldsymbol{\gamma}^o = [n_{so}(1 - p + p/K), n_{so}p/K, \ldots, n_{so}p/K]^T,$$

$$\boldsymbol{\Sigma}_{i,i}^o = \frac{\gamma_i^o(n_{so} - \gamma_i^o)}{n_{so}}, \quad \boldsymbol{\Sigma}_{i,j}^o = \frac{-\gamma_i^o \gamma_j^o}{n_{so}}.$$

If we denote a new $(K-1) \times K$ matrix as

$$\boldsymbol{T} = \begin{pmatrix} 1 & -1 & 0 & \ldots & 0 \\ 1 & 0 & -1 & \ldots & 0 \\ \vdots & \ddots & \ddots & \ldots & 0 \\ 1 & 0 & 0 & \ldots & -1 \end{pmatrix}$$

And the new $(K-1)$-dim random variable, i.e,

$$\boldsymbol{\eta} \doteq (\boldsymbol{T}, \boldsymbol{T}, 4\boldsymbol{T})(\bar{\boldsymbol{r}}, \tilde{\boldsymbol{r}}, \boldsymbol{r}^o)^T + (\bar{n}_v + \tilde{n}_v + 2n_{vo})\mathbf{1}$$

is still Gaussian, to be specific, if we denote its distribution as $\boldsymbol{\eta} \sim \mathcal{N}(\boldsymbol{\alpha}, \boldsymbol{M})$, then we have

$$\boldsymbol{\alpha} = ((\bar{n}_s + \tilde{n}_s + 2n_{so})(1 - p) + \bar{n}_v + \tilde{n}_v + 2n_{vo})\mathbf{1},$$

$$\boldsymbol{M}_{i,i} = (\bar{n}_s + \tilde{n}_s + 14n_{so})\frac{p(K + 2 - pK)}{K},$$

$$\boldsymbol{M}_{i,j} = (\bar{n}_s + \tilde{n}_s + 14n_{so})\frac{p(K + 1 - pK)}{K},$$

$\boldsymbol{G}(\bar{n}_v + \tilde{n}_v, \bar{n}_s + \tilde{n}_s, n_{vo}, n_{so}, 4)$ can be approximated as

$$\mathbb{P}(\boldsymbol{\eta}_1 > 0, \ldots, \boldsymbol{\eta}_{K-1} > 0),$$

which is equal to $F_p\left(((\bar{n}_s + \tilde{n}_s + 2n_{so})(1 - p) + \bar{n}_v + \tilde{n}_v + 2n_{vo})/\sqrt{\bar{n}_s + \tilde{n}_s + 14n_{so}}\right)$, and $F_p(\cdot)$ is defined in Appendix F.2.5.

### F.2.3 PROOF FOR OUTPUT SPACE ENSEMBLE

For ensemble model, we also denote

$$\hat{\boldsymbol{w}}(k) = \frac{\sum_{\bar{i}=1}^{\bar{n}_v - n_{vo}} \boldsymbol{\mu}_{v,\bar{i}}(k) + \sum_{\bar{j}=1}^{\bar{n}_s - n_{so}} \boldsymbol{\mu}_{s,\bar{j}}(k) + \sum_{\tilde{i}=1}^{\tilde{n}_v - n_{vo}} \boldsymbol{\mu}_{v,\tilde{i}}(k) + \sum_{\tilde{i}=1}^{\tilde{n}_s - n_{so}} \boldsymbol{\mu}_{s,\tilde{i}}(k) + 2\sum_{i=1}^{n_{vo}} \boldsymbol{\mu}_{v,i}(k) + 2\sum_{i=1}^{n_{so}} \boldsymbol{\mu}_{s,i}(k)}{\sqrt{\bar{n}_v + \bar{n}_s + \tilde{n}_v + \tilde{n}_s + 2n_{vo} + 2n_{so}}},$$

then the forecasting accuracy on IID case is

$$\mathbb{P}(\hat{y} = y) = \frac{1}{K} \sum_{k=1}^{K} \mathbb{E}_{\boldsymbol{x}|y=e_k}\{\mathbf{1}(\boldsymbol{x}\bar{\Phi}^\top \bar{\boldsymbol{w}}(k) + \boldsymbol{x}\tilde{\Phi}^T \tilde{\boldsymbol{w}}(k) > \boldsymbol{x}\bar{\Phi}^\top \bar{\boldsymbol{w}}(k') + \boldsymbol{x}\tilde{\Phi}^T \tilde{\boldsymbol{w}}(k'), \forall k' \neq k)\}$$

$$= \frac{1}{K} \sum_{k=1}^{K} \mathbb{P}_{\boldsymbol{z} \sim \mathcal{N}(\boldsymbol{0}, \sigma^2 \boldsymbol{I}_d)}\left((\hat{\boldsymbol{w}}(k) - \hat{\boldsymbol{w}}(k'))^T \boldsymbol{z} + \hat{\delta}_{k,k'} > 0, \forall k' \neq k\right),$$

in which

$$\hat{\delta}_{k,k'} = \frac{\sum_{\bar{i}=1}^{\bar{n}_v - n_{vo}} 1 + \sum_{\bar{j}=1}^{\bar{n}_s - n_{so}} 1 + \sum_{\tilde{i}=1}^{\tilde{n}_v - n_{vo}} 1 + \sum_{\tilde{j}=1}^{\tilde{n}_s - n_{so}} 1 + \sum_{i=1}^{n_{vo}} 2 + \sum_{i=1}^{n_{so}} 2}{\sqrt{\bar{n}_v + \bar{n}_s + \tilde{n}_v + \tilde{n}_s + 2n_{vo} + 2n_{so}}\sqrt{\bar{n}_v + \bar{n}_s + \tilde{n}_v + \tilde{n}_s - n_{vo} - n_{so}}}$$

$$= \frac{\bar{n}_v + \bar{n}_s + \tilde{n}_v + \tilde{n}_s}{\sqrt{\bar{n}_v + \bar{n}_s + \tilde{n}_v + \tilde{n}_s + 2n_{vo} + 2n_{so}}\sqrt{\bar{n}_v + \bar{n}_s + \tilde{n}_v + \tilde{n}_s - n_{vo} - n_{so}}} \doteq s < 1,$$

while $n_{vo} + n_{so} < (\bar{n}_v + \bar{n}_s + \tilde{n}_v + \tilde{n}_s)/2$, for any $k' \neq k$. And considering Assumption 2, we have

$$(\hat{\boldsymbol{w}}(k) - \hat{\boldsymbol{w}}(k'))^T (\hat{\boldsymbol{w}}(k) - \hat{\boldsymbol{w}}(k'')) = 1, \quad \| \hat{\boldsymbol{w}}(k) - \hat{\boldsymbol{w}}(k') \|_2^2 = 2,$$

for any $k \neq k' \neq k''$. Then with Lemma 2, the IID forecasting accuracy can be expressed as

$$\mathbb{P}(\boldsymbol{y} = \hat{\boldsymbol{y}}) = \boldsymbol{L}(s, \dots, s) \geq 1 - \epsilon,$$

which can be influenced by $\bar{n}_v, \bar{n}_s, \tilde{n}_v, \tilde{n}_s, n_{vo}, n_{so}$.

Then we turn to the OOD forecasting accuracy. Similar to the notation in equation 9, for class $k$, we suppose:

$$\bar{r}_k = |\{\mathbb{I}(\boldsymbol{\mu}_{s,i}(k) = \boldsymbol{\mu}_{s,i}(k))\}_{i=1}^{\bar{n}_s} - n_{so}|$$
$$\tilde{r}_k = |\{\mathbb{I}(\boldsymbol{\mu}_{s,i}(k) = \boldsymbol{\mu}_{s,i}(k))\}_{i=1}^{\tilde{n}_s - n_{so}}|$$
$$r_k^o = |\{\mathbb{I}(\boldsymbol{\mu}_{s,i}(k) = \boldsymbol{\mu}_{s,i}(k))\}_{i=1}^{n_{so}}|$$
$$\bar{r}_{k \to k'} = |\{\mathbb{I}(\boldsymbol{\mu}_{s,i}(k) = \boldsymbol{\mu}_{s,i}(k'))\}_{i=1}^{\bar{n}_s} - n_{so}|$$
$$\tilde{r}_{k \to k'} = |\{\mathbb{I}(\boldsymbol{\mu}_{s,i}(k) = \boldsymbol{\mu}_{s,i}(k'))\}_{i=1}^{\tilde{n}_s - n_{so}}|$$
$$r_{k \to k'}^o = |\{\mathbb{I}(\boldsymbol{\mu}_{s,i}(k) = \boldsymbol{\mu}_{s,i}(k'))\}_{i=1}^{n_{so}}|.$$

Then denoting $R_k(r) := [\bar{r}_k, \tilde{r}_k, r_k^o, [\bar{r}_{k \to k'}, \tilde{r}_{k \to k'}, r_{k \to k'}^o, \forall k' \neq k]]$, we have the corresponding probability is

$$\mathbb{P}(R_k(r)) = \frac{(\bar{n}_s + \tilde{n}_s - 2n_{so})!n_{so}!}{(\bar{r}_k + \tilde{r}_k)!r_k^o!\Pi_{k' \neq k}(\bar{r}_{k \to k'} + \tilde{r}_{k \to k'})!r_{k \to k'}^o!}(1-p+\frac{p}{K})^{\bar{r}_k + \tilde{r}_k + r_k^o}(\frac{K-1}{K}p)^{\bar{n}_s + \tilde{n}_s - n_{so} - \bar{r}_k - \tilde{r}_k - r_k^o},$$

and the conditional OOD forecasting accuracy on label $\boldsymbol{e}_k$ is

$$\mathbb{P}(\hat{\boldsymbol{y}} = \boldsymbol{e}_k \mid R_k(r), \boldsymbol{y} = \boldsymbol{e}_k)$$
$$= \mathbb{E}_{\boldsymbol{x}\hat{\Phi} \mid R_k(r), \boldsymbol{y} = \boldsymbol{e}_k} \left\{ \mathbf{1}(\boldsymbol{x}\bar{\Phi}^\top\bar{\boldsymbol{w}}(k) + \boldsymbol{x}\tilde{\Phi}^T\tilde{\boldsymbol{w}}(k) > \boldsymbol{x}\bar{\Phi}^\top\bar{\boldsymbol{w}}(k') + \boldsymbol{x}\tilde{\Phi}^T\tilde{\boldsymbol{w}}(k'), \forall k' \neq k) \right\}$$
$$= \boldsymbol{L}(\frac{\bar{n}_v + \tilde{n}_v + \bar{r}_k + \tilde{r}_k + 2r_k^o - \bar{r}_{k \to k'} - \tilde{r}_{k \to k'} - 2r_{k \to k'}^o}{\sqrt{\bar{n}_v + \bar{n}_s + \tilde{n}_v + \tilde{n}_s + 2n_{vo} + 2n_{so}}\sqrt{\bar{n}_v + \bar{n}_s + \tilde{n}_v + \tilde{n}_s - n_{vo} - n_{so}}}, \forall k' \neq k).$$

According to this, the OOD forecasting accuracy can be expressed as

$$\mathbb{P}(\hat{\boldsymbol{y}} = \boldsymbol{y}) = \mathbb{E}_{\boldsymbol{y}}[\mathbb{P}(\hat{\boldsymbol{y}} = \boldsymbol{y} \mid \boldsymbol{y})] = \mathbb{P}(\hat{\boldsymbol{y}} = \boldsymbol{e}_1 \mid \boldsymbol{y} = \boldsymbol{e}_1)$$
$$= \sum_{R_1(k)} \mathbb{P}(R_1(k))\boldsymbol{L}(\frac{\bar{n}_v + \tilde{n}_v + \bar{r}_1 + \tilde{r}_1 + 2r_1^o - \bar{r}_{1 \to k'} - \tilde{r}_{1 \to k'} - 2r_{1 \to k'}^o}{\sqrt{\bar{n}_v + \bar{n}_s + \tilde{n}_v + \tilde{n}_s + 2n_{vo} + 2n_{so}}\sqrt{\bar{n}_v + \bar{n}_s + \tilde{n}_v + \tilde{n}_s - n_{vo} - n_{so}}}, \forall k' \neq 1).$$

with Lemma 3, we can get related properties about $\boldsymbol{L}(\cdot)$, then take upper and lower bounds respectively.

To be specific, recalling the expression in equation 13 with $n_v = \bar{n}_v + \tilde{n}_v, n_s = \bar{n}_s + \tilde{n}_s, n_{vo} = n_{vo}, n_{so} = n_{so}, C = 2$, we can lower bound the OOD forecasting accuracy as

$$\mathbb{P}(\hat{\boldsymbol{y}} = \boldsymbol{y}) \geq \mathbb{P}(\mathcal{A})(1 - \epsilon) + \sum_{N=1}^{K-1} \mathbb{P}(\mathcal{C}(N))(h(N) - \epsilon)$$

$$\geq \mathbb{P}(\mathcal{A}) + \sum_{N=1}^{K-1} \mathbb{P}(\mathcal{C}(N))h(N) - \epsilon = \boldsymbol{G}(\bar{n}_v + \tilde{n}_v, \bar{n}_s + \tilde{n}_s, n_{vo}, n_{so}, 2) - \epsilon,$$

and on the other hand, it can be upper bounded by

$$\mathbb{P}(\hat{\boldsymbol{y}} = \boldsymbol{y}) \leq \mathbb{P}(\mathcal{A}) + \sum_{N=1}^{K-1} \mathbb{P}(\mathcal{C}(N))h(N) + \epsilon\mathbb{P}(\mathcal{B}) \leq G(\bar{n}_v + \tilde{n}_v, \bar{n}_s + \tilde{n}_s, n_{vo}, n_{so}, 2) + \epsilon,$$

Similar to the analysis before, for ID forecasting accuracy, we have

$$\mathcal{J}_{id} = 0 \leq 3\epsilon,$$

and for OOD forecasting accuracy, we can draw a conclusion that

$$\mathcal{J}_{ood} \geq \boldsymbol{G}(\bar{n}_v + \tilde{n}_v, \bar{n}_s + \tilde{n}_s, n_{vo}, n_{so}, 2) - \max\{\boldsymbol{G}(\bar{n}_v, \bar{n}_s, 0, 0, 0), \boldsymbol{G}(\tilde{n}_v, \tilde{n}_s, 0, 0, 0)\} - 3\epsilon.$$

Similar to the analysis above, we'd like to take some intuitive approximation for OOD forecasting accuracy in the ensemble model. As the number $\bar{n}_v, \bar{n}_s, \tilde{n}_v, \tilde{n}_s, n_{vo}, n_{so}$ are large enough, we can take approximation by multivariate Gaussian distribution. To be specific, we denote $\bar{\boldsymbol{r}} = [\bar{r}_1, \bar{r}_{1\rightarrow 2}, \ldots, \bar{r}_{1\rightarrow K}]$, $\tilde{\boldsymbol{r}} = [\tilde{r}_1, \tilde{r}_{1\rightarrow 2}, \ldots, \tilde{r}_{1\rightarrow K}]$ and $\boldsymbol{r}^o = [r_1^o, r_{1\rightarrow 2}^o, \ldots, r_{1\rightarrow K}^o]$, then can regard them as $\bar{\boldsymbol{r}} \sim \mathcal{N}(\bar{\boldsymbol{\gamma}}, \bar{\boldsymbol{\Sigma}})$, $\tilde{\boldsymbol{r}} \sim \mathcal{N}(\tilde{\boldsymbol{\gamma}}, \tilde{\boldsymbol{\Sigma}})$ and $\boldsymbol{r}^o \sim \mathcal{N}(\boldsymbol{\gamma}^o, \boldsymbol{\Sigma}^o)$ (they are independent), in which

$$\bar{\boldsymbol{\gamma}} = [(\bar{n}_s - n_{so})(1 - p + p/K), (\bar{n}_s - n_{so})p/K, \ldots, (\bar{n}_s - n_{so})p/K]^T,$$

$$\bar{\boldsymbol{\Sigma}}_{i,i} = \frac{\bar{\gamma}_i(\bar{n}_s - n_{so} - \bar{\gamma}_i)}{\bar{n}_s - n_{so}}, \quad \bar{\boldsymbol{\Sigma}}_{i,j} = \frac{-\bar{\gamma}_i\bar{\gamma}_j}{\bar{n}_s - n_{so}},$$

$$\tilde{\boldsymbol{\gamma}} = [(\tilde{n}_s - n_{so})(1 - p + p/K), (\tilde{n}_s - n_{so})p/K, \ldots, (\tilde{n}_s - n_{so})p/K]^T,$$

$$\tilde{\boldsymbol{\Sigma}}_{i,i} = \frac{\tilde{\gamma}_i(\tilde{n}_s - n_{so} - \tilde{\gamma}_i)}{\tilde{n}_s - n_{so}}, \quad \tilde{\boldsymbol{\Sigma}}_{i,j} = \frac{-\tilde{\gamma}_i\tilde{\gamma}_j}{\tilde{n}_s - n_{so}},$$

$$\boldsymbol{\gamma}^o = [n_{so}(1 - p + p/K), n_{so}p/K, \ldots, n_{so}p/K]^T,$$

$$\boldsymbol{\Sigma}_{i,i}^o = \frac{\gamma_i^o(n_{so} - \gamma_i^o)}{n_{so}}, \quad \boldsymbol{\Sigma}_{i,j}^o = \frac{-\gamma_i^o\gamma_j^o}{n_{so}}.$$

If we denote a new $(K-1) \times K$ matrix as

$$\boldsymbol{T} = \begin{pmatrix} 1 & -1 & 0 & \ldots & 0 \\ 1 & 0 & -1 & \ldots & 0 \\ \vdots & \ddots & \ddots & \ldots & 0 \\ 1 & 0 & 0 & \ldots & -1 \end{pmatrix}$$

And the new $(K-1)$-dim random variable, i.e,

$$\boldsymbol{\eta} \doteq (\boldsymbol{T}, \boldsymbol{T}, 4\boldsymbol{T})(\bar{\boldsymbol{r}}, \tilde{\boldsymbol{r}}, \boldsymbol{r}^o)^T + (\bar{n}_v + \tilde{n}_v + 2n_{vo})\mathbf{1}$$

is still Gaussian, to be specific, if we denote its distribution as $\boldsymbol{\eta} \sim \mathcal{N}(\boldsymbol{\alpha}, \boldsymbol{M})$, then we have

$$\boldsymbol{\alpha} = ((\bar{n}_s + \tilde{n}_s)(1 - p) + \bar{n}_v + \tilde{n}_v)\mathbf{1},$$

$$\boldsymbol{M}_{i,i} = (\bar{n}_s + \tilde{n}_s + 2n_{so})\frac{p(K + 2 - pK)}{K},$$

$$\boldsymbol{M}_{i,j} = (\bar{n}_s + \tilde{n}_s + 2n_{so})\frac{p(K + 1 - pK)}{K},$$

the OOD forecasting accuracy can be approximated as

$$\mathbb{P}(\eta_1 > 0, \ldots, \eta_{K-1} > 0),$$

which is equal to $F_p(((\bar{n}_s + \tilde{n}_s)(1 - p) + \bar{n}_v + \tilde{n}_v)/\sqrt{\bar{n}_s + \tilde{n}_s + 2n_{so}})$, and $F_p(\cdot)$ is defined in Appendix F.2.5.

### F.2.4 CASE STUDY FOR $K = 3$

To interpret the improvements on OOD accuracy of model average and model ensemble, here we set $K = 3$, and further take an insight on the representation function $G(\cdot)$.

Recalling the results calculated above, the OOD accuracy for single models can be approximated as

$$G(\bar{n}_v, \bar{n}_s, 0, 0, 0), \quad G(\tilde{n}_v, \tilde{n}_s, 0, 0, 0),$$

and for average model and ensemble model, we could focus on

$$G(\bar{n}_v + \tilde{n}_v, \bar{n}_s + \tilde{n}_s, n_{vo}, n_{so}, 4), \quad G(\bar{n}_v + \tilde{n}_v, \bar{n}_s + \tilde{n}_s, n_{vo}, n_{so}, 2).$$

To take specific calculations, we denote the random vector as $[r_1, r_1^o, r_{1\rightarrow 2}, r_{1\rightarrow 3}, r_{1\rightarrow 2}^o, r_{1\rightarrow 3}^o]$, and approximate them on probabilities related to the two-dimensional Gaussian random vector $\boldsymbol{\eta} \sim \mathcal{N}(0, H)$, in which the covariance matrix $H$ has components as

$$H_{ii} = \frac{p(5 - 3p)}{3}, \quad H_{ij} = \frac{p(4 - 3p)}{3}.$$

Then we could obtain

$$G(\bar{n}_v, \bar{n}_s, 0, 0, 0) = \mathbb{P}(\eta_1 \geq -\frac{(1-p)\bar{n}_s + \bar{n}_v}{\sqrt{\bar{n}_v}}, \eta_2 \geq -\frac{(1-p)\bar{n}_s + \bar{n}_v}{\sqrt{\bar{n}_v}}),$$

$$G(\tilde{n}_v, \tilde{n}_s, 0, 0, 0) = \mathbb{P}(\eta_1 \geq -\frac{(1-p)\tilde{n}_s + \tilde{n}_v}{\sqrt{\tilde{n}_v}}, \eta_2 \geq -\frac{(1-p)\tilde{n}_s + \tilde{n}_v}{\sqrt{\tilde{n}_v}}),$$

$$G(\bar{n}_v + \tilde{n}_v, \bar{n}_s + \tilde{n}_s, n_{vo}, n_{so}, 4)$$
$$= \mathbb{P}(\eta_1 \geq -\frac{(1-p)(\bar{n}_s + \tilde{n}_s + 2n_{so}) + \tilde{n}_v + \bar{n}_v + 2n_{vo}}{\sqrt{\bar{n}_s + \tilde{n}_s + 14n_{so}}}, \eta_2 \geq -\frac{(1-p)(\bar{n}_s + \tilde{n}_s + 2n_{so}) + \tilde{n}_v + \bar{n}_v + 2n_{vo}}{\sqrt{\bar{n}_s + \tilde{n}_s + 14n_{so}}}),$$

$$G(\bar{n}_v + \tilde{n}_v, \bar{n}_s + \tilde{n}_s, n_{vo}, n_{so}, 2)$$
$$= \mathbb{P}(\eta_1 \geq -\frac{(1-p)(\bar{n}_s + \tilde{n}_s) + \tilde{n}_v + \bar{n}_v}{\sqrt{\bar{n}_s + \tilde{n}_s + 2n_{so}}}, \eta_2 \geq -\frac{(1-p)(\bar{n}_s + \tilde{n}_s) + \tilde{n}_v + \bar{n}_v}{\sqrt{\bar{n}_s + \tilde{n}_s}}).$$

Here we denote a new function
$$F(x) := \mathbb{P}(\eta_1 \geq -x, \eta_2 \geq -x),$$
which implies that $F$ is monotonically increasing with respect to $x$. And it shows that average model and ensemble model could obtain higher OOD accuracy compared with single models due to

$$\frac{(1-p)(\bar{n}_s + \tilde{n}_s + 2n_{so}) + \tilde{n}_v + \bar{n}_v + 2n_{vo}}{\sqrt{\bar{n}_s + \tilde{n}_s + 14n_{so}}} \geq \max\{\frac{(1-p)\bar{n}_s + \bar{n}_v}{\sqrt{\bar{n}_v}}, \frac{(1-p)\tilde{n}_s + \tilde{n}_v}{\sqrt{\tilde{n}_v}}\},$$

$$\frac{(1-p)(\bar{n}_s + \tilde{n}_s) + \tilde{n}_v + \bar{n}_v}{\sqrt{\bar{n}_s + \tilde{n}_s + 2n_{so}}} \geq \max\{\frac{(1-p)\bar{n}_s + \bar{n}_v}{\sqrt{\bar{n}_v}}, \frac{(1-p)\tilde{n}_s + \tilde{n}_v}{\sqrt{\tilde{n}_v}}\}.$$

### F.2.5 CLOSE FORM OF $F_p(\cdot)$

Here we provide the explicit expression of function $F_p(x)$ in $K$ class situation, which is monotonically increasing with $x$.

We denote a $K - 1$-dim random variable $\boldsymbol{\eta} \sim \mathcal{N}(\boldsymbol{x}, \boldsymbol{M})$, in which

$$\boldsymbol{M}_{i,i} = \frac{p(K + 2 - pK)}{K}, \boldsymbol{M}_{i,j} = \frac{p(K + 1 - pK)}{K},$$

then $F_p(x)$ is defined as

$$F_p(x) = \mathbb{P}(\boldsymbol{\eta}_1 > 0, \dots, \boldsymbol{\eta}_{K-1} > 0).$$

### F.2.6 CLOSE FORM OF $G(n_v, n_s, n_{vo}, n_{so}, C)$

First, denoting a random vector $R_k(r) := [r_k, r_k^o, [r_{k \to k'}, r_{k \to k'}^o, \forall k' \neq k]] \in \mathbb{R}^{2K}$, we have the corresponding probability as

$$\mathbb{P}(R_k(r)) = \frac{(n_s - 2n_{so})! n_{so}!}{r_k! r_k^o! \Pi_{k' \neq k} r_{k \to k'}! r_{k \to k'}^o!} (1 - p + \frac{p}{K})^{r_k + r_k^o} (\frac{K-1}{K}p)^{n_s - n_{so} - r_k - r_k^o},$$

then we can define several sets:

$$\mathcal{A} := \{R_k(r) : r_1 + Cr_1^o - r_{1 \to k'} - Cr_{1 \to k'}^o + n_v > 0, \forall k' = 2, \dots, K\}, \tag{10}$$

$$\mathcal{B} := \{R_k(r) : \min_{k'=2,\dots,K} r_1 + Cr_1^o - r_{1 \to k'} - Cr_{1 \to k'}^o + n_v < 0\}, \tag{11}$$

$$\mathcal{C}(N) := \{R_k(r) : \min_{k'=2,\dots,K} r_1 + Cr_1^o - r_{1 \to k'} - Cr_{1 \to k'}^o + n_v = 0,$$
$$\text{the minimum can be achieved by N values}\}, \tag{12}$$

and related functions as

$$h(N) = \mathbb{P}_{\boldsymbol{z} \sim \mathcal{N}(0, \sigma^2 \boldsymbol{I}_N)} \left( \boldsymbol{a}_i^T \boldsymbol{z} > 0, \forall i = 1, \dots, N \right)$$

in which $\boldsymbol{a}_i^T \boldsymbol{a}_j = 1$ and $\| \boldsymbol{a}_i \|_2^2 = 1$ for any $i \neq j$.

$G(n_s, n_v, n_{so}, n_{vo}, C)$ is the probability defined as following:

$$G(n_s, n_v, n_{so}, n_{vo}, C) = \mathbb{P}(\mathcal{A}) + \sum_{N=1}^{K-1} \mathbb{P}(\mathcal{C}(N)) h(N) \tag{13}$$

where set $\mathcal{A}$ and $\mathcal{C}(N)$ are two sets of $R_k(r)$ defined in Equation equation 10 and equation 12, respectively. Note that $\mathbb{P}(R_k(r))$ and the set $\mathcal{A}$ and $\mathcal{C}(N)$ all depend on $n_s, n_v, n_{so}, n_{vo}$ and $C$.

### F.3 PROOF OF PROPOSITION 4

By the proof in Proposition 1 we have

$$\bar{\boldsymbol{w}}(k) = \sum_{i=1}^{2} \boldsymbol{\mu}_{v,i}(k) + \sum_{j=1}^{3} \boldsymbol{\mu}_{s,j}(k).$$

$$\tilde{\boldsymbol{w}}(k) = \sum_{i=3}^{4} \boldsymbol{\mu}_{v,i}(k) + \sum_{j=4}^{6} \boldsymbol{\mu}_{s,j}(k).$$

Then we consider the averaged mode about the value of $\hat{\boldsymbol{w}}^T \boldsymbol{x} \hat{\Phi}^T$, where $\hat{\boldsymbol{w}} = \frac{\bar{\boldsymbol{w}} + \lambda \tilde{\boldsymbol{w}}}{1+\lambda}$ and $\hat{\Phi} = \frac{\bar{\Phi} + \lambda \tilde{\Phi}}{1+\lambda}$

We first have:

$$\hat{\boldsymbol{w}}(k) = \frac{1}{1+\lambda}\left( \sum_{i=1,2} \boldsymbol{\mu}_{v,i}(k) + \lambda \sum_{i=3,4} \boldsymbol{\mu}_{v,i}(k) + \sum_{j=1,2,3} \boldsymbol{\mu}_{s,j}(k) + \sum_{j=4,5,6} \boldsymbol{\mu}_{s,j}(k) \right)$$

$$\boldsymbol{x}\hat{\Phi}^T|_{y=e_1} = \frac{1}{1+\lambda}(x\bar{\Phi}^T + \lambda x\tilde{\Phi}^T)$$

$$= \frac{1}{1+\lambda}\left( \sum_{i=1,2} \boldsymbol{\mu}_{v,i} Q_{v,i}(1) + \lambda \sum_{i=3,4} \boldsymbol{\mu}_{v,i} Q_{v,i}(1) \right.$$

$$\left. + \sum_{j=1,2,3} \boldsymbol{\mu}_{s,j} Q_{s,j}(1) + \lambda \sum_{j=4,5,6} \boldsymbol{\mu}_{s,j} Q_{s,j}(1) + \left( \sum_{i=1}^{5} z_i + \lambda \sum_{i=6}^{10} z_i \right) \right)$$

$$\hat{\boldsymbol{w}}(k)^T \boldsymbol{x}\hat{\Phi}^T|_{y=e_1} = \frac{1}{(1+\lambda)^2}\left( \sum_{i=1,2} \boldsymbol{\mu}_{v,i}(k)^T \boldsymbol{\mu}_{v,i} Q_{v,i}(1) + \lambda^2 \sum_{i=3,4} \boldsymbol{\mu}_{v,i}(k)^T \boldsymbol{\mu}_{v,i} Q_{v,i}(1) \right.$$

$$\left. + \sum_{j=1,2,3} \boldsymbol{\mu}_{s,j}(k)^T \boldsymbol{\mu}_{s,j} Q_{s,j}(1) + \lambda^2 \sum_{j=4,5,6} \boldsymbol{\mu}_{s,j}(k)^T \boldsymbol{\mu}_{s,j} Q_{s,j}(1) \right)$$

Then for class $k = 1$, we have

$$\hat{\boldsymbol{w}}(1)^T \boldsymbol{x}\hat{\Phi}^T|_{y=e_1} = \frac{1}{(1+\lambda)^2}(\underbrace{2+2\lambda^2}_{>12} + \sum_{j=1,2,3} \boldsymbol{\mu}_{s,j}(1)^T \boldsymbol{\mu}_{s,j} Q_{s,j}(1) + \underbrace{\lambda^2}_{>5} \sum_{j=4,5,6} \boldsymbol{\mu}_{s,j}(1)^T \boldsymbol{\mu}_{s,j} Q_{s,j}(1))$$

For the other two classes, we have

$$\hat{\boldsymbol{w}}(2)^T \boldsymbol{x}\hat{\Phi}^T|_{y=e_1} = \frac{1}{(1+\lambda)^2}\left( \sum_{j=1,2,3} \boldsymbol{\mu}_{s,j}(2)^T \boldsymbol{\mu}_{s,j} Q_{s,j}(1) \right) + \underbrace{\lambda^2}_{>5} \sum_{j=4,5,6} \boldsymbol{\mu}_{s,j}(2)^T \boldsymbol{\mu}_{s,j} Q_{s,j}(1))$$

$$\hat{\boldsymbol{w}}(3)^T \boldsymbol{x}\hat{\Phi}^T|_{y=e_1} = \frac{1}{(1+\lambda)^2}\left( \sum_{j=1,2,3} \boldsymbol{\mu}_{s,j}(3)^T \boldsymbol{\mu}_{s,j} Q_{s,j}(1) \right) + \underbrace{\lambda^2}_{>5} \sum_{j=4,5,6} \boldsymbol{\mu}_{s,j}(3)^T \boldsymbol{\mu}_{s,j} Q_{s,j}(1))$$

For simplicity of the discussion, we will ignore the constant factor $\frac{1}{(1+\lambda)^2}$, when $\lambda > \sqrt{5}$, we discuss the value of $\hat{\boldsymbol{w}}(1)^T \boldsymbol{x}\hat{\Phi}^T|_{y=e_1} - \hat{\boldsymbol{w}}(2)^T \boldsymbol{x}\hat{\Phi}^T|_{y=e_1}$ :

When $\sum_{j=4,5,6} \mathbb{I}(\boldsymbol{Q}_{s,j}(1) = \boldsymbol{e}_2) = 2$, it suffices to consider comparing:

$$\begin{cases} 2 + \sum_{j=1,2,3} \boldsymbol{\mu}_{s,j}(1)^T \boldsymbol{\mu}_{s,j} Q_{s,j}(1) + \lambda^2 \sum_{j=4,5,6} \boldsymbol{\mu}_{s,j}(1)^T \boldsymbol{\mu}_{s,j} Q_{s,j}(1) \\ \sum_{j=1,2,3} \boldsymbol{\mu}_{s,j}(2)^T \boldsymbol{\mu}_{s,j} Q_{s,j}(1) \end{cases}$$

$\hat{\boldsymbol{w}}(1)^T \boldsymbol{x}\hat{\Phi}^T|_{y=e_1} \leq \hat{\boldsymbol{w}}(2)^T \boldsymbol{x}\hat{\Phi}^T|_{y=e_1}$ only when $\sum_{j=4,5,6} \mathbb{I}(\boldsymbol{Q}_{s,j}(1) = \boldsymbol{e}_1) = 0$, that is,

$\sum_{j=4,5,6} \mathbb{I}(\boldsymbol{Q}_{s,j}(1) = \boldsymbol{e}_3) = 1$.

The probability of the aforementioned scenario is $3 \cdot (p/3)^3 = p^3/9$.

Then

$$\begin{cases} \hat{\boldsymbol{w}}(1)^T \boldsymbol{x}\hat{\Phi}^T|_{y=e_1} < \hat{\boldsymbol{w}}(2)^T \boldsymbol{x}\hat{\Phi}^T|_{y=e_1} \text{ if } \sum_{j=1,2,3} \mathbb{I}(\boldsymbol{Q}_{s,j}(1) = \boldsymbol{e}_2) = 3 \\ \hat{\boldsymbol{w}}(1)^T \boldsymbol{x}\hat{\Phi}^T|_{y=e_1} = \hat{\boldsymbol{w}}(2)^T \boldsymbol{x}\hat{\Phi}^T|_{y=e_1} \text{ if } \sum_{j=1,2,3} \mathbb{I}(\boldsymbol{Q}_{s,j}(1) = \boldsymbol{e}_2) = 2 \text{ and } \sum_{j=1,2,3} \mathbb{I}(\boldsymbol{Q}_{s,j}(1) = \boldsymbol{e}_3) = 1 \end{cases}$$

Therefore, the probability of "$<$" is $p^3/9 \cdot (p/3)^3 = p^6/243$, "$=$" is $p^3/9 \cdot 3 \cdot (p/3)^3 = p^6/81$.

When $\sum_{j=4,5,6} \mathbb{I}(\boldsymbol{Q}_{s,j}(1) = \boldsymbol{e}_2) = 3$, it suffices to consider comparing

$$\begin{cases} 2 + \sum_{j=1,2,3} \boldsymbol{\mu}_{s,j}(1)^T \boldsymbol{\mu}_{s,j} Q_{s,j}(1) \\ \sum_{j=1,2,3} \boldsymbol{\mu}_{s,j}(2)^T \boldsymbol{\mu}_{s,j} Q_{s,j}(1) + \lambda^2 \end{cases}$$

Trivially we have $\sum_{j=1,2,3} \boldsymbol{\mu}_{s,j}(2)^T \boldsymbol{\mu}_{s,j} Q_{s,j}(1) + \lambda^2 > 5 \geq 2 + \sum_{j=1,2,3} \boldsymbol{\mu}_{s,j}(1)^T \boldsymbol{\mu}_{s,j} Q_{s,j}(1)$, therefore, "$<$" holds under this case, the probability is $(p/3)^3 = p^3/27$.

When $\sum_{j=4,5,6} \mathbb{I}(\boldsymbol{Q}_{s,j}(1) = \boldsymbol{e}_2) = 0/1$,

$$\sum_{j=1,2,3} \boldsymbol{\mu}_{s,j}(2)^T \boldsymbol{\mu}_{s,j} Q_{s,j}(1)) + \lambda^2 \sum_{j=4,5,6} \boldsymbol{\mu}_{s,j}(2)^T \boldsymbol{\mu}_{s,j} Q_{s,j}(1)) \leq 3 + \lambda^2 < 2 + 2\lambda^2 + \cdots$$ Therefore,

"$\leq$" is impossible to hold in this case.

To sum up, for comparing the first class and the second class, the "$<$" probability is $p^6/243 + p^3/27$, the "$=$" probability is $p^6/81$.

Generally, the "$<$" probability is $2p^6/243 + 2p^3/27$, the "$=$" probability is $2p^6/81$, the otherwise probability is $1 - 8p^6/243 - 2p^3/27$. Then the total accuracy is approximately $1/2 \cdot 2p^6/81 + (1 - 8p^6/243 - 2p^3/27) = 1 - 5p^6/243 - 2p^3/27$, that is, the accuracy lies in $[1 - 5p^6/243 - 2p^3/27 - \varepsilon, 1 - 5p^6/243 - 2p^3/27 + \varepsilon]$.

### F.4 PROOF OF PROPOSITION 5

*Proof.* (a)**Two individual models**. The accuracy of two individual models are the same with Example (1-1) following the same proof. Specifically, so we have $\mathcal{A}_{\text{ood}}(\bar{f}) \in [1 - 5p^3/54 - \epsilon, 1 - 5p^3/54 + \epsilon]$. $\mathcal{A}_{\text{ood}}(\tilde{f}) \in [1 - 5p^3/27 - \epsilon, 1 - 5p^3/27 + \epsilon]$

(b) **Weight space ensemble**. We first solve the $\bar{\boldsymbol{w}}$ and $\tilde{\boldsymbol{w}}$ on the infinite ID samples. Lemma 5, for $k = 1, 2, 3$, we have

$$\bar{\boldsymbol{w}}(k) = \sum_{i=1}^{2} \boldsymbol{\mu}_{v,i}(k) + \sum_{j=1}^{3} \boldsymbol{\mu}_{s,j}(k),$$

$$\tilde{\boldsymbol{w}}(k) = \sum_{i=2}^{3} \boldsymbol{\mu}_{v,i}(k) + \sum_{j=3}^{5} \boldsymbol{\mu}_{s,j}(k),$$

So we have

$$\bar{\boldsymbol{w}}(k) + \tilde{\boldsymbol{w}}(k) = \sum_{i=1,3} \boldsymbol{\mu}_{v,i}(k) + 2\boldsymbol{\mu}_{v,2}(k) + \sum_{j=1,2,4,5} \boldsymbol{\mu}_{s,j}(k) + 2\boldsymbol{\mu}_{s,3}(k)$$

For samples from the first class, we also have

$$\boldsymbol{x}(\bar{\Phi} + \tilde{\Phi})|_{\boldsymbol{y}=\boldsymbol{e}_1} = \sum_{i=1,3} \boldsymbol{\mu}_{v,i} \boldsymbol{Q}_{v,i}(1) + 2\boldsymbol{\mu}_{v,2} \boldsymbol{Q}_{v,2}(1) + \sum_{j=1,2,4,5} \boldsymbol{\mu}_{s,j} \boldsymbol{Q}_{s,j}(1) + 2\boldsymbol{\mu}_{s,3} \boldsymbol{Q}_{s,3}(1) + \sum_{i=1}^{10} \boldsymbol{z}_i$$

where $\boldsymbol{z}_i \sim \mathcal{N}(0, \sigma^2 \boldsymbol{I}_d), \forall i$. We then have

$$(\bar{\boldsymbol{w}}(k) + \tilde{\boldsymbol{w}}(k))^\top \boldsymbol{x}(\bar{\Phi} + \tilde{\Phi})|_{\boldsymbol{y}=\boldsymbol{e}_1}$$
$$= \sum_{i=1,3} \boldsymbol{\mu}_{v,i}(k)^\top (\boldsymbol{\mu}_{v,i} \boldsymbol{Q}_{v,i}(1)) + 4\boldsymbol{\mu}_{v,2}(k)^\top (\boldsymbol{\mu}_{v,2} \boldsymbol{Q}_{v,2}(1))$$
$$+ \sum_{j=1,2,4,5} \boldsymbol{\mu}_{s,j}(k)^\top (\boldsymbol{\mu}_{s,j} \boldsymbol{Q}_{s,j}(1)) + 4\boldsymbol{\mu}_{s,3}(k)^\top (\boldsymbol{\mu}_{s,3} \boldsymbol{Q}_{s,3}(1)) + \xi$$

So

$$(\bar{\boldsymbol{w}}(1) + \tilde{\boldsymbol{w}}(1))^\top \boldsymbol{x}(\bar{\Phi} + \tilde{\Phi})|_{\boldsymbol{y}=\boldsymbol{e}_1}$$
$$= 6 + \sum_{j=1,2,4,5} \boldsymbol{\mu}_{s,j}(1)^\top (\boldsymbol{\mu}_{s,j} \boldsymbol{Q}_{s,j}(1)) + 4\boldsymbol{\mu}_{s,3}(1)^\top (\boldsymbol{\mu}_{s,3} \boldsymbol{Q}_{s,3}(1)) + \xi$$

Similarly, we have

$$(\bar{\boldsymbol{w}}(2)+\tilde{\boldsymbol{w}}(2))^\top \boldsymbol{x}(\bar{\Phi}+\tilde{\Phi})|_{\boldsymbol{y}=\boldsymbol{e}_1} = \sum_{j=1,2,4,5} \boldsymbol{\mu}_{s,j}(2)^\top \left(\boldsymbol{\mu}_{s,j}\boldsymbol{Q}_{s,j}(1)\right) + 4\boldsymbol{\mu}_{s,3}(2)^\top \left(\boldsymbol{\mu}_{s,3}\boldsymbol{Q}_{s,3}(1)\right) + \xi,$$

$$(\bar{\boldsymbol{w}}(3)+\tilde{\boldsymbol{w}}(3))^\top \boldsymbol{x}(\bar{\Phi}+\tilde{\Phi})|_{\boldsymbol{y}=\boldsymbol{e}_1} = \sum_{j=1,2,4,5} \boldsymbol{\mu}_{s,j}(3)^\top \left(\boldsymbol{\mu}_{s,j}\boldsymbol{Q}_{s,j}(1)\right) + 4\boldsymbol{\mu}_{s,3}(3)^\top \left(\boldsymbol{\mu}_{s,3}\boldsymbol{Q}_{s,3}(1)\right) + \xi.$$

Then

$$(\bar{\boldsymbol{w}}(1)+\tilde{\boldsymbol{w}}(1))^\top \boldsymbol{x}\left(\bar{\Phi}+\tilde{\Phi}\right)^\top - (\bar{\boldsymbol{w}}(2)+\tilde{\boldsymbol{w}}(2))^\top \boldsymbol{x}\left(\bar{\Phi}+\tilde{\Phi}\right)^\top$$

$$= \begin{cases} -2, \text{ if } \sum_{j=1,2,4,5} \mathbb{I}(\boldsymbol{Q}_{s,j}(1)=\boldsymbol{e}_2)=4, \text{ and } \mathbb{I}(\boldsymbol{Q}_{s,3}(1)=\boldsymbol{e}_2)=1 \\ -1, \text{ if } \left(\mathbb{I}(\boldsymbol{Q}_{s,3}(1)=\boldsymbol{e}_2)=1, \sum_{j=1,2,4,5} \mathbb{I}(\boldsymbol{Q}_{s,j}(1)=\boldsymbol{e}_2)=3, \text{ and } \sum_{j=1,2,4,5} \mathbb{I}(\boldsymbol{Q}_{s,j}(1)=\boldsymbol{e}_3)=1\right), \\ 0, \text{ if } \left(\mathbb{I}(\boldsymbol{Q}_{s,3}(1)=\boldsymbol{e}_2)=1, \sum_{j=1,2,4,5} \mathbb{I}(\boldsymbol{Q}_{s,j}(1)=\boldsymbol{e}_2)=3, \text{ and } \sum_{j=1,2,4,5} \mathbb{I}(\boldsymbol{Q}_{s,j}(1)=\boldsymbol{e}_1)=1\right) \text{ or} \\ \qquad \left(\mathbb{I}(\boldsymbol{Q}_{s,3}(1)=\boldsymbol{e}_2)=1, \sum_{j=1,2,4,5} \mathbb{I}(\boldsymbol{Q}_{s,j}(1)=\boldsymbol{e}_2)=2, \text{ and } \sum_{j=1,2,4,5} \mathbb{I}(\boldsymbol{Q}_{s,j}(1)=\boldsymbol{e}_3)=2\right). \\ \geq 1 \text{ otherwise.} \end{cases}, $$

Then we can compute the probability respectively,
For -2 case, the probability is given by $2 \cdot (p/3)^5 = 2p^5/243$
For -1 case, the probability is given by $2 \cdot 4 \cdot (p/3)^5 = 8p^5/243$
For 0 case, the probability is given by $2 \cdot (4 \cdot (1-2p/3) \cdot (p/3)^4 + 4C2 \cdot (p/3)^5) = 8p^4/81 - 4p^5/243$
Otherwise, the probability is: $1 - 8p^4/81 - 2p^5/81$

Then the total accuracy can be computed as approximately $1 - 4p^4/81 - 8p^5/243$,
the interval is $[1 - 4p^4/81 - 8p^5/243 - \varepsilon, 1 - 4p^4/81 - 8p^5/243 + \varepsilon]$

(c) **Output Space Ensemble**. Similar to the derivation of model averaging, we have

$$\bar{\boldsymbol{w}}(k)^\top \boldsymbol{x}\bar{\Phi} + \tilde{\boldsymbol{w}}(k)^\top \boldsymbol{x}\tilde{\Phi}|_{\boldsymbol{y}=\boldsymbol{e}_1}$$
$$= \sum_{i=1,3} \boldsymbol{\mu}_{v,i}(k)^\top \left(\boldsymbol{\mu}_{v,i}\boldsymbol{Q}_{v,i}(1)\right) + 2\boldsymbol{\mu}_{v,2}(k)^\top \left(\boldsymbol{\mu}_{v,2}\boldsymbol{Q}_{v,2}(1)\right)$$
$$+ \sum_{j=1,2,4,5} \boldsymbol{\mu}_{s,j}(k)^\top \left(\boldsymbol{\mu}_{s,j}\boldsymbol{Q}_{s,j}(1)\right) + 2\boldsymbol{\mu}_{s,3}(k)^\top \left(\boldsymbol{\mu}_{s,3}\boldsymbol{Q}_{s,3}(1)\right) + \xi.$$

Then we consider the fist class:

$$\bar{\boldsymbol{w}}(1)^\top \boldsymbol{x}\bar{\Phi} + \tilde{\boldsymbol{w}}(1)^\top \boldsymbol{x}\tilde{\Phi}|_{\boldsymbol{y}=\boldsymbol{e}_1}$$
$$= \sum_{i=1,3} \boldsymbol{\mu}_{v,i}(1)^\top \left(\boldsymbol{\mu}_{v,i}\boldsymbol{Q}_{v,i}(1)\right) + 2\boldsymbol{\mu}_{v,2}(1)^\top \left(\boldsymbol{\mu}_{v,2}\boldsymbol{Q}_{v,2}(1)\right)$$
$$+ \sum_{j=1,2,4,5} \boldsymbol{\mu}_{s,j}(1)^\top \left(\boldsymbol{\mu}_{s,j}\boldsymbol{Q}_{s,j}(1)\right) + 2\boldsymbol{\mu}_{s,3}(1)^\top \left(\boldsymbol{\mu}_{s,3}\boldsymbol{Q}_{s,3}(1)\right) + \xi.$$
$$= 4 + \sum_{j=1,2,4,5} \boldsymbol{\mu}_{s,j}(1)^\top \left(\boldsymbol{\mu}_{s,j}\boldsymbol{Q}_{s,j}(1)\right) + 2\boldsymbol{\mu}_{s,3}(1)^\top \left(\boldsymbol{\mu}_{s,3}\boldsymbol{Q}_{s,3}(1)\right) + \xi.$$

Similarly we have,

$$\bar{\boldsymbol{w}}(2)^\top \boldsymbol{x}\bar{\Phi} + \tilde{\boldsymbol{w}}(2)^\top \boldsymbol{x}\tilde{\Phi}|_{\boldsymbol{y}=\boldsymbol{e}_1} = \sum_{j=1,2,4,5} \boldsymbol{\mu}_{s,j}(2)^\top \left(\boldsymbol{\mu}_{s,j}\boldsymbol{Q}_{s,j}(1)\right) + 2\boldsymbol{\mu}_{s,3}(2)^\top \left(\boldsymbol{\mu}_{s,3}\boldsymbol{Q}_{s,3}(1)\right) + \xi.$$

$$\bar{\boldsymbol{w}}(3)^\top \boldsymbol{x}\bar{\Phi} + \tilde{\boldsymbol{w}}(3)^\top \boldsymbol{x}\tilde{\Phi}|_{\boldsymbol{y}=\boldsymbol{e}_1} = \sum_{j=1,2,4,5} \boldsymbol{\mu}_{s,j}(3)^\top \left(\boldsymbol{\mu}_{s,j}\boldsymbol{Q}_{s,j}(1)\right) + 2\boldsymbol{\mu}_{s,3}(3)^\top \left(\boldsymbol{\mu}_{s,3}\boldsymbol{Q}_{s,3}(1)\right) + \xi.$$

Then

$$(\bar{\boldsymbol{w}}(1)^\top \boldsymbol{x}\bar{\Phi} + \tilde{\boldsymbol{w}}(1)^\top \boldsymbol{x}\tilde{\Phi})|_{\boldsymbol{y}=\boldsymbol{e}_1} - (\bar{\boldsymbol{w}}(2)^\top \boldsymbol{x}\bar{\Phi} + \tilde{\boldsymbol{w}}(2)^\top \boldsymbol{x}\tilde{\Phi})|_{\boldsymbol{y}=\boldsymbol{e}_1}$$

$$= \begin{cases} -2, \text{ if } \sum_{j=1,2,4,5} \mathbb{I}(\boldsymbol{Q}_{s,j}(1)=\boldsymbol{e}_2)=4, \text{ and } \boldsymbol{Q}_{s,3}(1)=\boldsymbol{e}_2 \\ -1, \text{ if } \sum_{j=1,2,4,5} \mathbb{I}(\boldsymbol{Q}_{s,j}(1)=\boldsymbol{e}_2)=3, \sum_{j=1,2,4,5} \mathbb{I}(\boldsymbol{Q}_{s,j}(1)=\boldsymbol{e}_3)=1 \text{ and } \boldsymbol{Q}_{s,3}(1)=\boldsymbol{e}_2 \\ 0, \text{ if } (\sum_{j=1,2,4,5} \mathbb{I}(\boldsymbol{Q}_{s,j}(1)=\boldsymbol{e}_2)=2 \text{ and } \sum_{j=1,2,4,5} \mathbb{I}(\boldsymbol{Q}_{s,j}(1)=\boldsymbol{e}_3)=2) \text{ and } \boldsymbol{Q}_{s,3}(1)=\boldsymbol{e}_2) \\ \quad \text{or } (\sum_{j=1,2,4,5} \mathbb{I}(\boldsymbol{Q}_{s,j}(1)=\boldsymbol{e}_2)=4 \text{ and } \boldsymbol{Q}_{s,3}(1)=\boldsymbol{e}_3) \\ \quad \text{or } (\sum_{j=1,2,4,5} \mathbb{I}(\boldsymbol{Q}_{s,j}(1)=\boldsymbol{e}_2)=3, \sum_{j=1,2,4,5} \mathbb{I}(\boldsymbol{Q}_{s,j}(1)=\boldsymbol{e}_1)=1, \boldsymbol{Q}_{s,3}(1)=\boldsymbol{e}_2) \\ \geq 1 \text{ otherwise.} \end{cases}$$

Then we can compute the probability respectively:

For -2 case, the probability is given by $2 \cdot (p/3)^5 = 2p^5/243$

For -1 case, the probability is given by $2 \cdot 4 \cdot (p/3)^4 \cdot (p/3) = 8p^5/243$

For 0 case, the probability is given by $2 \cdot (4C2(p/3)^4 \cdot (p/3) + (p/3)^4 \cdot (p/3) + 4 \cdot (1 - 2p/3) \cdot (p/3)^3 \cdot (p/3)) = 8p^4/81 - 2p^5/243$

Otherwise, the probability is given by $1 - 8p^4/81 - 8p^5/243$

Then the probability can be computed as approximately $[1 - 4p^4/81 - p^5/27 - \varepsilon, 1 - 4p^4/81 - p^5/27 + \varepsilon]$.

$\square$

## F.5 Auxiliary Lemmas

**Lemma 1.** *For any $N$ i.i.d random variables $\{z_i\}_{i=1}^N \sim \mathcal{N}(0, \sigma^2)$ and $t > 0$, with probability at least $1 - Ne^{-\frac{t^2}{2\sigma^2}}$, we have*

$$z_i + t \geq 0, \quad \text{for any} \quad i = 1, \ldots, N.$$

*Proof.* For any index $i$, according to Markov inequality, with any $\lambda > 0$, we have

$$\mathbb{P}(z_i + t \leq 0) = \mathbb{P}(e^{\lambda z_i} \geq e^{\lambda t}) \leq \frac{\mathbb{E}e^{\lambda z_i}}{e^{\lambda t}} \leq \exp\{\frac{\lambda^2 \sigma^2}{2} - \lambda t\},$$

taking the minimum value on the right handside, with respect to $\lambda$, we can get

$$\mathbb{P}(z_i + t \leq 0) \leq \exp\{-\frac{t^2}{2\sigma^2}\}.$$

Then considering the random variables through all index $i = 1, \ldots, N$,

$$\mathbb{P}(\min_i z_i + t \leq 0) = \mathbb{P}(e^{\max_i \lambda z_i} \geq e^{\lambda t})) \leq \mathbb{E}(\exp\{\max_i \lambda z_i - \lambda t\}),$$

while

$$\mathbb{E}\exp\{\max_i \lambda z_i\} = \mathbb{E}\max_i e^{\lambda z_i} \leq \sum_i \mathbb{E}e^{\lambda z_i} \leq Ne^{\lambda^2 \sigma^2/2},$$

we can take the minimum value on the right hand side, with respect with $\lambda$, then further obtain

$$\mathbb{P}(\min_i z_i + t \geq 0) \leq Ne^{-t^2/2\sigma^2}.$$

$\square$

**Lemma 2.** *Suppose that $\boldsymbol{x} \sim \mathcal{N}(\boldsymbol{0}, \sigma^2 \boldsymbol{I}_d)$, $k - 1$ vectors $\{\boldsymbol{a}_1, \ldots, \boldsymbol{a}_{k-1}\}$ and $\delta_i \in \mathbb{R}$ for $i = 1, \ldots, k - 1$, then by Gram-Schmidt process, the probability of*

$$\begin{cases} \boldsymbol{a}_1^T \boldsymbol{x} + \delta_1 > 0, \\ \cdots \\ \boldsymbol{a}_{k-1}^T \boldsymbol{x} + \delta_{k-1} > 0, \end{cases}$$

*is equivalent to the probability of*

$$\begin{cases} e_1 + \dfrac{\delta_1}{\| \boldsymbol{a}_1 \|_2} > 0, \\ \sqrt{1 - (\dfrac{\boldsymbol{a}_2^T \boldsymbol{v}_1}{\| \boldsymbol{a}_2 \|_2 \| \boldsymbol{v}_1 \|_2})^2} e_2 + \dfrac{\boldsymbol{a}_2^T \boldsymbol{v}_1}{\| \boldsymbol{a}_2 \|_2 \| \boldsymbol{v}_1 \|_2} e_1 + \dfrac{\delta_2}{\| \boldsymbol{a}_2 \|_2} > 0, \\ \cdots \\ \sqrt{1 - \sum_{i=1}^{k-2}(\dfrac{\boldsymbol{a}_{k-1}^T \boldsymbol{v}_i}{\| \boldsymbol{a}_{k-1} \|_2 \| \boldsymbol{v}_i \|_2})^2} e_{k-1} + \sum_{i=1}^{k-2} \dfrac{\boldsymbol{a}_{k-1}^T \boldsymbol{v}_i}{\| \boldsymbol{a}_{k-1} \|_2 \| \boldsymbol{v}_i \|_2} e_i + \dfrac{\delta_{k-1}}{\| \boldsymbol{a}_{k-1} \|_2} > 0. \end{cases}$$

*in which $\{e_i\}$ are i.i.d. $\mathcal{N}(0, \sigma^2)$ and $\{\boldsymbol{v}_i\}_{i=1}^{k-1}$ are orthogonal vectors span on $\{\boldsymbol{a}_i\}_{i=1}^{k-1}$ as*

$$\begin{cases} \boldsymbol{v}_1 = \dfrac{\boldsymbol{a}_1}{\| \boldsymbol{a}_1 \|_2}, \\ \boldsymbol{v}_2 = \dfrac{\boldsymbol{a}_2 - (\boldsymbol{a}_2^T \boldsymbol{v}_1)\boldsymbol{v}_1}{\sqrt{\| \boldsymbol{a}_2 \|_2^2 - (\boldsymbol{a}_2^T \boldsymbol{v}_1)^2}}, \\ \cdots \\ \boldsymbol{v}_{k-1} = \dfrac{\boldsymbol{a}_{k-1} - \sum_{i=1}^{k-2}(\boldsymbol{a}_{k-1}^T \boldsymbol{v}_i)\boldsymbol{v}_i}{\sqrt{\| \boldsymbol{a}_{k-1} \|_2^2 - \sum_{i=1}^{k-2}(\boldsymbol{a}_{k-1}^T \boldsymbol{v}_i)^2}}, \end{cases}$$

**Lemma 3.** *Just consider two classes $K = 1, 2$ and denote $r_{1 \to 2}$ as the number of the events $\{\mathbb{I}(\boldsymbol{Q}_{s,i}(1) = e_2)\}$ that holds. Suppose Assumption 1 hold, then*

1. *when $n_s + n_v - 2r_{1 \to 2} > 0$, the related probability $\mathbb{P}((\boldsymbol{w}(1) - \boldsymbol{w}(2))^T \boldsymbol{x} \boldsymbol{\Phi} > 0 \mid_{y=e_1})$ is larger than $\sqrt{1 - \epsilon}$,*

2. *when $n_s + n_v - 2r_{1 \to 2} = 0$, the related probability $\mathbb{P}((\boldsymbol{w}(1) - \boldsymbol{w}(2))^T \boldsymbol{x} \boldsymbol{\Phi} > 0 \mid_{y=e_1})$ in $[1/2 - \epsilon, 1/2 + \epsilon]$,*

3. *when $n_s + n_v - 2r_{1 \to 2} < 0$, the related probability $\mathbb{P}((\boldsymbol{w}(1) - \boldsymbol{w}(2))^T \boldsymbol{x} \boldsymbol{\Phi} > 0 \mid_{y=e_1})$ is less than $\epsilon$.*

*Proof.* This can be directly deduced by using Lemma 4, which is the more general version. $\square$

**Lemma 4.** *Denote $r_{k \to l}$ as the number of the events $\{\mathbb{I}(\boldsymbol{Q}_{s,i}(k) = e_l)\}_{i=1}^{n_s}$ that hold. Suppose Assumption 1 hold, then for class $k$,*

- *when $n_s + n_v - \sum_{l \neq k} r_{k \to l} > \max_{l \neq k} r_{k \to l}$, the accuracy is larger than $1 - \epsilon$,*

- *when $n_s + n_v - \sum_{l \neq k} r_{k \to l} = \max_{l \neq k} r_{k \to l}$, denote $N$ as the number of the events that holds $\{\mathbb{I}(r_{k \to l} = n_s + n_v - \sum_{l'} r_{k \to l'})\}_{l \neq k}$, the accuracy in $[1/(N+1) - \epsilon, 1/(N+1) + \epsilon]$,*

- *when $n_s + n_v - \sum_{l \neq k} r_{k \to l} < \max_{l \neq k} r_{k \to l}$, the accuracy is less than $\epsilon$.*

*Proof.* Considering the conditional forecasting accuracy on class $k$, with respect to $r_{k \to 1}, \ldots, r_{k \to K}$, it is equivalent with $G(\{n_s + n_v - \sum_{l \neq k} r_{k \to l} - r_{k \to s}, \forall s \neq k\})$, which is defined previously. Then we can take analysis case by case:

- $n_s + n_v - \sum_{l \neq k} r_{k \to l} > \max_{l \neq k} r_{k \to l}$

  In this case, all elements in function $G(\cdot)$ are larger than 0, which means that no smaller than $1/(N'_v + N'_s)$. With Assumption 1, we have

  $$G(\{n_s + n_v - \sum_{l \neq k} r_{k \to l} - r_{k \to s}, \forall s \neq k\}) \geq \boldsymbol{F}^K(\frac{1}{\sigma(N'_v + N'_s)}) \geq 1 - \epsilon.$$

- $n_s + n_v - \sum_{l \neq k} r_{k \to l} = \max_{l \neq k} r_{k \to l}$

  In this case, all elements in function $G(\cdot)$ are no smaller than $1/(N'_v + N'_s)$, except $N$ zero elements. With Assumption 1, we have

  $$G(\{n_s + n_v - \sum_{l \neq k} r_{k \to l} - r_{k \to s}, \forall s \neq k\}) \geq \frac{1}{2^N}(1 - \epsilon) \geq \frac{1}{2^N} - \epsilon.$$

- $n_s + n_v - \sum_{l \neq k} r_{k \to l} < \max_{l \neq k} r_{k \to l}$

  In this case, there is at least one element in $G(\cdot)$ no larger than $-1/(N'_v + N'_s)$, still considering Assumption 1, we have

  $$G(\{n_s + n_v - \sum_{l \neq k} r_{k \to l} - r_{k \to s}, \forall s \neq k\}) \leq \boldsymbol{F}(-\frac{1}{\sigma(N'_v + N'_s)}) \leq \epsilon.$$

$\square$

**Lemma 5.** *With features $\{\boldsymbol{x}_{v,i}\}_{i=1}^{n_v}$ and $\{\boldsymbol{x}_{s,j}\}_{j=1}^{n_s}$, the classifier trained on infinite samples is equivalent with the mean value of $\boldsymbol{x} = \sum_{i=1}^{n_v} \boldsymbol{x}_i + \sum_{j=1}^{n_s} \boldsymbol{x}_j$.*

*Proof.* Considering the classifier, it should be $\max_w \mathbb{P}(\hat{\boldsymbol{y}} | \Phi(\boldsymbol{x})^\top w)$. From Definition 1, we have $\boldsymbol{x} \mid \boldsymbol{y} \sim \mathcal{N}(\boldsymbol{\mu} \circ \boldsymbol{y}, (n_v + n_s)\sigma^2)$, in which $\boldsymbol{\mu} = \sum_{i=1}^{n_v} \boldsymbol{\mu}_{v,i} + \sum_{j=1}^{n_s} \boldsymbol{\mu}_{s,j}$. For simplicity, we denote $\boldsymbol{z} = \boldsymbol{x} \mid \boldsymbol{y} - \boldsymbol{\mu} \circ \boldsymbol{y}$, and $\boldsymbol{z}$ is independent of $\boldsymbol{y}$.

Given samples as $\{(\boldsymbol{x}_i, \boldsymbol{y}_i)\}_i$, by maximum likelihood estimation (MLE), we have

$$\arg\max_w \Pi_{i=1}^n \frac{\exp\{\frac{1}{(n_v+n_s)\sigma^2}\boldsymbol{x}_i^\top(\boldsymbol{w}\circ\boldsymbol{y}_i)\}}{\sum_{\boldsymbol{e}_j}\exp\{\frac{1}{(n_v+n_s)\sigma^2}\boldsymbol{x}_i^\top(\boldsymbol{w}\circ\boldsymbol{e}_j)\}}$$

$$\iff \arg\max_w \sum_{i=1}^n \frac{1}{(n_v+n_s)\sigma^2}\boldsymbol{x}_i^\top(\boldsymbol{w}\circ\boldsymbol{y}_i) - \sum_{i=1}^n \log\left(\sum_{\boldsymbol{e}_j}\exp\{\frac{1}{(n_v+n_s)\sigma^2}\boldsymbol{x}_i^\top(\boldsymbol{w}\circ\boldsymbol{e}_j)\}\right),$$

then taking derivative for each $w_k$, we have

$$\frac{1}{(n_v+n_s)\sigma^2}\sum_{i^k=1}^{n^k}\boldsymbol{x}_{ik} - \sum_{i=1}^n \frac{1}{(n_v+n_s)\sigma^2}\frac{\exp\{\frac{1}{(n_v+n_s)\sigma^2}\boldsymbol{x}_i^\top w_k\}}{\sum_{\boldsymbol{e}_j}\exp\{\frac{1}{(n_v+n_s)\sigma^2}\boldsymbol{x}_i^\top(\boldsymbol{w}\circ\boldsymbol{e}_j)\}}\boldsymbol{x}_i = \boldsymbol{0}, \forall k = 1, \ldots, K.$$

On the other hand, using Bayesian formula to consider the conditional expectation of $\boldsymbol{x}$, we have

$$\mathbb{E}(\boldsymbol{x}\mid\boldsymbol{y}=\boldsymbol{e}_k) = \frac{\mathbb{E}(\boldsymbol{x}\mathbf{1}(\boldsymbol{y}=\boldsymbol{e}_k))}{\mathbb{P}(\boldsymbol{y}=\boldsymbol{e}_k)} = \frac{\mathbb{E}(\boldsymbol{x}\mathbb{E}[\mathbf{1}(\boldsymbol{y}=\boldsymbol{e}_k)\mid\boldsymbol{x}])}{\mathbb{P}(\boldsymbol{y}=\boldsymbol{e}_k)}$$

$$\iff \mathbb{E}(\boldsymbol{x}\mid\boldsymbol{y}=\boldsymbol{e}_k) = K\mathbb{E}\left(\boldsymbol{x}\frac{\exp\{\frac{1}{(n_v+n_s)\sigma^2}\boldsymbol{x}^\top(\boldsymbol{\mu}\circ\boldsymbol{e}_k)\}}{\sum_{r=1}^K\exp\{\frac{1}{(n_v+n_s)\sigma^2}\boldsymbol{x}_i^\top(\boldsymbol{\mu}\circ\boldsymbol{e}_r)\}}\right),$$

it implies that as sample size $n$ goes to infinity, the following term can maximize the likelihood function:

$$\boldsymbol{w}(k) = \boldsymbol{\mu}\circ\boldsymbol{e}_k = (\sum_{i=1}^{n_v}\boldsymbol{\mu}_{v,i} + \sum_{j=1}^{n_s}\boldsymbol{\mu}_j)\circ\boldsymbol{e}_k,$$

for $k = 1, \ldots, K$. And by scaling the classifier, we can get the estimated classifier as

$$\boldsymbol{w}(k) = \frac{1}{\sqrt{n_v+n_s}}\boldsymbol{\mu}\circ\boldsymbol{e}_k,$$

for any class $k = 1, \ldots, K$, which is the same as $(1/\sqrt{n_v+n_s})\mathbb{E}_{\boldsymbol{x}\mid\boldsymbol{y}=\boldsymbol{e}_k}[\boldsymbol{x}\mid\boldsymbol{y}=\boldsymbol{e}_k]$.

$\square$

# G    ILLUSTRATING THE THEORY OF WISE-FT

Recall Definition 2 that, $\bar{f}$ learns $\bar{n}_v$ invariant features and $\bar{n}_s$ spurious features, as well as another single model $\tilde{f}$ has $\tilde{n}_v$ invariant features and $\tilde{n}_s$ spurious features. Further, $\bar{f}$ and $\tilde{f}$ learns $n_{vo}$ overlapped invariant features and $n_{so}$ overlapped spurious features. Let $\bar{f}$ denote the pre-trained model and $\tilde{f}$ denote the fine-tuned model. WiSE-FT is specifically is the following: $\bar{f}$ has good OOD but bad ID, $\tilde{f}$ has bad OOD but good ID, and the weight space ensemble of $\bar{f}$ and $\tilde{f}$ has excellent OOD performance. These can be expressed as:

$$\begin{cases} \mathcal{A}_{id}(\bar{f}) < \mathcal{A}_{id}(\tilde{f}), \\ \mathcal{A}_{ood}(\bar{f}) > \mathcal{A}_{ood}(\tilde{f}), \\ \mathcal{A}_{ood}(\tilde{f}_{wse}) > \max\{\mathcal{A}_{ood}(\bar{f}), \mathcal{A}_{ood}(\tilde{f})\} \end{cases}$$

It straightforward that the ID accuracy satisfies the following inequality due to Assumption 2:

$$\mathcal{A}_{id}(\bar{f}) < \mathcal{A}_{id}(\tilde{f}), \text{if} \quad \bar{n}_v + \bar{n}_s < \tilde{n}_v + \tilde{n}_s.$$

Intuitively, if a model learns more feature, it can predict the label better in the ID setting.

As for the OOD accuracy, by Proposition 2, we have

$$\mathcal{A}_{ood}(\bar{f}) = F_p\left(\frac{(1-p)\bar{n}_s + \bar{n}_v}{\sqrt{\bar{n}_s}}\right), \quad \mathcal{A}_{ood}(\tilde{f}) = F_p\left(\frac{(1-p)\tilde{n}_s + \tilde{n}_v}{\sqrt{\tilde{n}_s}}\right).$$

Furthermore, by Proposition 3, the OOD accuracy of weight space ensemble (WSE) is

$$\mathcal{A}_{ood}(f_{wse}) = F_p\left(\frac{(1-p)(\bar{n}_s + \tilde{n}_s + 2n_{so}) + \bar{n}_v + \tilde{n}_v + 2n_{vo}}{\sqrt{\bar{n}_s + \tilde{n}_s + 14n_{so}}}\right)$$

Then the WiSE-FT phenomenon can be effectively explained if the following conditions holds:

$$\begin{cases} \bar{n}_v + \bar{n}_s < \tilde{n}_v + \tilde{n}_s, \\ \frac{(1-p)\bar{n}_s + \bar{n}_v}{\sqrt{\bar{n}_s}} > \frac{(1-p)\tilde{n}_s + \tilde{n}_v}{\sqrt{\tilde{n}_s}}, \\ \frac{(1-p)(\bar{n}_s + \tilde{n}_s + 2n_{so}) + \bar{n}_v + \tilde{n}_v + 2n_{vo}}{\sqrt{\bar{n}_s + \tilde{n}_s + 14n_{so}}} > \max\{\frac{(1-p)\bar{n}_s + \bar{n}_v}{\sqrt{\bar{n}_s}}, \frac{(1-p)\tilde{n}_s + \tilde{n}_v}{\sqrt{\tilde{n}_s}}\} \end{cases}$$

The theoretical results above characterize the conditions for the WiSE-FT phenomenon. To gain a better understanding, we use a concrete example for illustration: $p = 0.9$, there is no overlapped features learned by two models, i.e., $n_{so} = n_{vo} = 0$. The pretrained model $\bar{f}$ learns some invariant and spurious features, i.e., $\bar{n}_v = 2$, $\bar{n}_s = 4$. The fine-tuned $\tilde{f}$ model learns more spurious features and less invariant features, i.e., $\tilde{n}_v = 1, \tilde{n}_s = 6$. In this example, the fine-tuned model $\tilde{f}$ has better ID performance than the pre-trained $\bar{f}$ since $\tilde{n}_v + \tilde{n}_s = 7 > \bar{n}_v + \bar{n}_s = 6$. The fine-tuned model $\tilde{f}$ has worse OOD performance than the pretrained model $\bar{f}$ since $\tilde{f}$ focuses more on spurious features. Specifically, we have $\mathcal{A}_{ood}(\bar{f}) > \mathcal{A}_{ood}(\tilde{f})$ since

$$\mathcal{A}_{ood}(\bar{f}) = F_p\left(\frac{(1-p)\bar{n}_s + \bar{n}_v}{\sqrt{\bar{n}_s}}\right) \approx F_p(1.20),$$

$$\mathcal{A}_{ood}(\tilde{f}) = F_p\left(\frac{(1-p)\tilde{n}_s + \tilde{n}_v}{\sqrt{\tilde{n}_s}}\right) \approx F_p(0.97),$$

Based on Proposition 3, the OOD performance of wse is

$$\mathcal{A}_{ood}(f_{wse}) = F_p\left(\frac{(1-p)(\bar{n}_s + \tilde{n}_s + 2n_{so}) + \bar{n}_v + \tilde{n}_v + 2n_{vo}}{\sqrt{\bar{n}_s + \tilde{n}_s + 14n_{so}}}\right) \approx F_p(1.27).$$

Recall that $F_p(\cdot)$ is monotonically increasing, we can see that $\mathcal{A}_{ood}(f_{wse}) > \max\{\mathcal{A}_{ood}(\tilde{f}), \mathcal{A}_{ood}(\bar{f})\}$.

# H ILLUSTRATING THE EFFECTIVENESS OF BANG THROUGH THE LENS "ACCURACY ON THE CURVE"

Considering that Mixup and Label Smoothing (LS) enhance the OOD performance of the fine-tuned model, we investigate whether the improvement achieved by BANG is primarily due to better calibration or the fine-tuned model's enhanced OOD performance. In Appendix E.6, we present our findings, which include the following observations:

- Dividing the weight of the vanilla fine-tuned model by multiple scalars significantly enhances the performance of weight averaging, closely approaching the performance of BANG.
- BANG demonstrates the ability to correct a substantial number of misclassified samples compared to the fine-tuned model.

To further investigate the performance of BANG, we examine the concept of "Accuracy on the Line" (Miller et al., 2021; Liang et al., 2023). We generate many checkpoints of vanilla fine-tuning by using different hyper-parameters. Specifically, we fine-tune the model using various hyperparameters, including learning rates $(1e^{-5}, 2e^{-5}, 3e^{-5}, 5e^{-5})$, training epochs $(4, 8, 10, 12, 16, 20)$, and learning rate schedules (cosine, step decay). Notably, the default hyperparameters used in Section 4 and mentioned in Wortsman et al. (2022) are a learning rate of $3e^{-5}$, 10 training epochs, and a cosine scheduler. Weight averaging is applied to each fine-tuned checkpoint with the pretrained model.

Figure 21 illustrates the OOD performance of each fine-tuned model, as well as the averaged model that combines the fine-tuned model with the pre-trained model. Interestingly, we observe that the OOD accuracy of the averaged model forms a quadratic function with respect to the OOD accuracy of the fine-tuned model Liang et al. (2023), rather than a linear relationship as described in (Miller et al., 2021).

Furthermore, BANG demonstrates significant robustness in OOD scenarios, surpassing the curve of expected performance.

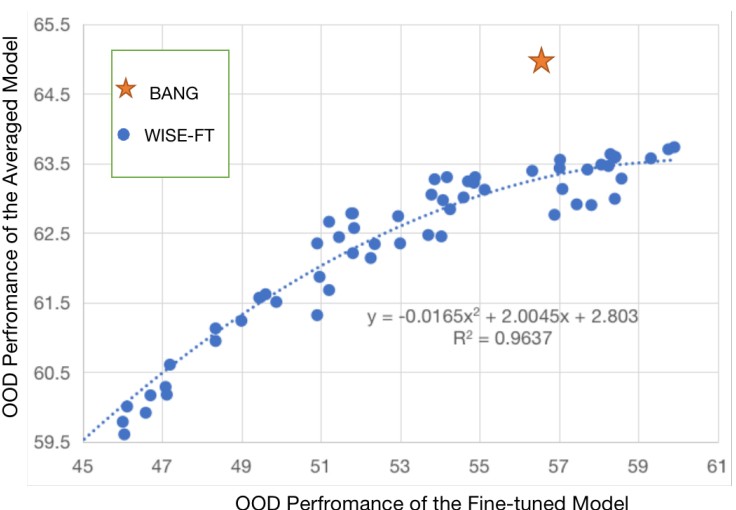

Figure 21: Illustrating the effectiveness of BANG through the lens of "accuracy on the curve".

