# OpenReview forum: "Spurious Feature Diversification Improves Out-of-distribution Generalization"
_ICLR.cc/2024/Conference — ICLR 2024 poster_

### Official Review · Reviewer_qFv2 · 2023-10-31

**Soundness:** 4 excellent
**Presentation:** 4 excellent
**Contribution:** 3 good
**Rating:** 8
**Confidence:** 4

**Summary:**

This paper investigates the WiSE-FT method's ability to enhance OOD performance in machine learning and identifies a notable "FalseFalseTrue" phenomenon that shows WiSE-FT correcting both individual model errors. Motivated by this, the authors conduct theoretical analysis in a multi-class setting. The theoretical results suggest that using diverse spurious features in ensemble methods improves OOD generalization, contradicting the traditional emphasis on invariant features. This theory is validated with experiments on the MultiColorMNIST dataset. Meanwhile, the authors also identify a weakness of WiSE-FT due to its overconfidence in fine-tuned models under OOD data. Based on that, the authors proposed a new method BANG that addresses the overconfidence problem and reaches better OOD performance.

**Strengths:**

1. The paper is well-written with a clear structure. Motivations are well-explained on why the authors study the problem and the contributions of this study are well discussed. The illustrative examples are helpful in understanding the concepts. Overall, it is easy to follow the logic and flow of the paper.
2. Theoretical results are solid and well-organized. The authors made the theoretical settings clear: definitions are well-explained and assumptions are clear. Proofs of the theory are sound as far as I read into.
3. Empirical results support the theoretical findings. The proposed method is also shown to be effective on the ImageNet dataset.

**Weaknesses:**

I have the following questions regarding the empirical results
1. The paper considers the setting of ensembling two individual models (FalseFalseTrue) and show that such ensemble indeed improves over single model's OOD performance on MultiColorMNIST. Is this finding generalizable to ensembles of more than two models? If we increase the number of models in Table 1, could the ensemble's performance potentially see further improvement?
2. For datasets that specifically focus on spurious correlations (Waterbirds), there will be only one strong spurious feature (background) in the training data. In such a case, will the ensemble method still be effective? If not, could we manually adding new spurious features to the training data (color, rotation, scale, ect.) and apply ensemble methods like BANG to improve the model's performance?

**Questions:**

See in weaknesses.

---

> ### Author Response · Authors · 2023-11-16
> **Response to Question 1/2**
>
> > Q1. The paper considers the setting of ensembling two individual models (FalseFalseTrue) and show that such ensemble indeed improves over single model's OOD performance on MultiColorMNIST. Is this finding generalizable to ensembles of more than two models? If we increase the number of models in Table 1, could the ensemble's performance potentially see further improvement?
>
>
> Response: Thank you for your valuable question. We conducted experiments on MultiColorMNIST and observed that increasing the number of models in the ensemble leads to significantly improved OOD performances. The table below demonstrates the results for different ensemble sizes. When the ensemble number is 1, it signifies the use of a single model without performing model ensemble. On the other hand, an ensemble number of 16 indicates that we independently trained 16 models with different initializations and used their ensemble for predictions. It is evident that increasing the ensemble number greatly boosts OOD performance.
>
> For instance, consider $p=0.8$ (as defined in Definition 1, where a larger $p$ indicates a larger distribution shift). The OOD performance of a single model (ensemble number equals 1) is 49.33\%. However, the ensemble of two models achieves 55.92\% OOD accuracy, while the ensemble of 16 models further increases the OOD accuracy to 64.85\%!
>
> These findings also provide insights into the effectiveness of the model soup approach [1], which involves averaging multiple checkpoints trained with different hyper-parameters. We appreciate your suggestions, and we have included these results in Appendix C.3.1.
>
> [1] Mitchell Wortsman, et.al., Model soups: averaging weights of multiple fine-tuned models improves accuracy without increasing inference time
>
>
> | $p$  | 0.70              | 0.75              | 0.80              | 0.85              | 0.90              |
> |--------------|-----------------|-----------------|-----------------|-----------------|-----------------|
> | Ensemble Number |                 |                 |                 |                 |                 |
> | 1            | 71.66 ± 2.06 | 60.68 ± 2.23 | 49.33 ± 2.02 | 37.74 ± 1.58 | 26.74 ± 1.05 |
> | 2            | 78.88 ± 1.24 | 68.34 ± 0.89 | 55.96 ± 0.77 | 42.91 ± 0.64 | 29.89 ± 0.63 |
> | 4            | 84.39 ± 1.33 | 74.00 ± 1.26 | 62.04 ± 1.32 | 47.92 ± 1.17 | 32.74 ± 0.75 |
> | 8            | 85.64 ± 1.22 | 75.73 ± 1.62 | 63.52 ± 1.61 | 49.15 ± 1.23 | 33.67 ± 0.93 |
> | 16           | 86.76 ± 0.55 | 77.31 ± 0.87 | 64.85 ± 1.09 | 50.63 ± 0.69 | 34.47 ± 0.40 |
>
> Table: Experiments on MultiColorMNIST

---

> ### Author Response · Authors · 2023-11-16
> **Response to Question 2/2**
>
> >  Q2: For datasets that specifically focus on spurious correlations (Waterbirds), there will be only one strong spurious feature (background) in the training data. In such a case, will the ensemble method still be effective? If not, could we manually adding new spurious features to the training data (color, rotation, scale, ect.) and apply ensemble methods like BANG to improve the model's performance?
>
>
> Response: It is intriguing to consider the scenario with a single spurious feature. In the WaterBird dataset, for instance, the dominating spurious feature is the background, and the dataset split for training and testing emphasizes situations where this spurious feature fails. To investigate model ensemble in this specific scenario, we have created a synthetic dataset called SingleColorMNIST in Appendix C.3. This dataset includes only one spurious feature, namely a single color patch. In the testing distribution, this spurious feature fails with a probability of $p$. Notably, SingleColorMNIST shares the same invariant feature, input size, and label space as MultiColorMNIST. However, since each individual model learns the same spurious feature, the ensemble of the model can not diversify the spurious feature, which can not improve the OOD performance. The empirical results presented below also show that vanilla ensemble methods do not improve OOD performance. We have included these new results in Appendix C.3.1. We truly appreciate your suggestions.
>
>
> | $p$  | 0.70              | 0.75              | 0.80              | 0.85              | 0.90              |
> |--------------|-----------------|-----------------|-----------------|-----------------|-----------------|
> | Ensemble Number |                 |                 |                 |                 |                 |
> | 1            | 37.40 ± 0.11 | 32.64 ± 0.11 | 27.90 ± 0.14 | 23.32 ± 0.14 | 19.32 ± 0.14 |
> | 2            | 37.32 ± 0.03 | 32.54 ± 0.05 | 27.78 ± 0.04 | 23.20 ± 0.04 | 19.18 ± 0.05 |
> | 4            | 37.35 ± 0.09 | 32.57 ± 0.12 | 27.81 ± 0.13 | 23.22 ± 0.11 | 19.23 ± 0.12 |
> | 8            | 37.30 ± 0.01 | 32.50 ± 0.01 | 27.74 ± 0.02 | 23.16 ± 0.02 | 19.16 ± 0.00 |
> | 16           | 37.35 ± 0.08 | 32.56 ± 0.09 | 27.82 ± 0.12 | 23.22 ± 0.10 | 19.24 ± 0.11 |
>
> Table: Experiments on SingleColorMNIST
>
> Regarding the idea of "manually adding new spurious features," we find it to be a novel concept that merits exploration. As you mentioned, data augmentation techniques could potentially be used to diversify a single spurious feature. Additionally, we could consider employing environment partition methods such as EIIL [2] to generate environments based on whether the spurious feature aligns or disagrees with the label. Subsequently, we could train diverse models in these environments and perform model ensemble. Given that implementing these ideas may require substantial effort, we can delve deeper into them as part of our future work.
>
> [2] Elliot Creager, et al., "Environment Inference for Invariant Learning"

---

### Official Review · Reviewer_KdWP · 2023-11-01

**Soundness:** 4 excellent
**Presentation:** 4 excellent
**Contribution:** 4 excellent
**Rating:** 8
**Confidence:** 3

**Summary:**

The paper examines WiSE-FT to understand how ensemble based models improve OOD performance. First, a “FalseFalseTrue” phenomenon is observed, where the ensembled model predicts samples correctly even when both the pretrained and finetuned model that are ensembled predict the samples incorrectly. The paper suggests an explanation for this related to ensembled models making use of diverse sparse features, and this reducing the influence of any given sparse feature below that of invariant features. This is then shown to be the case theoretically, and on a synthetic empirical setting. Finally, the paper makes the related observation that successful ensembling requires balanced models (so that the ensembled model benefts from diverse sparse features). This motivates a recommendation to finetune models with mixup and label smoothing to decrease confidence, and the resulting ensemble model is called BANG. BANG shows empirical performance stronger than that of WiSE-FT.

**Strengths:**

* Weight space ensembled models have been of great interest for OOD generalization and an explanation of how these models improve OOD performance beyond both pretrained and finetuned models—which this paper seeks to provide—is valuable.
* The FalseFalseTrue phenomenon is unexpected and interesting, especially since the FalseFalseTrue ratio is comparable to the total overall improvement provided by WiSE-FT.
* The analysis provides a partial mechanistic explanation for how EBM improves OOD performance beyond pretrained/finetuned models via diverse spurious features.
* The MultiColorMNIST experiment clearly demonstrates the value of diverse spurious features for the OOD performance of EBM.
* The observation that imbalanced models hurt EBM is actionable, and the algorithmic recommendation of BANG leads to OOD accuracy improvements over WiSE-FT.

**Weaknesses:**

I am not convinced that the performance gains of BANG are mainly due to better calibration as claimed in the paper and not because mixup/LS significantly improves OOD performance of finetuned models. While appendix E.6 shows that there are “oracle weights” to scale standard finetuned models when ensembling, it may not be that BANG models are bringing improvements for the same reason.

One way to demonstrate this would be analysis similar to “accuracy on the line” where there are models finetuned with mixup and LS at the same OOD accuracy as standard finetuned models. It would strengthen the claims of Section 4 if the BANG models in this setting outperform WiSE-FT even with finetuned models at the same OOD performance.

**Questions:**

* In tables 2 and 7, we see that on the in-distribution ImageNet test set, finetuned(Mixup+LS) models outperform standard finetuned models (by ~1.5%), but BANG does not outperform WiSE-FT. Is there intuition for what BANG is doing in the in-distribution setting that explains this?

* It would be good to see “FalseFalseTrue” analysis (like figure 2 left) of BANG models compared to WiSE-FT models. It appears from appendix E.6 that BANG models may be helpful in a “TrueFalseTrue” setting where the finetuned model was originally incorrect and overconfident. Does calibration help in the FalseFalseTrue setting?

---

> ### Author Response · Authors · 2023-11-16
> **Response to Question 1/3**
>
> > Question 1: I am not convinced that the performance gains of BANG are mainly due to better calibration as claimed in the paper and not because mixup/LS significantly improves OOD performance of finetuned models. While appendix E.6 shows that there are “oracle weights” to scale standard finetuned models when ensembling, it may not be that BANG models are bringing improvements for the same reason. One way to demonstrate this would be analysis similar to “accuracy on the line” where there are models finetuned with mixup and LS at the same OOD accuracy as standard finetuned models. It would strengthen the claims of Section 4 if the BANG models in this setting outperform WiSE-FT even with finetuned models at the same OOD performance.
>
> Response:
>
> We appreciate your insightful question and valuable suggestions regarding the "accuracy on the line" perspective.
> According to your suggestion, we have conducted additional experiments to address this aspect. To further investigate the performance of BANG, we examine the concept of ``Accuracy on the Line" [1, 2]. We try a lot of hyperparameters for the vanila fine-tuning. Specifically, we fine-tune the model using various hyperparameters, including learning rates ($1e^{-5}, 2e^{-5}, 3e^{-5}, 5e^{-5}$), training epochs ($4, 8, 10, 12, 16, 20$), and learning rate schedules (cosine, step decay). Notably, the default hyperparameters used in Section 4 and used in [3] are a learning rate of $3e^{-5}$, 10 training epochs, and a cosine scheduler. Weight averaging is applied to each fine-tuned checkpoint with the pretrained model. Figure 21 illustrates the OOD performance of each vanilla fine-tuned model, as well as the averaged model that combines the vanilla fine-tuned model with the pre-trained model. Interestingly, we observe that the OOD accuracy of the averaged model roughly forms a quadratic function with respect to the OOD accuracy of the vanilla fine-tuned model [2], rather than a linear relationship as described in [1].
>
> Furthermore, we also show the performance of our BANG on the figure. BANG demonstrates significant robustness in OOD scenarios, surpassing the curve of expected performance.
>
> [1] John Miller et.al., Accuracy on the Line: On the Strong Correlation Between Out-of-Distribution and In-Distribution Generalization
>
> [2] Weixin Lian et.al., Accuracy on the Curve: On the Nonlinear Correlation of ML Performance Between Data Subpopulations
>
> [3] Mitchell Wortsman et.al., Robust fine-tuning of zero-shot models

---

> ### Author Response · Authors · 2023-11-16
> **Response to Question 2/3 and 3/3**
>
> > Question 2: In tables 2 and 7, we see that on the in-distribution ImageNet test set, finetuned(Mixup+LS) models outperform standard finetuned models (by 1.5\%), but BANG does not outperform WiSE-FT. Is there intuition for what BANG is doing in the in-distribution setting that explains this?
>
> Response: Thank you for bringing up this intriguing phenomenon. We didn't notice this phenomenon before. We provide an possible explanation for this as follows: As we have demonstrated previously, the vanilla fine-tuned model tends to be over-confident. During the averaging process of WiSE-FT, there is a bias towards the fine-tuned model. Additionally, the fine-tuned model generally exhibits higher ID performance compared to the pre-trained model. Consequently, the ID performance of WiSE-FT can benefit from this bias towards the fine-tuned model. In contrast, BANG mitigates the issue of over-confidence in the fine-tuned model. By reducing the bias towards the fine-tuned model, larger weights are assigned to the pre-trained model. Again, it is worth noting that the pre-trained model has lower ID performance compared to the  fine-tuned model. As a result, BANG exhibits a relatively inferior ID performance compared to the model fine-tuned by MixUP and LS.
>
> > Question 3: It would be good to see “FalseFalseTrue” analysis (like figure 2 left) of BANG models compared to WiSE-FT models. It appears from appendix E.6 that BANG models may be helpful in a “TrueFalseTrue” setting where the finetuned model was originally incorrect and overconfident. Does calibration help in the FalseFalseTrue setting?
>
> Response: Thank you for your question. Let us use PM, FM and AM to denote the pre-trained, fine-tunede and averaged model, respectively.  We have found that the ratio of FalseFalseTrue (PM wrong, FM wrong and AM correct)increases from 2.5\% to 2.9\%. However, this increase is not as significant as the increase observed in the TrueFalseTrue (PM correct, FM wrong and AM correct) group, which goes from 10.6\% to 13.4\%. The visualization of the margin in Figure 20 of Appendix E.6 offers some insights. Specifically, in the TrueFalse group (PM correct, FM wrong), the Margin(PM) + Margin(FM) moves from the negative region to the positive region, crossing the boundary of y=-x. This indicates that BANG has a much higher probability of correcting the mistakes made by the fine-tuned model. Additionally, we also examine the FalseTrueTrue (PM wrong, FM correct and AM correct) group. This group demonstrates how much the mistakes made by the pre-trained model can be corrected by the fine-tuned model. The FalseTrueTrue ratio of BANG slightly decreases from 5.6\% to 4.4\% compared to vanilla fine-tuning. Importantly, this drop in the FalseTrueTrue ratio is much smaller than the increase observed in the TrueFalseTrue group (10.6\% to 13.4\%). These results suggest that alleviating the issue of over-confidence can benefit overall OOD performance. Furthermore, Figure 20 of Appendix E.6 provides further insights, showing that the Margin(PM) + Margin(FM) for BANG on the FalseTrue group (where the pre-trained model is false and the fine-tuned model is true) remains above the line y=-x. We hope this explanation clarifies the findings. If you have any more questions or need further clarification, we are more than willing to engage in further discussion.

---

### Official Review · Reviewer_kd2C · 2023-11-06

**Soundness:** 2 fair
**Presentation:** 3 good
**Contribution:** 3 good
**Rating:** 5
**Confidence:** 2

**Summary:**

The paper analyses ensembling to improve OOD robustness. They find that the individually incorrect models can average out to be correct when, loosely the models do depend on the invariant feature sufficiently in both models. They setup a linear data generating process and provide some evidence of when linear

**Strengths:**

- The paper studies an interesting phenomenon where individually incorrect models can be made correct. The idea is related to boosting, but the different models should rely on different spurious features.

 - The experiments are direct and encouraging, although not directly connected to the theory (WiSE-FT and WSE seem to not be compatible because the methods that work for WSE cannot be applied directly to WISE-FT.

 - The theoretical setup is interesting and does confirm some of the intuition of ensembling models that are diverse in their spuriousness.

**Weaknesses:**

The paper's weight space ensemble is too simple, when considering overconfident models. WiSE-FT ensembles all the weights, not just the linear layer. So it is unclear how the scaling can be assumed to be a simple multiplier on the linear layer.

This is a sizeable gap because WiSE-FT is weight space ensembling of very specific models (good OOD/bad ID and good ID/bad OOD). This is not reflected in the paper except loosely when they say low overlap is good weight space ensembles. The authors acknowledge this even when they talk about their method "However, this method can not be directly applied to WiSE-FT since WiSE-FT ensemble model weights instead of the outputs."

The paper does focus a lot of output ensembling, which is not known to work as well as WiSE-FT. In that case, there is no discussion of boosting, which seems to have the same flavor (especially when forcing models to rely on different features).


It is also unclear that Mix-up and label-smoothing have singular effects on the confidence, they also regularize. So BANG is really just finetuning with MIXup/LS and then doing WiSE-FT. A more direct correction of confidence would be the right one, like doing temperature scaling for each checkpoint during finetuning and then running WiSE-FT

**Questions:**

See weakneses.

---

> ### Author Response · Authors · 2023-11-16
> **Response to Question 1/5 (Part 1)**
>
> > Question 1: The paper's weight space ensemble is too simple, when considering overconfident models. WiSE-FT ensembles all the weights, not just the linear layer. So it is unclear how the scaling can be assumed to be a simple multiplier on the linear layer.
>
> Thanks for raising the concerns and sorry for that there might be some misunderstanding here.
>
> **Clarification on the theoretical models of weight space ensemble**. We would like to provide a clarification regarding the theoretical models of weight space ensemble. It is important to note that in our theoretical models, **we average  the weights of all layers, not just the output last layer**. Specifically, the observed input, denoted as $x = [x _ 1, x _ 2, ..., x _ {d _ s + d _ v}]$, is a concatenation of $d _ s + d _ v$ features, where each feature is represented in a d-dimensional space. Here $d _ v$ and $d _ s$ are the number of invariant and spurious features, respectively. The theoretical model follows a two-layer structure (also known as a bi-linear model). The first layer is parameterized by a binary feature mask $\Phi \in \{0, 1\}^{d _ s + d _ v}$. Consider the following example: Suppose we have a scenario with three features, $x = [x _ 1, x _ 2, x _ 3]$, and the feature mask $\Phi = [1, 1, 0]$, and the product of $x$ and $\Phi$ yields $x\Phi = x _ 1 + x _ 2$. The second layer of the model is a linear output layer, parameterized by $w \in \mathbb{R}^{d \times K}$, where $K$ represents the number of classes. The entire model can be expressed as $w^\top x \Phi$. Considering two individual models, denoted as $(\bar \Phi, \bar w)$ and $(\tilde \Phi, \tilde w)$, the weight space ensemble is computed as $f _ {wse} = \frac{1}{4}({\bar w^\top + \tilde w^\top}) x (\bar \Phi + \tilde{\Phi})$, based on Definition 4. So in conclusion, we ensemble the weights in all layers, not just the final layer. Later we will further discuss our theoretical model is clearly connected to localized a 2-layer DNN.
>
> In the next response, we are going to illustrate more details of our model  on weight space ensemble (WSE) and our findings on understanding WSE.

---

> ### Author Response · Authors · 2023-11-16
> **Response to Question 1/5  (Part 2)**
>
> Coutinued.
>
> > (Coutinued.) Question 1: The paper's weight space ensemble ... the linear layer.
>
> **Discussion on our model for weight space ensemble.** As you indicated, weight space ensemble is more effective than output space ensemble, however, the underlying reason behind this remains mysterious before our work. To the best of our knowledge, we are the first work that capture the intrinsic difference between weight and output space ensemble. Specifically, the difference between weight and output space ensemble (referred as OSE-WSE difference) is $f _ {wse} - f _ {ose} = \frac{1}{4}({\bar w^\top + \tilde w^\top}) x (\bar \Phi + \tilde{\Phi}) - \frac{1}{2} (\bar w^\top x \bar{\Phi} + \tilde w^\top x \tilde{\Phi})$. More importantly, in Appendix D.2, we show that the OSE-WSE difference captured  by our bi-linear model has a clear analogy with the OSE-WSE on two 2-layer DNNs that are not far away in the weight space.
>
> Specifically, consider a general 2-layer DNN parameterized by $(W _ a \in \mathbb{R}^{d _ 1 \times d _ 2}, W _ b \in \mathbb{R}^{d _ 2 \times K})$ with ReLU activation $\delta(\cdot)$ and output $f _ {dnn}(X) = W _ b^\top \delta(W _ a^\top X)$ for $X \in \mathbb{R}^{d _ 1}$. Here we use uppercase $X$ to avoid confusion with our previous $x$ since they have slightly different dimensions (See Appendix 1 for details). Since WSE is conducted on the models that is close to a pre-trained model, e.g., $(W _ {a0}, W _ {b0})$, so we consider $f _ {dnn}(X) = (W _ {b0} + \Delta W _ b)^\top \delta( (W _ {a0} + \Delta W _ a)^\top X)$ where $\Delta W _ a$ and $\Delta W _ b$ is small and trainable. Two individual models are
> $\bar f _ {dnn}(X)  = (W _ {b0} + \Delta \bar W _ b)^\top \delta( (W _ {a0} + \Delta \bar W _ a)^\top X)$,
> $\tilde f _ {dnn}(X)  = (W _ {b0} + \Delta \tilde W _ b)^\top \delta( (W _ {a0} + \Delta \tilde W _ a)^\top X)$;
> and the weight space ensemble (WSE) and output output space ensemble (OSE) are as following:
> $f _ {dnn, wse}(X)  = (W _ {b0} + 0.5 (\Delta \bar W _ b + \Delta \tilde W _ b) )^\top \delta( (W _ {a0} + 0.5 (\Delta \bar W _ b + \Delta \tilde W _ b))^\top X)$, $f _ {dnn, ose}(X) = 0.5(W _ {b0} + \Delta \bar W _ b)^\top \delta( (W _ {a0} + \Delta \bar W _ a)^\top X) + 0.5(W _ {b0} + \Delta \tilde W _ b)^\top \delta( (W _ {a0} + \Delta \tilde W _ a)^\top X)$.
> Denoting $\delta'$ as the derivative of $\delta$. In Appendix D.2, we show $f _ {dnn, wse}(X) - f _ {dnn, ose}(X) = 0.25(\Delta \bar W _ {b} + \Delta \tilde W _ {b})\delta'(W _ {a0}^\top X)((\Delta \bar W _ a + \Delta \tilde W _ a)^\top X)  -  0.5(\Delta \bar W _ {b} \delta'(W _ {a0}^\top X)(\Delta \bar W _ a^\top X) + \Delta \tilde W _ {b} \delta'(W _ {a0}^\top X)(\Delta \tilde W _ a^\top X)).$ In our model, we have $f _ {wse} - f _ {ose}= 0.25(\bar w + \tilde w)^\top x (\bar \Phi + \tilde \Phi) - 0.5(\bar w^\top x \bar \Phi + \tilde w^\top x \tilde \Phi)$. We can see that $w$ is analogous to $\Delta W _ {b}$ and $\Phi$ is analogous to $\Delta W _ {a}$. They differ by a fixed matrix $\delta'(W _ {a0}^\top X)$, which is independent of the trainable parameter  $(\Delta  W _ a, \Delta  W _ b)$
>
> In summary, if we consider two individual 2-layer DNNs that lies in linear connectivity region where they are close to each other in the weight space, we can use our bi-linear model to explain the  OSE-WSE difference on these two DNNs.
>
> **Understanding the difference between weight and output space ensemble**. When $\bar \Phi$ and $\tilde \Phi$ select non-overlapped features, $f _ {wse}$ and $f _ {ose}$ makes the same prediction since  $x \bar \Phi \perp \tilde w$ and  $x \tilde \Phi \perp \bar w$ by Assumption 2 and further $(\bar w + \tilde w)^\top \left(x(\bar \Phi + \tilde \Phi)\right) \propto \bar w^\top x \bar \Phi + \tilde w^\top x \tilde \Phi$.  When $\bar \Phi$ and $\tilde \Phi$ learns overlapped features, WSE would be different from OSE: (a) for WSE, the coefficient of overlapped features is  amplified by 2 in $\bar \Phi + \tilde \Phi$, and further amplified twice in $\bar w +\tilde w$.
> This results in coefficient of the overlapped feature becoming 4 in $(\bar w + \tilde w)^\top x(\bar \Phi + \tilde \Phi)$. (b) for  OSE, i.e., $\bar w^\top x \bar \Phi + \tilde w^\top \tilde x \Phi$, the coefficient of the overlapped feature is 2. So if $\bar \Phi$ and  $\tilde \Phi$ learns more
> overlapped invariant features than overlapped spurious features, WSE would have better OOD performance than OSE (details shown in Appendix D.7.2). We further provide a discussion on why WSE is usually better than OSE in real-world applications, i.e., why $\bar \Phi$ and  $\tilde \Phi$ usually learn more
> overlapped invariant features than overlapped spurious features. Furthermore, we provide empirical verification on real-world datasets for our theoretical findings by showing that if $\bar \Phi$ and  $\tilde \Phi$ learns non-overlapped invariant features, WSE would be no better than OSE in OOD. Details of this part was included in Appendix D.7.3.

---

> ### Author Response · Authors · 2023-11-16
> **Response to Question 2/5**
>
> > Question 2 "This is a sizeable gap because WiSE-FT is weight space ensembling of very specific models (good OOD/bad ID and good ID/bad OOD). This is not reflected in the paper except loosely when they say low overlap is good weight space ensembles"
>
> Thank you for the questions.
>
> **The specific theoretical results for WiSE-FT**. Sorry for missing the results for the WiSE-FT which is the weight space ensembling of very specific models (good OOD/bad ID and good ID/bad OOD). Our theoretical framework can incorporate WiSE-FT as a special case.
>
> Recall Definition 2:  $\bar{f}$ learns $\bar{n} _ v$ invariant features and $\bar{n} _ s$ spurious features;  another single model $\tilde{f}$ has $\tilde{n} _ v$ invariant features and $\tilde{n}  _  s $ spurious features. Further, $\bar{f}$ and $\tilde{f}$ learns $n _ {vo}$ overlapped invariant features and $n _ {so}$ overlapped spurious features.  Let $\bar f$ denote the pre-trained model and $\tilde f$ denote the fine-tuned model. WiSE-FT is specifically is the following: $\bar f$ has good OOD but bad ID, $\tilde f$ has bad OOD but good ID, and the weight space ensemble of  $\bar f$ and $\tilde f$ has excellent OOD performance. These can be expressed as:
> (1) $\mathcal{A} _ {id} (\bar{f}) < \mathcal{A} _ {id} (\tilde{f})$; (2) $\mathcal{A} _ {ood} (\bar{f}) > \mathcal{A} _ {ood} (\tilde{f})$ and (3) $\mathcal{A} _ {ood} (\tilde{f} _ {wse}) > \max (\mathcal{A} _ {ood} (\bar{f}), \mathcal{A} _ {ood} (\tilde{f}) )$.
>
> It straightforward that the ID accuracy satisfies: $\mathcal{A} _ {id} (\bar{f}) < \mathcal{A} _ {id} (\tilde{f}), \text{if} \quad \bar{n} _ v + \bar{n} _ s < \tilde{n} _ v + \tilde{n} _ s.$ Intuitively, if a model learns more feature, it can predict the label better in the ID setting.  As for the OOD accuracy, by Proposition 2, we have $\mathcal{A} _ {ood}(\bar{f}) = F _ p ( \frac{(1-p)\bar{n} _ s + \bar{n} _ v}{\sqrt{\bar{n} _ s}} ),  \mathcal{A} _ {ood}(\tilde{f}) = F _ p ( \frac{(1-p)\tilde{n} _ s + \tilde{n} _ v}{\sqrt{\tilde{n} _ s}} ).$ Furthermore, by Proposition 3 the OOD accuracy of weight space ensemble (WSE) is
> $\mathcal{A} _ {ood} (f _ {wse}) = F _ p ( \frac{(1-p)(\bar{n} _ s + \tilde{n} _ s + 2 n _ {so}) + \bar{n} _ v + \tilde{n} _ v + 2 n _ {vo}}{\sqrt{\bar{n} _ s + \tilde{n} _ s + 14 n _ {so} }} )$. Then the WiSE-FT phenomenon can be effectively explained if the following conditions holds: (1) $\bar n _ v + \bar n _ s < \tilde n _ v + \tilde n _ s,$ (2) $\frac{(1-p)\bar{n} _ s + \bar{n} _ v}{\sqrt{\bar{n} _ s}}  > \frac{(1-p)\tilde{n} _ s + \tilde{n} _ v}{\sqrt{\tilde{n} _ s}}$ ,and (3) $\frac{(1-p)(\bar{n} _ s + \tilde{n} _ s + 2 n _ {so}) + \bar{n} _ v + \tilde{n} _ v + 2 n _ {vo}}{\sqrt{\bar{n} _ s + \tilde{n} _ s + 14 n _ {so} }} > \max (
>         \frac{(1-p)\bar{n} _ s + \bar{n} _ v}{\sqrt{\bar{n} _ s}}, \frac{(1-p)\tilde{n} _ s + \tilde{n} _ v}{\sqrt{\tilde{n} _ s}}
>         )$.
> We show the general conditions for WiSE-FT above. Here we use a concrete example for illustration: suppose $p = 0.9$ and there is no overlapped features learned by two models, i.e., $n _ {so} = n _ {vo} = 0$. The pretrained model $\bar f$ learns some invariant and spurious features, i.e., $\bar{n} _ v = 2$, $\bar{n} _ s = 4$. The fine-tuned $\tilde f$ model learns more spurious features and less invariant features, i.e., $\tilde{n} _ v =1, \tilde{n} _ s = 6$.  Then, the fine-tuned model $\tilde f$  has better ID accuracy than the pre-trained $\bar f$ since $\tilde{n} _ v + \tilde{n} _ s = 7 > \bar{n} _ v + \bar{n} _ s = 6$. The  $\tilde f$ has worse OOD accuracy than  $\bar f$ since $\tilde f$ focuses more on spurious features. Specifically, we have $\mathcal{A} _ {ood} (\bar{f}) > \mathcal{A} _ {ood} (\tilde{f})$ since
> $\mathcal{A} _ {ood}(\bar{f}) = F _ p \left( \frac{(1-p)\bar{n} _ s + \bar{n} _ v}{\sqrt{\bar{n} _ s}} \right) \approx F _ p(1.20)$,
> $\mathcal{A} _ {ood}(\tilde{f}) = F _ p \left( \frac{(1-p)\tilde{n} _ s + \tilde{n} _ v}{\sqrt{\tilde{n} _ s}} \right) \approx F _ p(0.97)$,
> Based on Proposition 3, the OOD accuracy of wse is
> $\mathcal{A} _ {ood} (f _ {wse}) = F _ p \left( \frac{(1-p)(\bar{n} _ s + \tilde{n} _ s + 2 n _ {so}) + \bar{n} _ v + \tilde{n} _ v + 2 n _ {vo}}{\sqrt{\bar{n} _ s + \tilde{n} _ s + 14 n _ {so} }} \right) \approx F _ p(1.27).$
> Recall that $F _ p(\cdot)$ is monotonically increasing,  we can see that  $\mathcal{A} _ {ood}(f _ {wse}) >  \max (\mathcal{A} _ {ood}(\tilde{f}),  \mathcal{A} _ {ood} (\bar{f}))$.
>
> Thanks for the suggestion! We included this in Appendix G, which makes our work more complete.
>
> **Clarification on "low overlap is good weight space ensembles"**.  Our main message is that both weight and output space ensemble can benefit if $\bar f$ and $\tilde f$ learns low overlap spurious features. Meanwhile, weight space ensemble can even outperform output space ensemble if $\bar f$ and $\tilde f$ learns more overlapped invariant features than overlapped spurious features.

---

> ### Author Response · Authors · 2023-11-16
> **Response to Question  3/5**
>
> > 3. Question: ``The authors acknowledge this even when they talk about their method "However, this method can not be directly applied to WiSE-FT since WiSE-FT ensemble model weights instead of the outputs.}''.
>
> Response:
>
> Thank you for the comment.
>
> **Clarifications**.  We'd like to clarify that our theoretical model (weight space ensemble) ensemble the weights of all the layers instead of the output, which has been extensively discussed above. By "However, this method can not be directly applied to WiSE-FT since WiSE-FT ensemble model weights instead of the outputs", we mean the direct temperature scaling of [1] can not be applied to WiSE-FT. However, our method and theoretical results operate on all the layers, can be applied to WiSE-FT.
>
> **Detailed Illustrations**. Specifically, in Proposition 4, we consider a overconfident individual model $\tilde f _ \lambda = (\lambda \tilde w, \lambda \tilde \Phi)$ parameterized by $\lambda$ and a normal model $\bar f = ( \bar w, \bar \Phi)$. (Recall that we use a two-layer bilinear model where $w \in \mathbb{R}^{d \times K}$ and $\Phi \in (0, 1)^{d _ s + d _ t}$).
> Notably,  $\tilde f _ \lambda$ is overconfident in both layers, leading to the skewed weight space ensemble which is imbalanced in each layer, i.e., $f _ {wse} = \frac{1}{4}(\bar w + \lambda \tilde w )^\top x (\bar \Phi + \lambda \tilde \Phi )$. Proposition 4 shows the OOD performance of $f _ {wse}$ due to the overconfidence caused by $\lambda$. In our paper, we argue that directly tuning the temperature (as adopted in [1]) is not applicable in WiSE-FT. Specifically,  the imbalanced issue exists in both layers as shown in the model $f _ {wse} = \frac{1}{4}(\bar w + \lambda \tilde w )^\top x (\bar \Phi + \lambda \tilde \Phi )$ in our Proposition 4. Tuning the temperature of the output of $(\lambda \tilde w, \lambda \tilde \Phi)$ is equivalent to tuning the scaling of $\tilde w$. For example, if we set the temperature as $1/\lambda$ which is equivalent to replacing $\tilde w$ with $\frac{1}{\lambda}\tilde w$, the weight space ensemble would be $f _ {wse} = \frac{1}{4}(\bar w + \tilde w )^\top x (\bar \Phi + \lambda \tilde \Phi )$, which still suffers from the overfitting issue in the $\Phi$ layer.
>
> **Summary**. In conclusion, our  weight space ensemble model is the ensemble of all layer weights, not just the output layer. Our theoretical model can illustrate the drawback of output temperature tuning adopted in [1] for WiSE-FT. Inspired by this, we propose our BANG by changing the fine-tuned procedure by incorporating MixUP and Label Smoothing, which can hopefully alleviate the over-confidence in all the layers. See response 5 for more details.
>
> Thank you for raising the question and sorry that there is might be some confusion. We will try our best to improve the readability.
>
> [1] Ananya Kumar et.al., Calibrated ensembles can mitigate accuracy tradeoffs under distribution shift

---

> ### Author Response · Authors · 2023-11-16
> **Response to Question  4/5**
>
> > Question: ``The paper does focus a lot of output ensembling, which is not known to work as well as
> WiSE-FT. In that case, there is no discussion of boosting, which seems to have the same flavor
> (especially when forcing models to rely on different features)"
>
> **Clarification on the scope of this paper**. The objective of our paper is twofold: to provide a comprehensive understanding and theoretical framework for ensemble-based methods, and to introduce a new perspective called spurious feature diversification for OOD generalization. Our theoretical framework encompasses output space ensembles, weight space ensembles (which involve ensembling weights across all layers), and can trivially cover methods based on feature concatenations [1].
>
>
> To clearly illustrate the core theoretical motivation, we initially focus on studying output space ensembles.  **In the later part of Section 3.2, as well as Appendices D.1, D.7.1, D.7.2, and D.7.3, we delve into a comprehensive investigation of weight space ensembles.** While we have already demonstrated the effectiveness of output space ensembles, we emphasize the distinctions between weight and output space ensembles. To the best of our knowledge, **our work is the first to explain why and when weight space ensembles outperform output space ensembles, supported by empirical evidence on real-world datasets**.
>
> Due to space constraints, we provide only a brief introduction to the main results of weight space ensembles, reserving the majority of the weight space ensemble discussion for the appendices. We appreciate your concern and assure you that we will make every effort to include more results on weight space ensembles in the main section.
>
> **Discussion on boosting**. It is very interesting to explore boosting in the context of OOD. According to our theory, boosting can benefit by training multiple models, where each model corrects the mistakes made by the previous ones and each model would possibly utilize different subsets of features. While previous studies on boosting mainly focused on ID scenarios [2, 3], we show that in the context of OOD, the improvement in performance due to using diverse features can be even more significant. This is because different irrelevant features can cause different errors when the distribution changes, and diversifying the features helps reduce the impact of each individual feature (as shown in Figure 1). By utilizing a diverse set of models, boosting allows us to take advantage of a wider range of features and effectively deal with the challenges posed by OOD situations.
>
> Thank you for your suggestion. We have added discussions in the related works. It is an exciting idea to explore boosting in the framework of spurious feature diversification. We will certainly look into it in our future work.
>
> [1] Jianyu Zhang et.al., Learning useful representations for shifting tasks and distributions.
>
> [2] Schapire  et.al., The Strength of Weak Learnability.
>
> [3] Yoav Freund et.al., A Decision-Theoretic Generalization of On-Line Learning and an Application to Boosting.

---

> ### Author Response · Authors · 2023-11-16
> **Response to Question  5/5**
>
> > Question: ``It is also unclear that Mix-up and label-smoothing have singular effects on the confidence, they also regularize. So BANG is really just finetuning with MIXup/LS and then doing WiSE-FT. A more direct correction of confidence would be the right one, like doing temperature scaling for each checkpoint during finetuning and then running WiSE-FT".
>
> Response: Thank you for your feedback. In fact, we have already conducted the experiments  in Appendix E.6 as you suggested. Specifically, in Appendix E.6, we demonstrate that re-scaling (down-weighting) each layer of the fine-tuned model significantly enhances the performance of WiSE-FT. This means that by manually reducing the over-confidence of the fine-tuned model, WiSE-FT can benefit greatly. Specifically, the OOD performance of vanilla WiSE-FT is 63.0% and we can increase its performance to over 64.5% when we re-scale each layer of the fine-tuned model to reduce its over-confidence.
> We apologize for placing this information in the appendix and only briefly mentioning it in the main section. In the revised version, we will provide more detailed coverage of these results in the main section.

---

> ### Author Response · Authors · 2023-11-17
> **Invitation for a Discussion**
>
> Dear Reviewer kd2C:
>
>    Thank you very much for taking the time to review our paper and for your valuable input.
>
>    We have made extensive clarifications, discussions, and provided experimental results as you indicated. We firmly believe we have effectively addressed your concerns.
>
>    We are eager to hear your valuable opinion on the efforts we have made during the rebuttal period. If you have any further questions that you would like us to address, we are more than willing to discuss them in detail.
>
>    Thank you once again for your invaluable feedback!

---

> ### Author Response · Authors · 2023-11-20
> **Waiting for your comments**
>
> Dear Reviewer kd2C:
>
> Thank you sincerely for dedicating your valuable time and efforts to reviewing our papers.
>
> We have provided extensive discussions, clarifications and empirical results as you indicated.
>
> With the deadline for discussions swiftly approaching, we eagerly await your feedback. Your input is of immense importance to us
>
> Thank you once again!

---

> ### Author Response · Authors · 2023-11-22
>
> Dear review kd2C:
>
> Thank you for your time and effort in reviewing our paper.
>
> We firmly believe that our response and revisions can fully address your concerns. We are open to discussion if you have any additional questions or concerns, and if not, we kindly ask you to reevaluate your score.
>
> Thank you again for your reviews which helped to improve our paper!
>
> Authors

---

> ### Author Response · Authors · 2023-11-23
>
> Dear review kd2C:
>
> Thanks for your valuable time for reviewing our paper.
>
> Since the ddl of the discussion period is fastly approaching, could you kindly take a look at the responses? We have provided extensive discussions, clarifications and empirical results as you indicated.
>
> If you have any additional questions or require further clarification, we are more than willing to engage in discussion within the next few hours.
>
> Authors

---

### Author Response · Authors · 2023-11-19
**General Response and Invitation for a Discussion**

We sincerely appreciate the time and efforts of the reviewers in providing their valuable feedback. We have incorporated the suggested modifications in our script, which are highlighted in blue.

The key points of our rebuttal can be summarized as follows:

1. Provide extensive discussions and clarifications from the following aspects

	1.1 For the theoretical model of weight space ensemble, we ensemble the weights of all layers, not just the final linear layer ([link](https://openreview.net/forum?id=d6H4RBi7RH&noteId=Ml3oNbIkyF))

	1.2 Our theoretical model for weight space ensemble is closely connected to two-layer DNNs that lie in linear connctivity regions ([link](https://openreview.net/forum?id=d6H4RBi7RH&noteId=e1vBJdh0Lu)).

	1.3 Our novel contributions on the understanding weight space ensemble with empirical verification on real-world datasets ([link](https://openreview.net/forum?id=d6H4RBi7RH&noteId=e1vBJdh0Lu)).

2. Additional theretical results for WiSE-FT which is the weight space ensemeble of specific models (good OOD/bad ID and good ID/bad OOD, [link](https://openreview.net/forum?id=d6H4RBi7RH&noteId=iT2Zou2RIb)) .

3. Additional experiments to verify BANG's effectivenss through the lens of "Accuracy on the line" ([link](https://openreview.net/forum?id=d6H4RBi7RH&noteId=sKcOIGy3JU)).

4. Additional experiments that we can signficantly improve the ensemble performance on MultiColorMNIST further by incorperating a larger number of individual models in ensemble ([link](https://openreview.net/forum?id=d6H4RBi7RH&noteId=Oplpad08jZ)).

We eagerly await feedback from the reviewers and look forward to engaging in a fruitful discussion.

---

### Meta-Review · Area_Chair_BkHS · 2023-12-15

**Metareview:**

The paper provides a clean perspective of fine-tuning and ensembling via the hypothesis that fine-tuned models tend to use diverse spurious features. This provides a neat way to tie in some empirical (known) results about fine-tuning and ensembling. All this is formalized in a stylized simple model. The authors also provide a slightly better ensembling that provides gains OOD.

The claims of the paper are sound and all experimental results seem convincing and thorough. Majority of the reviewers recommend acceptance and the one reviewer who recommends reject seems to raise concerns that are addressed and not fatal flaws.

The main weakness of the paper is that it does not provide a very novel insight nor does it explain what I think is the main interesting piece here - *why* do fine-tuned models end up learning diverse spurious features? (this doesn't happen when training from scratch in practice). I also think the methodological contributions are somewhat incremental.

**Justification For Why Not Higher Score:**

See weaknesses above: contribution isn't too novel or significant and I don't see a clear argument for why these would lead to interesting methodogical innovations down the line

**Justification For Why Not Lower Score:**

Claims are sound, experiments thorough, majority reviewers positive, timely topic, generally well presented

---

### Decision · Program_Chairs · 2024-01-16

Accept (poster)